# Re2IC: Realism-Enhanced Region-Based Implicit Codec with Wavelet–Wasserstein Distortion

## Abstract

Recent advances in perceptual image compression improve quality through powerful probabilistic models but often incur prohibitive decoding costs. This paper proposes a low-complexity alternative called **Re**alism-enhanced **Re**gion-based **I**mplicit **C**odec, Re$^2$IC, that models visual perception region-by-region with saliency-guided partitioning. To further enhance realism, we introduce wavelet–Wasserstein distortion (WA-WD), which balances fidelity and perception through explicit frequency-aware modeling. Together, these components enable fine-grained spatial–spectral optimization, allowing Re$^2$IC to achieve superior rate–perception (RP) trade-offs and outperform generative codecs like HiFiC while using less than $1\%$ of their decoding cost. Re$^2$IC achieves faster encoding convergence, low decoding, and high-quality reconstructions that preserve both fine textures and natural details. Experiments and user studies show that Re$^2$IC sets a new state of the art in RP performance among overfitted codecs. Beyond compression, WA-WD also serves as a standalone, tunable metric, that aligns more closely with human preference, achieving the highest Pearson ($94.6\%$) and Spearman ($92.3\%$) correlations with Elo scores, and demonstrating leading performance across multiple IQA datasets.

## 1 Introduction

Image compression has long relied on hand-crafted transforms to capture statistical correlations and perceptual redundancies (Sullivan et al., 2012; Bross et al., 2021). Recent deep learning approaches replace these with neural networks (Ballé et al., 2018; Cheng et al., 2020; He et al., 2022), achieving strong rate–distortion (RD) gains. However, optimizing RD alone often fails to preserve perceptual quality, especially at low bit rates, which has spurred growing interest in perceptual compression.

Perceptual quality, or "realism" (Hamdi et al., 2025; Qiu et al., 2024), measures how closely reconstructions resemble natural images. Prior work (Blau & Michaeli, 2019) shows a fundamental trade-off between rate, distortion, and perception, implying that optimizing RD alone cannot guarantee naturalness. To bridge the gap, the recent learned codecs shift from deterministic decoding to sampling from conditional distributions with generative priors. GAN (Mentzer et al., 2020) and diffusion-based approaches (Theis et al., 2022) produce compelling reconstructions but often rely on complex models and large-scale training, often exceeding the cost of commercial codecs. Achieving perceptual quality at low complexity, therefore, remains a pressing open challenge.

Meanwhile, overfitted codecs provide a low-complexity alternative by fitting each image with compact implicit neural representations (INRs) (Sitzmann et al., 2020; Dupont et al., 2021; Ladune et al., 2023; Kim et al., 2024; Wu et al., 2025), achieving promising RD performance with minimal decoding cost. However, their perceptual potential remains largely unexplored. Recent studies show that Wasserstein distortion (WD) (Qiu et al., 2024; Ballé et al., 2025) improves rate–perception (RP) trade-offs by unifying fidelity and realism through saliency-weighted WD optimization. However, this paradigm assumes independence across features for tractability, neglecting feature correlations. Focusing solely on saliency for WD optimization may also overlook regional structure and frequency characteristics. As a result, it may over-resample textures in smooth, non-salient areas (Fig. 31) or enforce excessive fidelity in high-frequency salient regions (Fig. 32). Additionally, further percep-

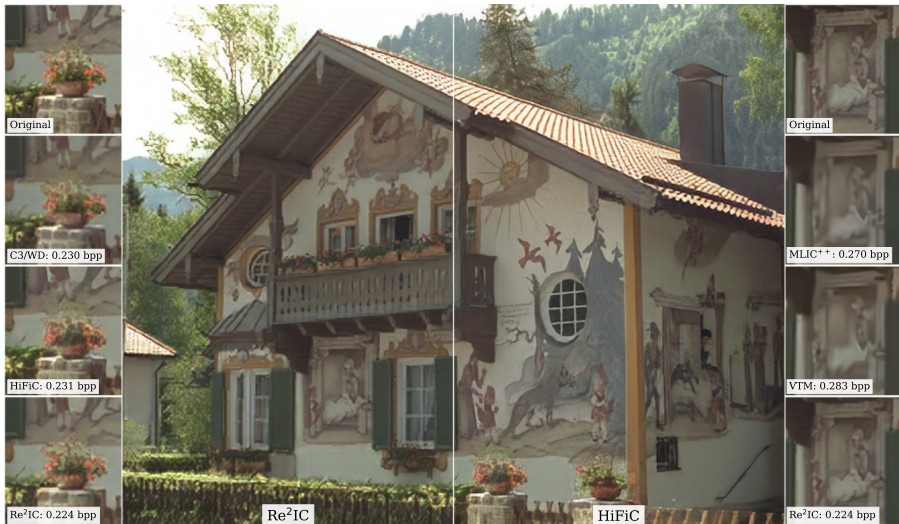

Figure 1: Visual comparison between codecs: Re$^2$IC delivers the best perceptual realism, closely matching the original image, while C3/WDs and HiFiC exhibit artifacts. Even with higher bit budgets, codecs such as VTM and MLIC$^{++}$ lose texture. Much more static and dynamic comparisons across methods, bitrates, and datasets are provided in the Appendix and supplementary materials.

tion gains in realism typically require more complex INRs or costly optimization strategies. These limitations highlight the need for a new framework that explicitly incorporates regional structure and frequency-aware saliency to achieve perceptual quality at low complexity.

Human visual perception is characterized by spatial and spectral non-uniformity, showing fine detail recognition in the fovea, lower resolution in the periphery, with regional biases further amplified by attention to salient, frequency-specific content. Motivated by this, we propose Re$^2$IC, a ***Realism-enhanced Re**gion-based **I**mplicit **C**odec*, that models visual perception in a region-wise manner. At its core, Re$^2$IC introduces two main innovations: (1) *Region-wise perceptual modeling*: improving RP trade-offs with faster convergence and cheaper decoding; and (2) *Wavelet–Wasserstein control*: decomposing WD into subbands to balance fidelity and realism for fine-grained optimization.

Rather than overfitting the entire image with a single network, Re$^2$IC partitions it into salient regions, each modeled by a dedicated local perceptual network (LPN). Arbitrary region boundaries are efficiently encoded using low-cost chain coding (Chang et al., 1992), with the decoded contour for identifying the corresponding LPN. A global perceptual modulator (GPM) is introduced to provide global context to mitigate artifacts from purely local modeling. This design simplifies coding, accelerates convergence, and keeps each LPN lightweight, thereby lowering both encoding and decoding costs. Finally, Re$^2$IC decomposes perception modeling into wavelet subbands, enabling explicit control over fidelity and realism across frequencies. The multi-scale wavelet structure naturally supports WD approximation, while saliency-guided band adaptation provides fine-grained bit allocation that accounts for both regional structure and intra-region frequency characteristics, yielding perception-enhanced compression at low complexity (see Fig. 1).

Compared with competitive neural codecs (Jiang & Wang; Fu et al., 2024), generative-based methods (Mentzer et al., 2020), diffusion-based methods (Lee et al., 2024; Yang & Mandt, 2023), and prior overfitted codecs (Ballé et al., 2025), Re$^2$IC delivers higher realism at lower decoding complexity, achieving superior RP performance across user studies and various quantitative metrics. Notably, it surpasses HiFiC (Mentzer et al., 2020) with less than $1\%$ of its decoding cost. Leveraging wavelet-WD optimization and local–global perceptual modulation, Re$^2$IC enables fine-grained fidelity–realism control, faster training, and low decoding complexity, challenging the long-held trade-off between quality, speed, and efficiency. Our contributions are as follows:

- We propose Re$^2$IC, a realism-enhanced implicit codec that models visual perception region by region, achieving strong RP performance while maintaining low decoding complexity.

- Imitating visual perception, Re$^2$IC integrates saliency-based partitioning with local–global perceptual modulation, and applies wavelet–Wasserstein optimization for fine-grained spatial–spectral control, enabling flexible bit allocation and efficient coding.

- User studies and extensive quantitative evaluations show that Re$^2$IC achieves state-of-the-art RP performance among overfitted codecs, surpassing HiFiC with lower decoding cost.

- We introduce wavelet–Wasserstein distortion as a standalone, tunable perceptual metric, which adapts to diverse scenarios and aligns strongly with human ratings (PCC 94.6%, SRCC 92.3%). Additional image quality assessment experiments further confirm its leading performance as a general perceptual metric.

## 2 RELATED WORK

**Overfitted codec.** Early work, such as (Dupont et al., 2021) trained INRs with quantized parameters forming the bitstream, later extended via meta-learning (Dupont et al., 2022a). COMBINER (Guo et al., 2023) improved RD with variational INRs and relative entropy coding. COOL-CHIC and successors (Ladune et al., 2023; Kim et al., 2024; Blard et al., 2024) introduced learnable latents, entropy models, and architectural advances, surpassing HEVC (Sullivan et al., 2012). More recently, LotteryCodec (Wu et al., 2025) explored random networks for better RD and flexible complexity, and (Ballé et al., 2025) incorporated saliency-guided Wasserstein distortion for RP optimization. Nevertheless, perception-oriented overfitted codecs remain underdeveloped and fall short of generative methods, underscoring the need for realism-enhanced yet low-cost designs.

**Region partitioning for INRs.** INRs model signals as continuous functions and benefit greatly from spatial decomposition, as shown by input partitioning (Jiang et al., 2020; Tretschk et al., 2020), local latent modulation (Mehta et al., 2021), and coarse-to-fine strategies (Liu et al., 2023; Ashkenazi & Treister, 2024). In the compression setting, although region- and shape-based coding attracted research interest in the past (Schmalz, 2004), its adoption was constrained by the difficulty of designing transforms for arbitrary shapes (Sikora et al., 1995; Li & Li, 2000). In contrast, INRs naturally support arbitrary shapes by training directly on irregular regions, and with explicit chain-coded contour representations (Chang et al., 1992), they are well-suited for modeling non-uniform human perception, enabling perception-enhanced region-based coding.

**Perception neural codec.** Early realism-enhanced codecs employed GANs such as HiFiC (Mentzer et al., 2020) and MS-ILLM (Muckley et al., 2023), as well as latent generative models (Jia et al., 2024), to synthesize visually plausible reconstructions at low bitrates. More recent methods leverage diffusion priors (Yang & Mandt, 2023; Hoogeboom et al., 2023; Vonderfecht & Liu; Ohayon et al.) and text-to-image capabilities (Careil et al., 2023; Jiang et al., 2023; Lee et al., 2024) for even stronger realism at extremely low bpp. However, these approaches require orders-of-magnitude higher computation than commercial codecs, limiting their practicality. This motivates a realism-enhanced implicit codec that models perception directly, rather than learning a generative distribution, achieving competitive RP performance at far lower decoding complexity.

## 3 METHODOLOGY

### 3.1 PRELIMINARY.

**Region-based overfitted codec.** Re$^2$IC adopts a region-based perceptual coding framework, partitioning each image into $N$ regions $\{\boldsymbol{S}^1, \ldots, \boldsymbol{S}^N\}$, each defined by a contour $\boldsymbol{\tau}^i$ and spatial coordinates $\boldsymbol{x}^i$. Each region is independently modeled using a dedicated INR $f_{\boldsymbol{W}^i}$ and a latent code $\boldsymbol{z}^i$, both quantized and entropy-coded (see Fig. 2). Contours $\boldsymbol{\tau}^i$ are explicitly compressed using chain coding. During decoding, they guide the selection of $\{\boldsymbol{x}^i, \boldsymbol{z}^i, \boldsymbol{W}^i\}$ for region-wise reconstruction via $\hat{\boldsymbol{S}} = \bigcup_{i=1}^{N} f_{\boldsymbol{W}^i}(\boldsymbol{x}^i, \hat{\boldsymbol{z}}^i)$, where $\bigcup$ denotes the merging of all regions and $\hat{\boldsymbol{z}}^i$ is the quantized latent codes. The perceptual distortion and the rate cost can be expressed as:

$$P = \mathbb{E}_{\boldsymbol{S} \sim p_s}[d(\boldsymbol{S}, \hat{\boldsymbol{S}})], \quad R = \mathbb{E}_{\boldsymbol{S} \sim p_s}\left[\sum_{i=0}^{N}\left[-\log_2 p_{\hat{\psi}}(\hat{\boldsymbol{z}}^i) - \log_2 p(\hat{\boldsymbol{W}}^i) - \log_2 p(\boldsymbol{\tau}^i)\right] + R_{\hat{\psi}}\right],$$

$$(1)$$

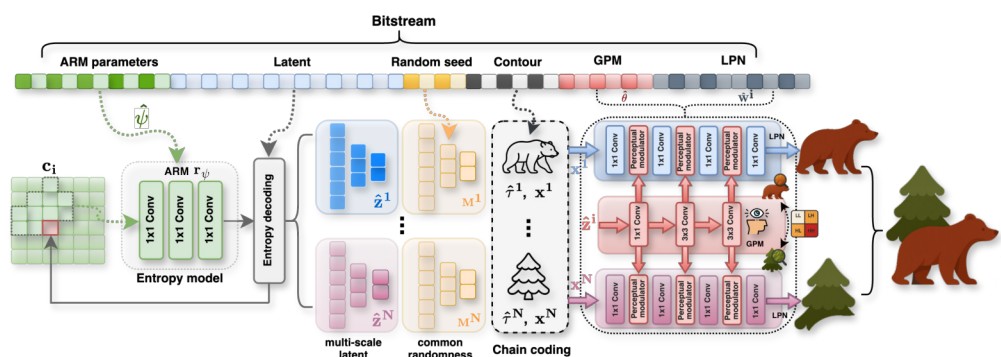

Figure 2: Workflow of Re²IC: the image is partitioned into $N$ regions, each reconstructed by a local perceptual network. A shared global perceptual modulator provides global modulation for perception compensation, using latent vectors decoded from a shared entropy model.

where $d$ denotes a perceptual distance function, $p_{\hat{\psi}}$ is a shared entropy model for all $\hat{z}^i$, and $R_{\hat{\psi}}$ accounts for transmitting its parameters.

**Wasserstein Distortion.** WD (Qiu et al., 2024) measures perceptual differences in feature space by explicitly modeling *foveal* and *peripheral* sensitivity through a spatially varying $\sigma$-map. Derived from saliency, this $\sigma$-map modulates tolerance to *texture resampling*, thereby enabling *flexible, perception-guided* bit allocation in image compression. Formally, local WD for a feature map $z$ at location $(x, y)$ is defined as the 2-Wasserstein distance between feature distributions within a pooling window. The pooling weights follow a family of probability mass functions (PMFs) $q_\sigma(k)$ parameterized by the pooling width $\sigma(x, y)$. This yields locally pooled measures $y_\sigma$ over which WD is evaluated, and the final distortion is obtained by spatial averaging across all locations.

For efficiency, $y_\sigma$ are often approximated as *independently distributed Gaussians* (Olkin & Pukelsheim, 1982). Under this approximation, the local WD between each feature component $z_i$ and its reconstruction $\hat{z}_i$ is: $D_{i,\sigma} = (\mu_i - \hat{\mu}_i)^2 + \left(\sqrt{V_i} - \sqrt{\hat{V}_i}\right)^2$, where $\mu_i = \sum_k q_\sigma(k)\, z_{i+k}$, and $V_i = \sum_k q_\sigma(k)\left(z_{i+k} - \mu_i\right)^2$ denote the local mean and variance of each element, with $k \in \mathbb{Z}$ indexing spatial offsets within the pooling window determined by $\sigma$-map. The final WD is averaged over all locations $\mathcal{I}$ as: $WD(z, \hat{z}) = \frac{1}{|\mathcal{I}|} \sum_{i \in \mathcal{I}} D_{i,\sigma}$. **Detailed derivations, interpretations, theoretical properties, and practical implementations are provided in Appendix D.1.**

### 3.2 Wavelet Wasserstein Distortion

While WD provides a principled foundation for perceptual compression, its independence assumption often fails in neural feature spaces, and saliency-only optimization overlooks essential regional and frequency structure. These limitations motivate wavelet-WD (WA-WD), a more interpretable and perception-consistent modeling approach for implicit codecs. **Detailed motivations, comparisons, theoretical properties, and validation experiments are provided in Appendix D.2.**

**Formulation.** WA-WD measures perceptual differences in a frequency-decomposed feature space, producing subband-aware distortion signals that enable fine-grained control and more flexible bit allocation. Specifically, WA-WD applies an orthonormal discrete wavelet transform (DWT) to decompose feature maps into orthogonal subbands and then computes WD. This absorbs intra-block correlations into subband-specific variances, making the Gaussianized independence assumption more accurate. Subband-wise evaluation naturally disentangles fidelity (LL) from realism (LH/H-L/HH), and multi-level decompositions further decorrelate structural and textural statistics.

Consider a $2 \times 2$ feature block from the source image $\mathbf{x}$ with vectorized form $\mathbf{w}_i \in \mathbb{R}^4$. The Haar transform projects this block onto basis vectors $\boldsymbol{h_b} \in \mathbb{R}^4$, yielding $z_i^b = \boldsymbol{h_b}^\top \boldsymbol{w_i}$, with $b \in \{LL, HL, LH, HH\}$. Given the pooling PMF $q_\sigma$, the resultant pooled mean and variance

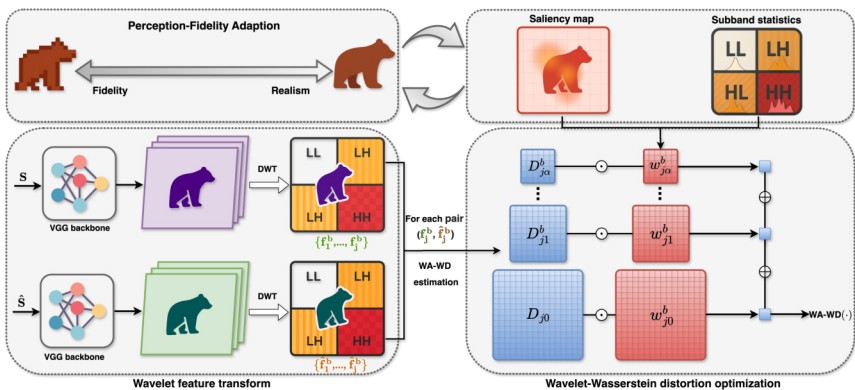

Figure 3: Illustration of WA-WD: Paired images are processed by a VGG backbone and DWT, decomposing features into orthogonal subbands for multi-scale WA-WD estimation. Combined with saliency and subband statistics, WA-WD enables fine-grained realism–fidelity optimization.

are: $\mu_i^b = \sum_k q_\sigma(k) z_{i+k}^b = h_b^\top \left( \sum_k q_\sigma(k) w_{i+k} \right) \triangleq h_b^T \bar{w}_i$, $V_i^b = \sum_k q_\sigma(k) \left( z_{i+k}^b - \mu_i^b \right)^2 = h_b^T C_i h_b$, where $C_i \triangleq \sum_k q_\sigma(k)(w_{i+k} - \bar{w}_i)(w_{i+k} - \bar{w}_i)^\top$ denote a pooled $4 \times 4$ *block covariance matrix*. Similarly to WD, with independence assumption, the per-band WD and the overall WA-WD are given as: $D_{i,\sigma}^b = (\mu_i^b - \hat{\mu}_i^b)^2 + \left( \sqrt{V_i^b} - \sqrt{\hat{V}_i^b} \right)^2$ and $WD_{\text{wave}}(z, \hat{z}) = \frac{1}{4} \sum_{b \in \{LL, LH, HL, HH\}} \frac{1}{|\mathcal{I}_b|} \sum_{i \in \mathcal{I}_b} D_{i,\sigma}^b$, respectively. We use equal subband weights by default for simplicity, though they can be tuned for improved perceptual evaluation or different degradations and bpp regimes, as detailed in the later section.

**Theoretical properties.** A core design principle of WA-WD is to maintain the desirable theoretical guarantees of WD while improving perceptual alignment and region-wise controllability. Since DWT is orthonormal and invertible, applying WD in transformed domain preserves metric structure.

**Theorem 1** (Metric preservation). *Let $W \in \mathbb{R}^{d \times d}$ be an orthonormal DWT and define $WD_{\text{wave}}(z, \hat{z}) \triangleq WD(Wz, W\hat{z})$. Then: (1) If the PMFs $q_\sigma$ have no spectral nulls and $d(\cdot)$ is a metric (i.e, $WD(z, \hat{z})^{1/p}$ is a metric), then $WD_{\text{wave}}(z, \hat{z})^{1/p}$ is a metric. (2) If the feature extractor $\phi(\cdot)$ is invertible (i.e, $WD(x, \hat{x})^{1/p}$ is a metric), then $WD_{\text{wave}}(x, \hat{x})^{1/p}$ is also a metric.*

Theorem 1 shows that WA-WD preserves the metric structure, supporting its role as a perceptual measure. Beyond this, its diagonal-Gaussian approximation bound becomes tighter after an orthonormal DWT due to its strong decorrelating effect, as formalized in the following Corollary:

**Corollary 1** (Tighter bound for diagonal-Gaussian approximation). *Let $X$ and $Y$ be local Gaussian patches with covariances $\Sigma_X$ and $\Sigma_Y$. Let $\widetilde{WD}$ and $\widetilde{WD}_{\text{wave}}$ denote their diagonal-Gaussian approximations in original and wavelet domains. They both satisfy error bounds of the same form:*

$$|WD(X, Y) - \widetilde{WD}(X, Y)| \leq C(\|\text{offdiag}(\Sigma_X)\|_F + \|\text{offdiag}(\Sigma_Y)\|_F),$$

$$|WD_{\text{wave}}(X, Y) - \widetilde{WD}_{\text{wave}}(X, Y)| \leq C\left(\|\text{offdiag}(W\Sigma_X W^\top)\|_F + \|\text{offdiag}(W\Sigma_Y W^\top)\|_F\right),$$

*where the constant $C > 0$ follows Lemma 1 and $\text{offdiag}(\Sigma) \triangleq \Sigma - \text{diag}(\Sigma)$. Since wavelet coefficients of natural images are typically far more decorrelated, the off-diagonal terms of $W\Sigma_X W^\top$ and $W\Sigma_Y W^\top$ are much smaller, yielding a tighter error bound in practice.*

**Lemma 1.** *The constant $C$ in Corollary 1 depends only on the eigenvalue bounds of $\Sigma_X, \Sigma_Y$. If the eigenvalues lie in $[m, M]$, an admissible choice is $C = 2\sqrt{d}(1 + \frac{M}{2m} + \frac{M^{3/2}}{2m^{3/2}})$*

Consequently, the error bound in Corollary 1 becomes tighter in the wavelet domain, indicating that the diagonal-Gaussian approximation in WA-WD is statistically more accurate. Evaluating WD within individual wavelet subbands places the computation in a domain where the independence assumptions hold more closely, resulting in more stable and perceptually consistent optimization.

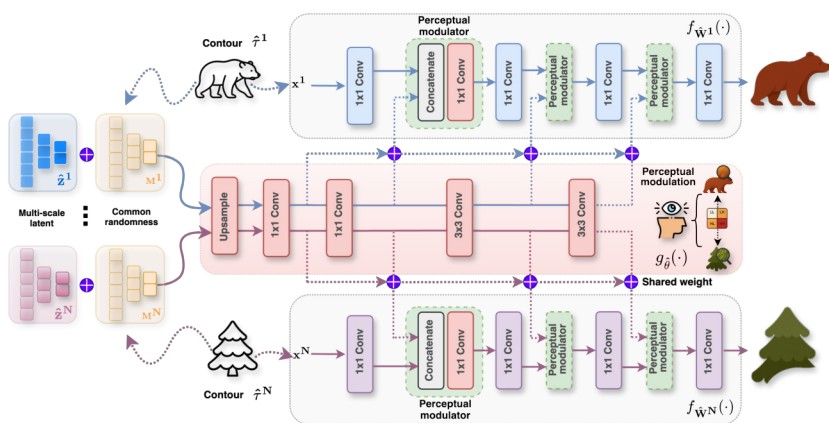

Figure 4: Global-local design of Re²IC: each LPN is optimized with WA-WD to model visual perception, while red GPM blocks denote parameter-shared components that inject global context by modulating multi-scale features, thereby mitigating artifacts from pure local overfitting.

**Corollary 2** (Statistical properties of WA-WD). *WA-WD satisfies the following statistical properties: it (a). preserves the same first-order statistics as WD; (b). captures intra-block correlations that WD ignores under its diagonal-Gaussian approximation; (c) reduces to MSE when $\sigma = 0$.*

Corollary 2 shows that WA-WD explicitly captures intra-block correlations ignored by WD under diagonal-Gaussian approximation. Additional examples are given in Table 5. All proof of theoretical properties of WA-WD are in Appendix E. More visualizations for validation are in Appendix F.4.

**Practical implementation.** WA-WD is approximated by pre-computing local moments at discrete pooling scales, same as in (Ballé et al., 2025). As shown in Fig.3, original image $S$ and its reconstruction $\hat{S}$ are passed through a VGG backbone and a Haar wavelet transform, yielding multiscale features $f_j^b$ and $\hat{f}_j^b$. Each feature pair is further downsampled by with scale factor $\alpha$ to compute element-wise first- and second-order moments (Eqns.33, 34), from which a WD map $D_{j\alpha}^b$ is obtained with associated pooling scales $\sigma_{j\alpha}^b$. Similarly, $\sigma_{j\alpha}^b$ are derived from EML-Net saliency maps, where scores $s \in [0,1]$ are converted into a spatial likelihood $p = p_{\min} + (1 - p_{\min}) \cdot \frac{s}{\bar{s}}$, with $p_{\min} = 0.5$ ensuring positivity and unit mean. The sigma field is then set as $\sigma = \sigma_{\max}^b \cdot \frac{p_{\min}}{p(x,y)}$, with $\sigma_{\max}^b$ subband-dependent. The integration of VGG and DWT yields multiscale features, naturally aligned with pooling widths, serving as references for estimating WD at arbitrary $\sigma$. WA-WD is then interpolated with a weight map $w_{j\alpha}^b$ (with elements $\max\!\big(0, 1 - |\log_2 \sigma_{j\alpha}^b - \alpha|\big)$), and aggregated across different feature and subbands as: $WD_{\text{wave}} = \sum_{b \in \{\text{LL,LH,HL,HH}\}} \sum_{j,\alpha} \big(w_{j\alpha}^b \odot D_{j\alpha}^b\big)$. (For a complete illustration of the WD approximation, we refer to (Ballé et al., 2025), which adopts the same approach as ours.)

### 3.3 Re²IC ARCHITECTURE: REGION-WISE PERCEPTION MODELING

As shown in Fig. 2, Re²IC comprises five components: **region partition**, **adaptive chain coding**, **local–global perceptual networks** (LPNs $f_{W^i}$ and GPM $g_\theta$), **latent vectors** ($z^i$ and randomness prior $M^i$), and a **shared entropy model** ($r_\psi$). Details of each component are in Appendix C.

**Region partition.** To encode an image, Re²IC partitions images into regions using saliency predictions. An EML-Net (Jia & Bruce, 2020) is employed to generate a saliency map, from which we threshold and extract $N - 1$ largest connected components as sub-regions for LPN allocation.

**Adaptive chain coding.** To compress contours efficiently, we adopt adaptive chain coding (Chang et al., 1992) (open-sourced in supplementary materials). It encodes curves with a starting point and directional steps, adapting to curvature for low bit cost. In Re²IC, we further apply contour dilation,

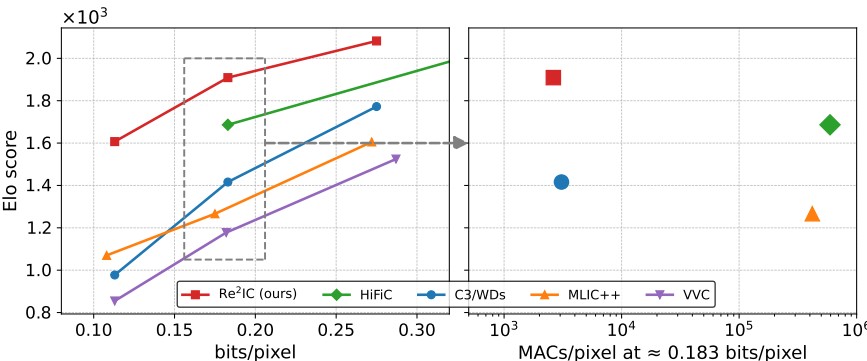

Figure 5: Human rating results and decoder complexity on Kodak. Left: Evaluation of different methods vs. bit rate. Right: Decoding complexity at the middle bit-rate regime. $\text{Re}^2\text{IC}$ achieves the best perceptual quality by human preference while maintaining the lowest decoding complexity.

as saliency maps typically lack precise pixel-level semantics. Combined with GPM, this improves robustness to lossy contours and high-curvature boundaries while reducing cost.

**Latent vectors.** For the $i$-th region, the learnable latent vectors are defined as $z^i \triangleq \{z_1^i, z_2^i, \ldots, z_L^i\}$, where each $z_j^i \in \mathbb{R}^{\frac{K^i}{4^{j-1}}}$ corresponds to a $2^{j-1} \times 2^{j-1}$ downsampling of the region, with $K^i$ denoting its pixel count. To improve perceptual quality, we introduce common randomness via a Gaussian matrix $M^i \sim \mathcal{N}(0,1)$ (same shape as $z^i$), generated synchronously by the same random seed at both ends without coding cost (Ballé et al., 2025). The decoded $z^i$ and $M^i$ are concatenated and fed into the GPM for perceptual compensation.

**Local-global perceptual networks.** Each LPN reconstructs pixel values from region coordinates $x^i$, while the GPM injects global context using the quantized latent vectors as input. As shown in Fig. 4, each $f_{W^i}$ consists of four convolutional layers, each modulated by $\mathcal{M}(\cdot)$, which fuses the layer output with a vector from $g_\theta$ through a shared MLP. Sharing $\mathcal{M}(\cdot)$ across regions reduces bitrate and implicitly blends local and global features for improved perceptual modeling. The GPM $f_\theta$ itself comprises an upsampling module followed by four convolutional layers.

### 3.4 RATE-PERCEPTION OPTIMIZATION.

To balance RP trade-off, $\text{Re}^2\text{IC}$ with an $N$-region configuration is trained to overfit each datapoint into the parameter set: $\Omega \triangleq \left\{ \psi,\ \theta,\ \bigcup_{i=1}^{N} \left( z^i,\ \tau^i,\ W^i \right) \right\}$, by modeling visual perception with the following loss: $\mathcal{L}(\Omega) = \left[ WD_{\text{wave}}(S^i, \bigcup_{i=1}^{N} \hat{S}^i) + \lambda R(\hat{z}^i) \right]$, where $\hat{S}^i = f_{W^i}(f_\theta(\hat{z}^i, M^i), x^i)$ denotes the $i$-th region reconstruction, $WD_{\text{wave}}(\cdot)$ is Wavelet–WD, $R(\hat{z}^i)$ is the latent rate term, and $\lambda$ is a trade-off hyperparameter. Further implementation details are given in Appendix C.2.

## 4 EXPERIMENTAL RESULTS

### 4.1 EXPERIMENT SETUP

We evaluate $\text{Re}^2\text{IC}$[1] on the Kodak (Kodak, 1991) and the CLIC2020 professional validation set (Toderici et al., 2020), using both human rater studies and quantitative assessments. Baselines include the classical codec VTM 19.1 (Bross et al., 2021), state-of-the-art autoencoder codecs $\text{MLIC}^+$ and $\text{MLIC}^{++}$ (Jiang & Wang), the generative codec HiFiC (Mentzer et al., 2020), and perception-enhanced overfitted codec C3/WDs (Ballé et al., 2025). For each method, we target $\{0.075, 0.15, 0.3\}$ bpp on CLIC, and on Kodak $\{0.113, 0.183, 0.275\}$ bpp to align with open-sourced baselines. More diffusion and DWT-based baselines, such as TACO (Lee et al., 2024), CDC (Yang & Mandt, 2023), and WeConvene (Fu et al., 2024) are provided in the Appendix

---

[1]In all experiments, $\text{Re}^2\text{IC}$ is trained for 80k steps with fast convergence though further gains are possible.

Table 1: Quantitative comparison of different methods on CLIC2020. We report performance on PSNR↑, MS-SSIM↑, LPIPS↓, DISTS↓, FID↓,KID↓, NIQE↓, MUSIQ↑, ClipIQA↑, and MANIQA↑. **Red** and blue indicate the best and second-best results, respectively. VTM and CDC is excluded from the high-BPP regime comparisons since its rate is significantly higher or lower.

| | Model | BPP↓ | PSNR↑ | MS-SSIM↑ | LPIPS↓ | DISTS↓ | FID↓ | KID↓ | NIQE↓ | MUSIQ↑ | ClipIQA↑ | MANIQA↑ | WD↓ |
|---|---|---|---|---|---|---|---|---|---|---|---|---|---|
| Low | VTM | 0.0790 | 30.37 | 0.9440 | 0.3205 | 0.2196 | 114.88 | 57.29 | 5.9644 | 52.96 | 0.3471 | 0.3263 | 2.5756 |
| | MLIC$^+$ | 0.0780 | **30.92** | **0.9488** | 0.3035 | 0.2201 | 105.93 | 46.66 | 5.9993 | **57.11** | 0.4321 | **0.3506** | 2.4406 |
| | C3/WDs | 0.0740 | 25.97 | 0.8972 | 0.1228 | 0.0844 | 24.57 | 5.79 | 5.1572 | 53.40 | 0.4773 | 0.3199 | 1.3082 |
| | HiFiC | - | - | - | - | - | - | - | - | - | - | - | - |
| | Re$^2$IC | 0.0750 | 26.57 | 0.9072 | **0.1136** | **0.0708** | 17.16 | 2.34 | 4.3611 | 55.58 | 0.5226 | 0.3352 | 1.1131 |
| Medium | VTM | 0.1480 | 32.39 | 0.9665 | 0.2495 | 0.1837 | 89.83 | 47.54 | 5.3504 | 57.34 | 0.4230 | 0.3521 | 1.1709 |
| | MLIC$^+$ | 0.1490 | **32.99** | **0.9703** | 0.2334 | 0.1840 | 85.65 | 38.75 | 5.3692 | 59.84 | 0.4777 | **0.3539** | 1.5845 |
| | C3/WDs | 0.1490 | 27.94 | 0.9357 | 0.0767 | 0.0459 | 13.71 | 2.63 | 4.7688 | 56.38 | 0.5140 | 0.3352 | 0.6487 |
| | HiFiC | 0.1420 | 29.83 | 0.9594 | **0.0589** | 0.0695 | 18.81 | 2.33 | **3.3044** | 59.91 | 0.5140 | 0.3390 | 1.0509 |
| | Re$^2$IC | 0.1500 | 28.53 | 0.9422 | 0.0754 | **0.0381** | **11.82** | **1.41** | 4.4016 | 57.24 | **0.5509** | 0.3429 | **0.5498** |
| High | VTM | 0.3880 | 36.14 | 0.9865 | 0.1571 | 0.1224 | 53.41 | 32.07 | 4.5570 | 61.57 | 0.5335 | 0.3815 | 0.7759 |
| | MLIC$^+$ | 0.3020 | **35.44** | **0.9843** | 0.1708 | 0.1417 | 64.27 | 32.40 | 4.7365 | **61.97** | 0.5263 | **0.3748** | 0.9233 |
| | C3/WDs | 0.2950 | 29.59 | 0.9572 | 0.0498 | 0.0223 | 7.90 | 1.23 | 4.5353 | 58.59 | 0.5432 | 0.3440 | 0.3086 |
| | HiFiC | 0.2720 | 31.84 | 0.9774 | **0.0380** | 0.0492 | 12.74 | 1.40 | **3.4455** | 60.23 | 0.5142 | 0.3406 | 0.6168 |
| | Re$^2$IC | 0.2970 | 30.41 | 0.9636 | 0.0483 | **0.0187** | **7.10** | **0.80** | 4.1808 | 58.89 | **0.5492** | 0.3486 | **0.2411** |

(Tables 6, 7, and 13). For quantitative evaluation, we use PSNR for fidelity; MS-SSIM (Wang et al., 2003), LPIPS (Zhang et al., 2018), DISTS (Ding et al., 2022), FID (Heusel et al., 2017), and KID (Bińkowski et al., 2018) for reference-based perception; CLIP-IQA (Wang et al., 2023), MUSIQ (Ke et al., 2021), and MANIQA (Tu et al., 2023) as learned no-reference preference metrics; and NIQE (Mittal et al., 2013) as a handcrafted naturalness prior. We further measure MACs/pixel and latency for decoding efficiency. For user studies, we test all three target rates for baselines and the two lower rates for HiFiC, giving 14 method–rate combinations per dataset.

## 4.2 RATE-PERCEPTION PERFORMANCE

**User study.** Our evaluation protocol follows the CLIC framework with its open-source rating model. User is required to select one between two reconstructions closer to the original, and Elo scores for each method and rate are computed by minimizing the cross-entropy between observed and predicted preferences (more details provided in Appendix B.1). The main results on Kodak are shown in Fig. 5. While C3/WDs outperforms MLIC$^{++}$ and VVC, it lags behind generative-based HiFiC. Re$^2$IC closes this gap, surpassing HiFiC while operating at two orders of magnitude lower decoding complexity. Since HiFiC's open-sourced rates are higher than those of other baselines, we further conducted direct pairwise comparisons between HiFiC and Re$^2$IC at its released rates (see Fig. 10), further providing a clear performance gain for Re$^2$IC.[2] This highlights the promise of Re$^2$IC: by modeling perception region-by-region rather than learning a distribution, it can achieve superior RP performance at drastically reduced complexity.

**Quantitative evaluations.** We conduct a comprehensive quantitative assessment of Re$^2$IC, with results reported in Table 1 for CLIC2020 and Table 2 for Kodak. Across all bitrates in Table 1, Re$^2$IC consistently leads in perceptual metrics (such as LPIPS, DISTS, FID, KID, NIQE, and CLIP-IQA), and requires significantly less decoding complexity. Similar advantages can also be observed in Table 2. Interestingly, overfitted codecs often show greater performance gains at higher bi-trates (Kim et al., 2024; Wu et al., 2025), but for perceptual quality, the advantage is more pro-nounced at low bitrates. Notably, as shown in Table 1, Re$^2$IC not only surpasses C3/WDs in perceptual quality but also improves distortion, with up to a 0.82 dB PSNR gain, underscoring its leading role among overfitted codecs. Moreover, Re$^2$IC achieves better WD than C3/WDs de-spite not being directly trained on it. This highlights the effectiveness of Re$^2$IC's design, making WD optimization more effective and enhancing overall performance. Considering that WA-WD is tunable via learnable subband weights, we also report tuned Re$^2$IC results in Table 2, demon-

---

[2]More detailed comparisons on CLIC2020 and Kodak are provided in the supplementary materials, with interactive one-by-one flipping for easier comparisons.

Table 2: Quantitative comparison of different methods on Kodak. We report scores on BPP↓, PSNR↑, MS-SSIM↑, LPIPS↓, DISTS↓, FID↓, NIQE↓, MUSIQ↑, ClipIQA↑, and MANIQA↑. **Red** and blue indicate the best and second-best results, respectively. HiFiC is excluded from the high-BPP regime comparisons since its rate is significantly higher. Results for the Re$^2$IC counterparts with sub-band tuning are also included, demonstrating adaptive trade-offs for additional RP gains.

| | Model | BPP↓ | PSNR↑ | MS-SSIM↑ | LPIPS↓ | DISTS↓ | FID↓ | NIQE↓ | MUSIQ↑ | ClipIQA↑ | MANIQA↑ |
|---|---|---|---|---|---|---|---|---|---|---|---|
| Low | VTM | 0.113 | 28.51 | 0.9279 | 0.3246 | 0.2118 | 186.59 | 5.7412 | 66.74 | 0.3595 | 0.3564 |
| | MLIC$^{++}$ | 0.108 | **29.16** | **0.9339** | 0.3043 | 0.2076 | 175.71 | 5.9582 | **71.21** | 0.4803 | **0.4289** |
| | C3/WDs | 0.113 | 24.41 | 0.8958 | 0.1717 | 0.1179 | 72.83 | 5.6089 | 64.98 | 0.4803 | 0.3538 |
| | HiFiC | - | - | - | - | - | - | - | - | - | - |
| | Re$^2$IC | 0.113 | 24.67 | 0.8802 | **0.1458** | **0.0975** | **63.89** | **3.9574** | 68.37 | **0.5255** | 0.3932 |
| | ↪ Tuning | 0.113 | 23.76 | 0.8408 | 0.1432 | 0.0860 | 58.21 | 3.9574 | 69.96 | 0.5797 | 0.4125 |
| Medium | VTM | 0.182 | 30.10 | 0.9520 | 0.2538 | 0.1767 | 147.87 | 5.1808 | 70.68 | 0.4827 | 0.4084 |
| | MLIC$^{++}$ | 0.175 | **30.72** | **0.9555** | 0.2358 | 0.1749 | 141.69 | 5.3280 | 73.33 | 0.5552 | 0.4495 |
| | C3/WDs | 0.183 | 26.23 | 0.9324 | 0.1150 | 0.0786 | 49.87 | 5.5448 | 68.49 | 0.5062 | 0.3899 |
| | HiFiC | 0.183 | 27.56 | 0.9456 | **0.0665** | 0.0889 | 54.89 | **2.7894** | **73.76** | **0.6562** | **0.4576** |
| | Re$^2$IC | 0.183 | 26.13 | 0.9168 | 0.1022 | **0.0616** | 40.01 | 4.0660 | 70.62 | 0.5988 | 0.4209 |
| | ↪ Tuning | 0.183 | 25.65 | 0.9006 | 0.1017 | 0.0572 | 37.05 | 4.0302 | 71.86 | 0.6149 | 0.4388 |
| High | VTM | 0.287 | 31.84 | 0.9694 | 0.1864 | 0.1416 | 106.87 | 4.6125 | 73.13 | 0.5623 | 0.4586 |
| | MLIC$^{++}$ | 0.272 | **32.33** | **0.9719** | 0.1720 | 0.1425 | 111.99 | 4.6692 | **74.72** | 0.6027 | **0.4692** |
| | C3/WDs | 0.275 | 27.79 | 0.9529 | 0.0802 | 0.0537 | 35.39 | 4.8479 | 70.15 | 0.5491 | 0.4085 |
| | HiFiC | 0.351 | 29.65 | 0.9707 | 0.0428 | 0.0639 | 34.37 | 2.9543 | 74.19 | **0.6721** | 0.4588 |
| | Re$^2$IC | 0.274 | 27.27 | 0.9406 | **0.0752** | **0.0411** | 28.23 | 4.2442 | 71.98 | 0.6264 | 0.4411 |
| | ↪ Tuning | 0.275 | 26.74 | 0.9231 | 0.0764 | 0.0374 | 26.09 | 3.8981 | 72.27 | 0.6446 | 0.4446 |

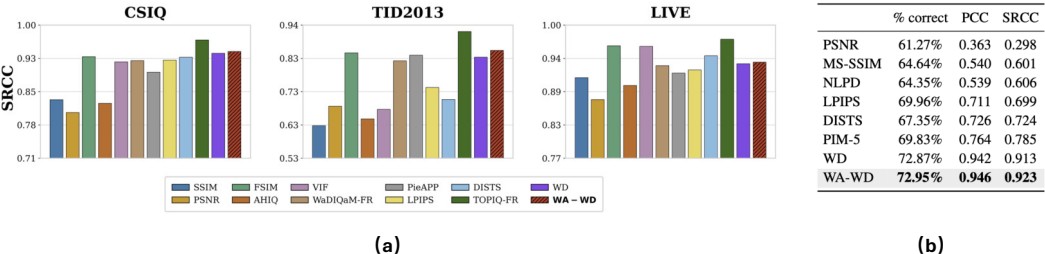

| | % correct | PCC | SRCC |
|---|---|---|---|
| PSNR | 61.27% | 0.363 | 0.298 |
| MS-SSIM | 64.64% | 0.540 | 0.601 |
| NLPD | 64.35% | 0.539 | 0.606 |
| LPIPS | 69.96% | 0.711 | 0.699 |
| DISTS | 67.35% | 0.726 | 0.724 |
| PIM-5 | 69.83% | 0.764 | 0.785 |
| WD | 72.87% | 0.942 | 0.913 |
| WA-WD | **72.95%** | **0.946** | **0.923** |

(a)         (b)

Figure 7: Evaluation of WA-WD: (a) performance across IQA datasets; (b) human-rating prediction.

strating its potential for improved RP performance and flexible R–D–P trade-offs. More quantitative results validating the advantages of Re$^2$IC are also provided in Tables 6,7 and Figs. 13,14.

A breakdown of bit cost in Fig. 6 shows that, Re$^2$IC allocates bits more evenly across latent vectors, capturing both low-frequency structures (typically in low-resolution latents) and high-frequency details (in high-resolution latents). This balanced allocation encourages richer cross-frequency interactions, consistent with our Wavelet-WD design intuition, leading to more efficient bit usage and a better PD trade-off. Latent visualizations in Fig. 20 further confirm that Re$^2$IC enhances these interactions.

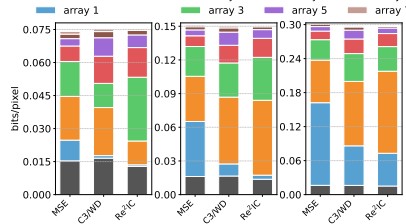

Figure 6: Bit allocation across latents from array 1-7 (highest to lowest resolution) at 0.075, 0.15, and 0.3 bpps.

### 4.3 PERCEPTION METRIC

To validate WA-WD as an effective perceptual indicator for compression, we directly use it to predict user preferences in the open-source human rating study (Ballé et al., 2025), where prediction quality was measured using Pearson (PCC) and Spearman (SRCC) correlations. As shown in Fig. 7 (b), WA-WD achieved the highest alignment with Elo scores. We further evaluate WA-WD on multiple image quality assessment (IQA) datasets (Fig. 7

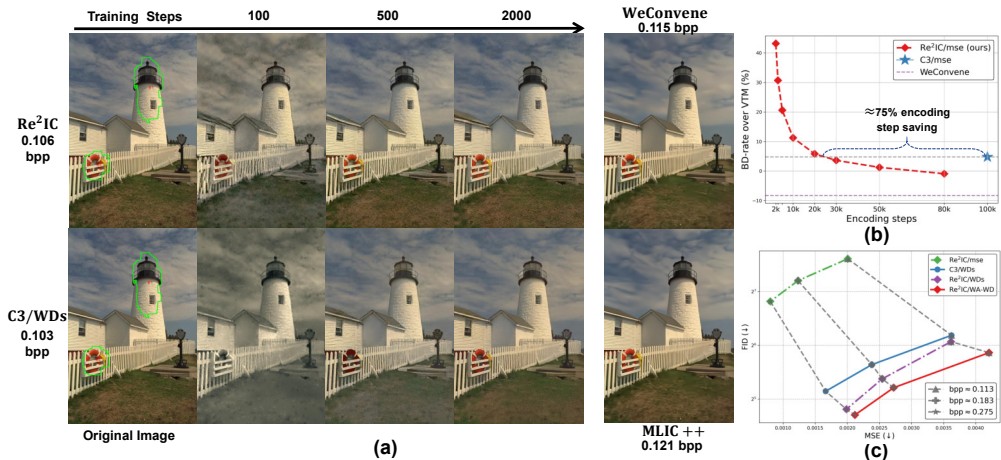

Figure 8: Encoding convergence: (a) RP and (b) RD; (c) P–D trade-offs on Kodak.

(a)), where it consistently outperforms WD and ranks among the top perceptual metrics. Notably, these results rely on a plain VGG backbone; even larger gains are expected with WA-WD tuning or more advanced backbones.

### 4.4 ABLATION STUDIES

**Convergence speed.**    Another advantage of $Re^2IC$ is its fast convergence. We evaluate the convergence speed of both RD and RP ($\sigma = 0/8$) on Kodak. For RD (Fig. 8 (b)), $Re^2IC$ reaches the BD-rate of C3 (Kim et al., 2024) (100k encoding steps) with only 25k steps, highlighting the effectiveness of partition strategy. For RP (Fig. 8 (a)), $Re^2IC$ delivers satisfactory perceptual quality within 2k steps, far fewer than C3/WDs (Ballé et al., 2025) (with more examples in Fig. 22).

**PD trade-off.**    We compare $Re^2IC$ with variants trained using MSE and WD. As shown in Fig. 8 (c), optimizing for MSE alone significantly reduces perceptual quality. $Re^2IC/WDs$ outperforms C3/WDs, validating the benefit of region-wise coding, while adding wavelet control provides further perceptual gains. This progression, from fidelity-only to full $Re^2IC$ paradigm, highlights the cumulative effect of each module for improved perceptual performance and aligns with the R-D-P trade-off. More detailed ablation studies for each component are provided in Tables 10 and 11.

**Contour accuracy.**    We examine the effect of artifacts from lossy contour compression. As shown in Fig. 21, for $Re^2IC/mse$ ($\sigma = 0$) at low bitrates, artifacts may arise at LPN boundaries or where distributions change abruptly. In contrast, $Re^2IC/WA-WD$ avoids such issues via texture resampling. This demonstrates that WA-WD, by modeling visual inhomogeneity and integrating low- and high-frequency interactions, improves visual robustness at region boundaries.

## 5 CONCLUSION

We introduced $Re^2IC$, a realism-enhanced region-based implicit codec that achieves perceptual image compression at low complexity. By combining saliency-driven region partitioning with lightweight local-global modulation, $Re^2IC$ delivers efficient region-wise perception modeling with reduced decoding cost. Furthermore, the proposed wavelet–Wasserstein distortion enables explicit control of fidelity and realism across frequency subbands, supporting fine-grained optimization and flexible bit allocation. Extensive experiments and user studies demonstrate that $Re^2IC$ surpasses state-of-the-art overfitted codecs and even generative methods like HiFiC, while using less than 1% of their decoding cost. Beyond compression, the proposed wavelet–WD also serves as a standalone, tunable metric that better aligns with human perception. These results highlight the potential of region- and frequency-aware modeling to break conventional trade-offs, opening new directions for low-cost and perceptually driven compression.

## ETHICS STATEMENT

This work involves a human subjective evaluation study (see our Fig. 5) to assess perceptual quality. Participants were recruited voluntarily and provided informed consent before taking part. No personally identifiable information was collected, and all data were anonymized and used solely for research purposes. The study design followed standard ethical practices for user studies, with minimal risk to participants.

Beyond the human evaluation, our work relies only on publicly available standard datasets ( Kodak, CLIC2020,) that do not contain sensitive or private information. We foresee potential positive societal impacts through improved efficiency in visual communication systems. As with other compression and generative frameworks, there is a possibility of misuse for creating manipulated content, which we caution against. We encourage responsible and fair use of our methods.

## REPRODUCIBILITY STATEMENT

We have taken care to ensure the reproducibility of our results. Detailed training and evaluation settings are provided in the main paper and the appendix, including network architectures, loss functions, datasets, hyperparameters, and optimization schedules. To further facilitate reproducibility, we will release our source code, checkpoints, and scripts for evaluation and visualization upon publication. Most of them are shown in our supplementary materials.

## USE OF LARGE LANGUAGE MODELS

Large Language Models (LLMs) were employed solely for non-scientific writing assistance, such as polishing the text and checking grammar or clarity. No part of the conceptual design, experimental methodology, analysis, or results was generated by an LLM. All scientific contributions are the original work of the authors.

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

# A APPENDIX

## APPENDIX CONTENTS

# A OPEN RESOURCES

**(a) Adaptive chain coding method.** We open-source an adaptive chain-coding method (Chang et al., 1992), additionally including a move-to-front (MTF) transform to further improve the compression efficiency, which can support both lossy and lossless coding for contour.

**(b) Code and checkpoints.** Code and checkpoints will be released after the review process. Preliminary versions (under file size limitations) are included in the supplementary materials.

**(c) Detailed evaluation results at various rates.** We provide reconstructions with their rates on Kodak and CLIC2020 to facilitate future research and comparisons.

*Given the subjectivity of human ratings, we strongly encourage reviewers to open the supplementary materials, which provide detailed one-by-one perceptual comparisons. (The example of interface is shown in Fig. 9. All results will be released on a project page after the review process.)*

# B BASELINE AND EXPERIMENTAL SETTING

## B.1 EXPERIMENT SETUP

We evaluate $Re^2IC$ on the Kodak dataset (24 images)(Kodak, 1991) and the CLIC2020 professional validation set (41 images)(Toderici et al., 2020), using both human rater studies and quantitative assessments. Baselines include classical codecs (VTM (Bross et al., 2021)), currently state-of-the-art autoencoder codecs (MLIC$^+$, MLIC$^{++}$(Jiang & Wang)), the generative codec HiFiC (Mentzer et al., 2020), and perception-enhanced overfitted codecs (C3/WDs (Ballé et al., 2025)), and fidelity-oriented overfitted codecs (C3/MSE (Kim et al., 2024)). Additionally, diffusion-based perception codecs, such as TACO (Lee et al., 2024) and CDC (Yang & Mandt, 2023), and DWT-based codec WeConvene (Fu et al., 2024) are also introduced in Tables 6, 7, and 13.

For each method, we evaluate compressed reconstructions at three dataset-average bitrates (e.g., low, medium, and high bpp regimes). On CLIC2020, we target $\{0.075, 0.15, 0.3\}$ bpp (as in the CLIC competition) and all results come from the open-sourced reconstructions from (Ballé et al., 2025). For Kodak, we use $\{0.113, 0.183, 0.275\}$ bpp to align with open-sourced baselines.[3] Note that MLIC$^{++}$ is used for Kodak to match the target rate regime, while MLIC$^+$ is used for CLIC2020 to align with prior work (Ballé et al., 2025).

For quantitative evaluation, we use PSNR for fidelity; MS-SSIM, LPIPS, DISTS, FID (patch 256), and KID (patch 128, multiplied by $10^3$) for reference-based perceptual similarity; CLIP-IQA, MUSIQ, and MANIQA as learned no-reference preference metrics; and NIQE as a handcrafted naturalness prior.[4] We also measure MACs per pixel and coding latency to assess decoding complexity and efficiency.

---

[3]All results are adopted from their officially released reconstructions or codes.

[4]All above experiment-related validations, including metric computation and image reconstructions, will be open-sourced to facilitate reproducibility.

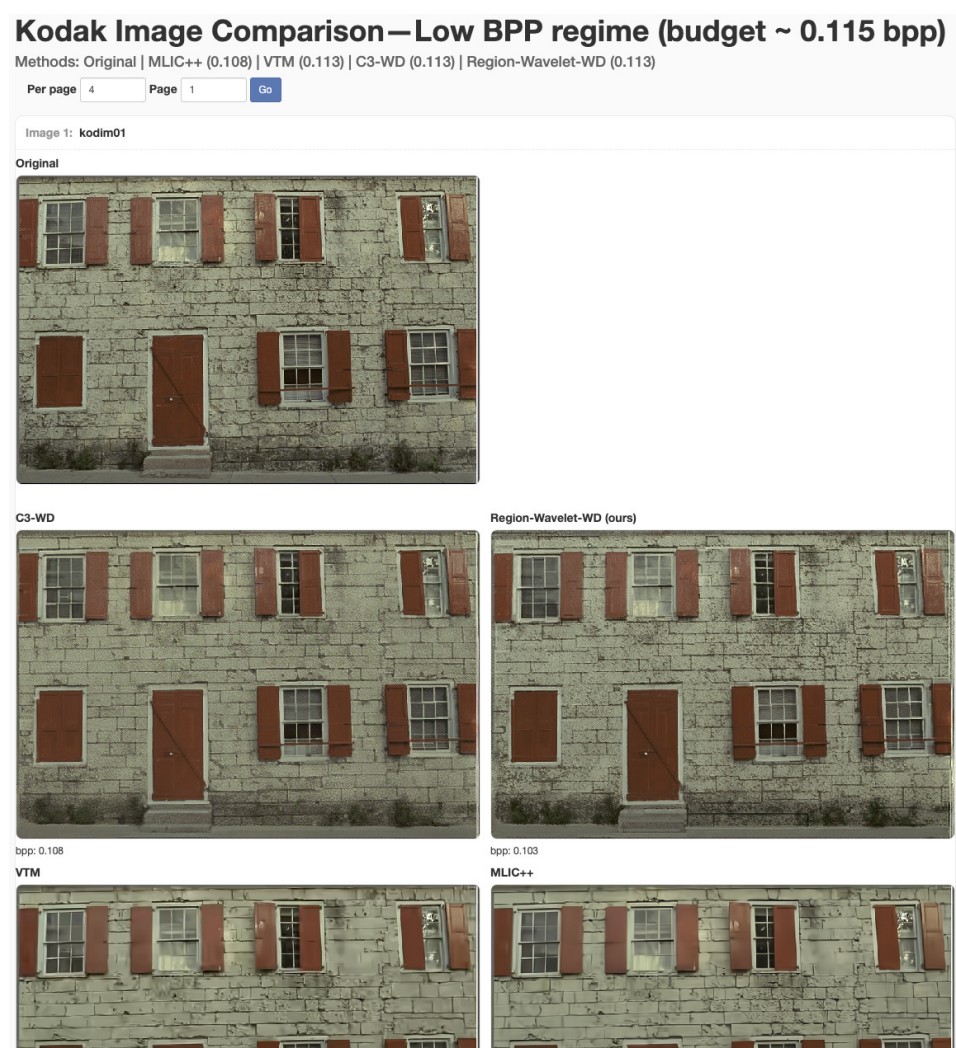

Figure 9: Example of our interactive comparison interface (included in the supplementary materials), which supports zooming and flip-toggle comparisons across methods, enabling clear visualization of the perceptual advantages of our approach.

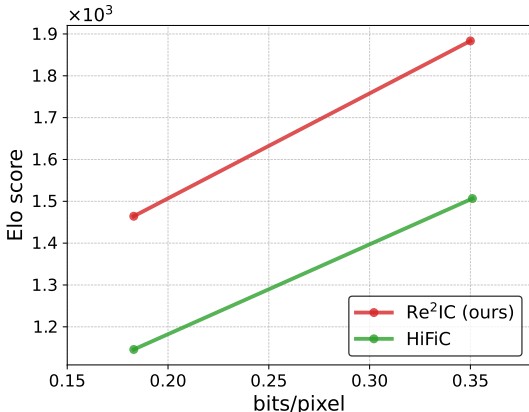

Figure 10: Human rating results for Re$^2$IC and HiFiC across more rates.

For the user study, our evaluation protocol follows the CLIC framework and uses their open-source rating model. In each trial, raters are shown a random $512 \times 512$ crop from Kodak, with the original on one side and two reconstructions on the other. By flipping between reconstructions, they select the one closer to the original, and Elo scores for each method and rate are computed by minimizing the cross-entropy between observed and predicted preferences. We included all three target rates for the baselines and the two lower rates for HiFiC, yielding 14 method–rate combinations per dataset and thus 14 reconstructions per image for both Kodak. In total, 10 users participate in our rating. For the evaluation of the WA-WD metric, we directly employ open-resourced records in (Ballé et al., 2025), as they have more comprehensive assessments for a better verification of our method.

For IQA experiments, we consider CSIQ (Larson & Chandler, 2010), TID2013 (Ponomarenko et al., 2015), LIVE (Sheikh, 2005) datasets, and report SRCC.

## C  RE$^2$IC IMPLEMENTATION

### C.1  REGION PARTITION AND CONTOUR COMPRESSION

We partition images into regions using saliency predictions. An EML-Net (Jia & Bruce, 2020) is employed to generate a saliency map, from which we threshold and extract $N - 1$ connected components as sub-regions for LPN allocation. While high-resolution images could benefit from more sub-networks for finer perception modeling, we find that one or two are typically sufficient for a competitive RP trade-off. A detailed illustration of the region partitioning and contour coding strategy is provided below (also see our supplementary materials).

```
Region partition and contour coding

## Region partition:
regions = extract_top_saliency_regions(
          saliency_map,N=2,dilate_kernel_size=3,
          threshold=np.percentile(saliency_map, 90),
          min_area_ratio=0.005,
       )
## Chain coding for contour:
total_bits = contour_compression(input_contour, step_length,
                                 step_angle)
```

The process first thresholds the saliency map to generate a binary mask, followed by dilation and morphological opening to remove noise and merge nearby salient regions. External contours are extracted as candidate regions. Each region's perceptual score, given by mean saliency $\times$ area, is used to rank and select the top-$N$ regions, discarding those below a minimum area ratio. For contour compression, a move-to-front (MTF) transform (Elias, 1987; Bentley et al., 1986) is applied to further reduce redundancy.

### C.2  QUANTIZATION AND ENTROPY CODING DETAILS

In Re$^2$IC, the latent modulations $z^i$, LPN parameters $W^i$, and those of ARM and GPM are quantized and entropy-coded using the same methods, as in (Kim et al., 2024; Ladune et al., 2024; Wu et al., 2025). Specifically, during training, latent modulation $z^i$ is soft-rounded with Kumaraswamy noise (Kim et al., 2024), while hard rounding is applied at inference. Rate terms for $\{\tau^i, \theta, W^i, \psi\}$ are omitted from the loss due to their negligible contributions and simplified optimization. In practice, network parameters are quantized and entropy-coded with a non-learned distribution. All components are included in the final bitstream.

**Quantization-aware optimization.**  A two-stage quantization-aware scheme is employed: each $z^i$ is optimized in a continuous space during the training, with quantization approximated via soft-rounding and Kumaraswamy noise for differentiability:

$$\hat{z}^i = \begin{cases} \mathcal{S}_T(z^i) + u_{kum}, & \text{Stage I (80,000 steps)} \\ Q(z^i), & \text{Stage II (8000 steps),} \end{cases} \tag{2}$$

where $\mathcal{S}_T$ is the temperature for soft rounding, $Q$ is hard rounding (same with the inference phase), and $\boldsymbol{u}_{kum}$ is Kumaraswamy noise. Noise and temperature are gradually annealed for stable training.

**Entropy coding.** For entropy coding, we adopt a factorized auto-regressive model (ARM) $r_{\boldsymbol{\psi}}$ (Ballé et al., 2018; Ladune et al., 2023). Each latent element $\hat{z}^i_{k,j}$ ($i$-th region, $k, j$ position) is conditioned on $C$ neighbors $\boldsymbol{c^i_{k,j}} \in \mathbb{Z}^C$:

$$p_{\boldsymbol{\psi}}(\hat{\boldsymbol{z}^i}) = \prod_{k,j} p_{\boldsymbol{\psi}}(\hat{z}^i_{k,j}|\boldsymbol{c^i_{k,j}}), \tag{3}$$

where $p_{\boldsymbol{\psi}}(\hat{\boldsymbol{z}^i})$ is modeled as an integrated Laplace distribution $p_{\boldsymbol{\psi}}(\hat{z}^i_{k,j}|\boldsymbol{c^i_{k,j}}) = \int_{\hat{z}^i_{k,j}-0.5}^{\hat{z}^i_{k,j}+0.5} g(z)dz$, and $g \sim \mathcal{L}(\mu^i_{k,j}, \sigma^i_{k,j})$, $\mu^i_{k,j}, \sigma^i_{k,j} = r_{\boldsymbol{\psi}}(\boldsymbol{c^i_{k,j}})$. With range coding [5], the rate for $\hat{\boldsymbol{z}^i}$ becomes:

$$R(\hat{\boldsymbol{z}^i}) = -\log_2 p_{\boldsymbol{\psi}}(\hat{\boldsymbol{z}^i}) = \sum_{k,j} -\log_2 p_{\boldsymbol{\psi}}(\hat{z}^i_{k,j}|\boldsymbol{c^i_{k,j}}). \tag{4}$$

The final bit cost for all latent vectors are $\boldsymbol{b_z} = \sum_{i=0}^{N} R(\hat{\boldsymbol{z}^i})$.

---

**Algorithm 1** Encoding stage of the Re²IC

---

**Input:** Source image $\boldsymbol{S}$, coordinate $\boldsymbol{x^i}$, Learning rates for Stage I and II: $\alpha, \beta$,
Latent and random vectors: $(\boldsymbol{z^i}, \boldsymbol{M^i})$, and networks $(\boldsymbol{W^i}, \boldsymbol{\theta^i},$ and $\boldsymbol{\psi})$,
**Output:** Bits stream of $\boldsymbol{b_z}, \boldsymbol{b_\tau}, \boldsymbol{b_W}, \boldsymbol{b_\theta}, \boldsymbol{b_\psi}$.

1:   $\boldsymbol{S_a} = \text{EML}_{net}(\boldsymbol{S}), \boldsymbol{S_a} \to \{\boldsymbol{S^1}, \ldots, \boldsymbol{S^N}\}; \{\boldsymbol{\tau^1}, \ldots, \boldsymbol{\tau^N}\}; \{\boldsymbol{x^1}, \ldots, \boldsymbol{x^N}\},$      # Partition the source

2:   $\boldsymbol{z^i} \triangleq \{\boldsymbol{z^i_1}, \ldots, \boldsymbol{z^i_L}\}, \boldsymbol{M^i} \triangleq \{\boldsymbol{m^i_1}, \ldots, \boldsymbol{m^i_L}\}$ with each $\boldsymbol{z^i_k}, \boldsymbol{m^i_k} \in \mathbb{R}^{\frac{K^i}{4^{k-1}}}$      # Initialize latent vectors

3:   **for** the $k$-step within the Stage I **do**

4:     **for** each $i$-th region within the $N$ regions **do**

5:       $\hat{\boldsymbol{z}}_i = \mathcal{S}_T(\boldsymbol{z}_i) + \boldsymbol{u}_{kum}, \hat{\boldsymbol{S}}_i = f_{\boldsymbol{W^i}}(f_{\boldsymbol{\theta}}(\hat{\boldsymbol{z}^i}, \boldsymbol{M^i}), \boldsymbol{x^i})$      # Quantization-aware training

6:     **end for**

7:     $\hat{\boldsymbol{S}} = \bigcup_{i=1}^N \hat{\boldsymbol{S}^i},$      # Merge regions

8:     $\mathcal{L}(\boldsymbol{\Omega}) = WD_{\text{wave}}(\boldsymbol{S}, \hat{\boldsymbol{S}}) + \sum_{i=0}^N \lambda R(\hat{\boldsymbol{z}^i}),$      # Compute the PD loss function

9:     $\boldsymbol{\theta} \leftarrow \boldsymbol{\theta} - \alpha \nabla_{\boldsymbol{\theta}} \mathcal{L},$      # Update the GPM for all regions

10:    $\boldsymbol{z^i} \leftarrow \boldsymbol{z^i} - \alpha \nabla_{\boldsymbol{z^i}} \mathcal{L},$      # Update the latent for each region

11:    $\boldsymbol{W^i} \leftarrow \boldsymbol{W^i} - \alpha \nabla_{\boldsymbol{W^i}} \mathcal{L},$      # Update the LPN for each region

12:    $\boldsymbol{\psi} \leftarrow \boldsymbol{\psi} - \alpha \nabla_{\boldsymbol{\psi}} \mathcal{L},$      # Update ARM for all regions

13: **end for**

14: **for** the $k$-th step within the Stage II **do**

15:     **for** each $i$-th region within the $N$ regions **do**

16:       $\hat{\boldsymbol{z}^i} = Q(\boldsymbol{z^i}), \hat{\boldsymbol{S}}_i = f_{\boldsymbol{W^i}}(f_{\boldsymbol{\theta}}(\hat{\boldsymbol{z}^i}, \boldsymbol{M^i}), \boldsymbol{x^i}),$      # Hard rounding

17:     **end for**

18:     $\hat{\boldsymbol{S}} = \bigcup_{i=1}^N \hat{\boldsymbol{S}^i},$      # Merge regions

19:     $\mathcal{L}(\boldsymbol{\Omega}) = WD_{\text{wave}}(\boldsymbol{S}, \hat{\boldsymbol{S}}) + \sum_{i=0}^N \lambda R(\hat{\boldsymbol{z}^i}),$      # Compute the PD loss function

20:    $\boldsymbol{\theta} \leftarrow \boldsymbol{\theta} - \beta \nabla_{\boldsymbol{\theta}} \mathcal{L},$      # Update the GPM for all regions

21:    $\boldsymbol{z^i} \leftarrow \boldsymbol{z^i} - \beta \nabla_{\boldsymbol{z^i}} \mathcal{L},$      # Update the latent for each region

22:    $\boldsymbol{W^i} \leftarrow \boldsymbol{W^i} - \beta \nabla_{\boldsymbol{W^i}} \mathcal{L},$      # Update the LPN for each region

23:    $\boldsymbol{\psi} \leftarrow \boldsymbol{\psi} - \beta \nabla_{\boldsymbol{\psi}} \mathcal{L},$      # Update ARM for all regions

24: **end for**

25: $\boldsymbol{b_z} = \sum_{i=0}^N \mathcal{A}(Q(\boldsymbol{z^i})),$      # Quantization and entropy coding over $\boldsymbol{z^i}$

26: $\boldsymbol{b_\tau} = \sum_i^N \mathcal{C}(\boldsymbol{\tau^i}),$      # Lossy chain coding for contours

27: $\Delta_{\boldsymbol{\theta}}, \Delta_{\boldsymbol{W^i}}, \Delta_{\boldsymbol{\psi}} = \mathcal{G}(\boldsymbol{\theta}, \boldsymbol{W^i}, \boldsymbol{\psi})$      # Search for optimal quantization steps for networks

28: $\boldsymbol{b_\theta} = \mathcal{A}(Q(\boldsymbol{\theta}, \Delta_{\boldsymbol{\theta}})), \boldsymbol{b_\psi} = \mathcal{A}(Q(\boldsymbol{\psi}, \Delta_{\boldsymbol{\psi}})) \boldsymbol{b_W} = \sum_{i=0}^N \mathcal{A}(Q(\boldsymbol{W^i}, \Delta_{\boldsymbol{W^i}})),$
     # Quantization and entropy coding over all network parameters

---

**Model compression.** Parameters of all networks are quantized with scalar quantizers $Q(\cdot, \Delta)$ and entropy-coded using Laplace priors ($\boldsymbol{W^i}$ with a step size $\Delta$ as an example): $\hat{\boldsymbol{W}^i} = Q(\boldsymbol{W^i}, \Delta_{\boldsymbol{W^i}})$,

---

[5] We employ the range-coder in PyPI for range-coding. Note that here fixed-point fractional-multiplier quantization is necessary to ensure bit-exact consistency for reliable range decoding.

---

**Algorithm 2** Decoding stage of the Re$^2$IC

**Input:** Bits stream of $\boldsymbol{b_z}, \boldsymbol{b_\tau}, \boldsymbol{b_W}, \boldsymbol{b_\theta}, \boldsymbol{b_\psi}$.

**Output:** Reconstruction of image $\hat{\boldsymbol{S}}$.

1: $\hat{\boldsymbol{\tau}}^i, \boldsymbol{x}^i \leftarrow \mathcal{C}(\boldsymbol{b_\tau})$ ,                 Decode contour to identify region partition

2: $\hat{\boldsymbol{\psi}} = Q(\mathcal{A}(\boldsymbol{b_\psi})), \hat{\boldsymbol{\theta}} = Q(\mathcal{A}(\boldsymbol{b_\theta})), \hat{\boldsymbol{W}}^i = Q(\mathcal{A}(\boldsymbol{b_{W^i}}))$ ,            Decode networks

3: $\hat{\boldsymbol{z}}^i = Q(\mathcal{A}_{\hat{\tau}}(\boldsymbol{b_z})), \boldsymbol{M}^i \sim \mathcal{N}(0,1)$ ,     Decode the latent vectors and synchronize a random commonness

4: **for** each $i$ region within $N$ regions **do**

5:      $\hat{\boldsymbol{S}}_i = f_{\hat{\boldsymbol{W}}^i}(f_{\hat{\boldsymbol{\theta}}}(\hat{\boldsymbol{z}}^i, \boldsymbol{M}^i), \boldsymbol{x}^i)$,                   # Reconstruct each local region

6: **end for**

7: $\hat{\boldsymbol{S}} = \bigcup_{i=1}^{N} \hat{\boldsymbol{S}}^i$,                               Merge the source image

---

and $p(\hat{W}_k) = \int_{\hat{W}_k - 0.5}^{\hat{W}_k + 0.5} g(W) dW$,    with    $g \sim \mathcal{L}(0, \text{stddev}(\hat{\boldsymbol{W}}))$. The total parameter rate is:

$$R_{\text{MLP}} = -\sum_{k,i} \log_2 p(\hat{W}_k^i) - \sum_k \log_2 p(\hat{\theta}_k) - \sum_k \log_2 p(\hat{\psi}_k). \tag{5}$$

Quantization steps are searched via greedy PD optimization:

$$\min_{\Delta \hat{W}^i, \Delta \hat{\psi}, \Delta \hat{\theta}} d\left(\boldsymbol{S}^i, \bigcup_{i=1}^N \left[ f_{\hat{\boldsymbol{W}}^i}(f_{\hat{\boldsymbol{\theta}}}(\hat{\boldsymbol{z}}^i), \boldsymbol{x}^i) \right]\right) + \lambda \sum_i^N R(\hat{\boldsymbol{z}}^i) + \lambda R_{MLP}. \tag{6}$$

**Model architecture.** For the ARM model with 24 contextual inputs, we use three linear (or $1 \times 1$ convolutional) layers with GELU activations, mapping $(24 \times 24) \rightarrow (24 \times 24) \rightarrow (24 \times 2)$.

The GPM model takes $L = 7$ resolution-layered latent vector as input with 4 convolutional layers, mapping $(7 \times 18) \rightarrow (18 \times 3) \rightarrow (3 \times 3) \rightarrow (3 \times 3)$. Specifically, as shown in Fig. 4, GPM $f_{\boldsymbol{\theta}}$ consists of an upsampling module and 4 convolutions. The quantized $\hat{\boldsymbol{z}}^i$ is first upsampled by transposed convolutions to $\boldsymbol{U}_0^i$, then to generate hierarchical modulation vectors for each LPN: $\boldsymbol{U}_k^i = f_{\boldsymbol{\theta}}^{(k)}(\boldsymbol{U}_{k-1}^i)$, where $\boldsymbol{U}_{k-1}^i$ denotes the $k$-th layer output of $g_{\boldsymbol{\theta}}$ for $i$-the region. Inspired by (Mehta et al., 2021; Wu et al., 2025; Perez et al., 2018), GPM modulates each region perception in a concatenated fashion: the modulation vector $\boldsymbol{M}_k^i \triangleq \text{Concatenate}(\boldsymbol{U}_1^i, \ldots, \boldsymbol{U}_k^i)$ for the $k$-th layer of $f_{\boldsymbol{W}^i}$. At each layer of LPN, $\boldsymbol{M}_k^i$ is concatenated with the current output and processes it through a shared MLP, enabling consistent global context injection and dynamic refinement.

Each LPN maps pixel coordinates $\boldsymbol{x}^i$ to RGB with 4 layers, modulated by $\mathcal{M}_k^i$ using modulations from $f_{\boldsymbol{\theta}}$. The input and output dimensions are given as: $(2 \times 3) \rightarrow ([3 + 18 + 3] \times 3) \rightarrow (3 \times 3) \rightarrow ([3 + 18 + 3 + 3] \times 3) \rightarrow (3 \times 3) \rightarrow ([3 + 18 + 3 + 3 + 3] \times 3) \rightarrow (3 \times 3)$, where the orange MLP layers are globally shared across regions.

**Optimization details.** We introduce a 700-step warm-up stage to stabilize region-based coding, training 7 candidates with different random seeds and retaining the best for the remaining training. Adam optimizer with a learning rate of $1e - 4$ is employed.

For WA-WD optimization, we use a single-level DWT to reduce computation while maintaining satisfactory performance, though further gains are expected with deeper decompositions of DWT or by replacing standard downsampling operations.

**Model parameters.** Table 3 summarizes all parameter settings used in our experiments. We highlight that rate selection across methods, particularly under WD or WA-WD optimization, costs a lot of computational resources to ensure fair comparison. To support future research and reproducibility, we release the full $\lambda$ configuration used in our experiments.

**Pseudocode for the algorithm.** A detailed pseudocode of encoding and decoding process of Re$^2$I is presented in Algorithm 1 and Algorithm 2, respectively.

Table 3: Hyper-parameter settings, with all reconstruction results provided on our project page after review process (currently on supplementary materials).

| Main experiments | | |
|---|---|---|
| Values of $\lambda$ for Kodak | | $\{4, 8.9, 23\}$ |
| Values of $\lambda$ for CLIC2020 | | $\{1.1, 4.3, 15.5\}$ |
| **Abaltion experiments** | | |
| Values of $\lambda$ for Tuning WA-WD on Kodak | | $\{3, 7, 20.5\}$ |
| Values of $\lambda$ for Ablation study on Kodak w/o common randomness | | $\{4, 8.9, 23\}$ |
| Values of $\lambda$ for Ablation study on Kodak w/o LPN | | $\{6, 12, 26\}$ |
| Values of $\lambda$ for Ablation study on Kodak w/o GPM | | $\{5.1, 11.2, 27\}$ |
| Values of $\lambda$ for Ablation study on Kodak w/o partition | | $\{5.7, 11.5, 25\}$ |
| Values of $\lambda$ for Ablation study on Kodak w/o wavelet | | $\{4.5, 10, 25\}$ |
| Values of $\lambda$ for RP convergence on Kodak for C3/WDs and Re$^2$IC | | $\{27 \text{ vs } 28\}$ |
| Values of $\lambda$ for RD convergence on Kodak | | $\{8e^{-3}, 2e^{-3}, 1e^{-3}, 5e^{-4}, 2e^{-4}, 1e^{-4}\}$ |
| **Hyper parameter** | **Initial values** | **Final values** |
| **Quantization – Stage I** | | |
| Number of encoding steps | | $8 \times 10^4$ |
| Learning rate $\beta$ | $10^{-2}$ | 0 |
| Scheduler for learning rate | | Cosine scheduler |
| Temperature $T$ for soft rounding | 0.3 | 0.1 |
| Noise strength $\alpha$ for Kumaraswamy noise | 2.0 | 1.0 |
| Scheduler for Soft-rounding and Kumaraswamy noise | | Linear scheduler |
| **Quantization – Stage II** | | |
| Number encoding steps | | $8 \times 10^3$ |
| Learning rate | $10^{-4}$ | $10^{-8}$ |
| Decay learning rate if loss has not improved for this many steps | | 40 |
| Decay factor | | 0.8 |
| Temperature $T$ for soft rounding | | $10^{-4}$ |
| **Warm-up and WA-WD setting** | | |
| Round 1 candidates | | 7 |
| Round 1 steps | | 400 |
| Round 2 candidates | | 3 |
| Round 2 steps | | 400 |
| $\sigma_{max}$ for Kodak | | 8 |
| $\sigma_{max}$ for CLIC | | 16 |
| DWT level | | 1 |
| PMF levels | | 5 |
| VGG slices | | 5 |
| Salient network | | EML-Net |
| Salient region numbers | | $1 \sim 2$ |
| **Architecture** | | |
| **Entropy model** | | |
| Output channel numbers for ARM model | | $(24 \times 24) \to (24 \times 24) \to (24 \times 2)$ |
| Activation function | | GELU |
| Log-scale of Laplace is shifted before $\exp$ | | 4 |
| Scale parameter of Laplace is clipped to | | $[10^{-2}, 150]$ |
| **Latent modulations** | | **Values** |
| Number of latent vectors $L$ | | 7 |
| Initialization of $z$ | | 0 |
| **Global perceptual modulator** | | **Values** |
| Upsampling kernel | | $8 \times 8$ |
| Output channels of the $1 \times 1$ convolutions | | $(7 \times 18) \to (18 \times 3)$ |
| Output channels of the $3 \times 3$ convolutions | | $(3 \times 3) \to (3 \times 3)$ |
| Modulation vector dimension | | $\{3 \to 3\}$ |
| **Local perceptual network** | | **Values** |
| Output dimensions of each layer | | $\{2 \to 3 \to 3 \to 3\}$ |
| Out-channels of each shared perceptual modulator | | |
| $\{[3 + 18 + 3] \to 3\}; \{[3 + 18 + 3 + 3] \to 3\}; \{[3 + 18 + 3 + 3 + 3] \to 3\}$ | | |

# D WAVELET-WASSERSTEIN DISTORTION

## D.1 WASSERSTEIN DISTORTION

**Definition.** Similar to LPIPS (Zhang et al., 2018) and DISTS (Ding et al., 2022), WD (Qiu et al., 2024) is a perceptual distance metric that measures image similarity in a feature-embedding space. Specifically, WD incorporates models of foveal and peripheral vision by computing point-wise feature-space distances with a spatially varying $\sigma$-map, *typically derived from saliency map*, which determines how permissive the model is to *texture resampling*.

Let $\boldsymbol{x} = \{x_n\}_{n=-\infty}^{\infty}$ denote the source sequence with infinite length (with straightforward extension to images) and $\boldsymbol{z} = \{z_n\}_{n=-\infty}^{\infty}$ with $z_n = \phi(\boldsymbol{x}) \in \mathbb{R}^d$ denote its feature representation from a neural network $\phi$. WD is defined through a family of pooling probability mass functions (PMFs) $q_\sigma(k)$ (parameterized by a $\sigma$-map) over integers $k$, satisfying the *two-sided geometric distribution*[6]:

$$q_\sigma(k) = \begin{cases} \frac{e^{1/\sigma}-1}{e^{1/\sigma}+1} \cdot e^{-|k|/\sigma} & \text{if } \sigma > 0 \\ 1 & \text{if } \sigma = 0 \text{ and } k = 0 \\ 0 & \text{otherwise.} \end{cases} \tag{7}$$

This induces local empirical measures $\boldsymbol{y}_\sigma = \{y_{n,\sigma}\}_{n=-\infty}^{\infty}$, where $y_{n,\sigma} \triangleq \sum_{k=-\infty}^{\infty} q_\sigma(k)\delta_{z_{n+k}}$, and $\delta$ is Dirac delta measure. Each $y_{n,\sigma}$ represents the *pooled feature statistics* around location $n$ with an *effective pooling width* $\sigma \geq 0$.

For a reconstruction $\hat{\boldsymbol{x}} = \{\hat{x}_n\}_{n=-\infty}^{\infty}$ with features $\hat{\boldsymbol{z}} = \{\hat{z}_n\}_{n=-\infty}^{\infty}$ and pooled measures $\hat{\boldsymbol{y}}_\sigma = \{\hat{y}_{n,\sigma}\}_{n=-\infty}^{\infty}$, the local distortion is:

$$D_{n,\sigma} = W_p^p(y_{n,\sigma}, \hat{y}_{n,\sigma}), \tag{8}$$

where $W_p$ denotes the Wasserstein distance of order $p$ (Villani et al., 2008):

$$W_p(\rho, \hat{\rho}) = \inf_{Z \sim \rho, \hat{Z} \sim \hat{\rho}} \mathbb{E}\big[d^p(Z, \hat{Z})\big]^{1/p}, \tag{9}$$

where $\rho$ and $\hat{\rho}$ are probability measures on $\mathbb{R}^d$ and $d(\cdot)$ is a metric.

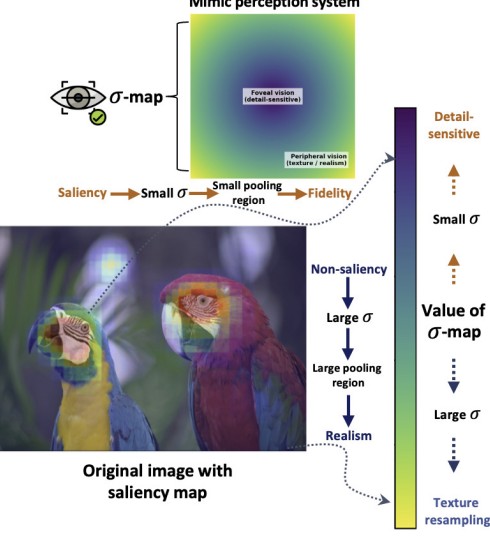

Figure 11: WD for foveal-peripheral vision.

The overall WD is the spatial average as:

$$WD(\boldsymbol{x}, \hat{\boldsymbol{x}}) = \frac{1}{2N+1} \sum_{n=-N}^{N} D_{n,\sigma(n)} = \frac{1}{2N+1} \sum_{n=-N}^{N} W_p^p\big(y_{n,\sigma(n)}, \hat{y}_{n,\sigma(n)}\big),$$

where $\sigma(n)$ is the spatially varying pooling width (i.e., the $\sigma$-map).

**Interpretation.** As illustrated in Fig. 11, saliency assigns spatially varying $\sigma$ values that modulate pooling statistics and balance the trade-off between realism and fidelity. When the $\sigma$-map approaches 0 in regions of high visual importance (e.g., foveal or salient areas), WD reduces towards a pointwise distance (can be equivalent to MSE), emphasizing fine detail. Conversely, larger $\sigma$ values mimic peripheral vision by expanding the pooling extent, thereby prioritizing realistic texture over fine-grained accuracy.

In summary, the $\sigma$-map determines the *pooling extent* used in the distortion computation and is explicitly shaped to reflect *foveal and peripheral vision*, thereby mimicking human perceptual behavior. In the context of *image compression*, $\sigma$-map controls how permissive WD is to *texture resampling*, **which in turn enables more flexible *bit allocation*.**

---

[6] The PMF is designed for an adjustable, visually compliant pooling kernel without spectral zeros. This also precisely motivates our WA-WD design, where wavelet subbands offer a *spectrally complete* and *human vision consistent* feature basis that **reinforces** these theoretical requirements.

**Theoretical properties.**    The key theoretical properties of WD can be summarized as:

- **Property 1. Fidelity–Realism Parameterization:** It unifies *fidelity* (small $\sigma$) and *realism* (large $\sigma$) within a *single*, continuously parameterized framework by allowing $\sigma$ to vary over the full real-valued range.

- **Property 2. Metric Consistency:** If the PMF $q_\sigma(\cdot)$ has *no spectral nulls* and the base distance $d$ *is a metric*, then $WD(\boldsymbol{z}, \hat{\boldsymbol{z}})^{1/p}$ itself satisfies positivity, symmetry, and the triangle inequality, thus *is a metric*. If $\phi$ is *invertible*, $WD(\boldsymbol{x}, \hat{\boldsymbol{x}})^{1/p}$ *is also a metric*.

- **Property 3. Saliency-Adaptive Bit Allocation:** When combined with *human-perception saliency*, WD enables *flexible bit allocation* that adapts to varying fidelity–realism requirements across regions in image compression [7], leading to improved RP performance.

**Practical implementation.**    For computational efficiency (Olkin & Pukelsheim, 1982), Eqn.9 can be replaced by a proxy (Dupont et al., 2022b; Heusel et al., 2017; Liu et al., 2021):

$$|\mu - \hat{\mu}|_2^2 + \mathrm{Tr}\left(C + \hat{C} - 2(\hat{C}^{1/2}C\hat{C}^{1/2})^{1/2}\right), \tag{10}$$

which is equivalent to a 2-Wasserstein distance when $p = 2$, $d$ is Euclidean distance, and the distributions of probability measures $\rho$ are Gaussian with means $\mu$ and covariances $C$. In papers (Qiu et al., 2024; Ballé et al., 2025), a simplified version of Eqn. 10 is employed as:

$$\sum_{i=1}^{d}(\mu_i - \hat{\mu}_i)^2 + \left(\sqrt{V_i} - \sqrt{\hat{V}_i}\right)^2, \tag{11}$$

where $\mu_i$ and $V_i$ are the mean and variance of the $i$-th component. [8]

Even under the independent Gaussian assumption, computing WD in image compression for learned codec optimization still remains computationally impractical, as it requires aggregating local statistics using a distinct pooling kernel at every spatial location. A practical solution is to approximate WD by discretizing $\sigma$ to powers of two, replacing the original PMF with Gaussian downsampling kernels, and leveraging the inherent downsampling in VGG, together serving as an implicit pooling (PMFs) mechanism (see (Ballé et al., 2025) for details).

While efficient and empirically effective, these assumptions can be unrealistic in practice: (1). Neural features are rarely independent; (2). Log-scale approximations for WD at different $\sigma$ values can exacerbate correlation effects; (3). The design of implicit PMFs may introduce spectral blind points, leading to metric degeneracy for *Property 2* of WD. These considerations motivate our WA-WD formulation, which strengthens the robustness of WD, with more details provided in Appendix D.2.

D.2    Towards Wavelet-WD

Although WD provides a principled foundation for perceptual compression, several practical challenges remain, including the independence assumption over features, the implicit PMF approximation, and unstable gradients across mixed-frequency content. Motivated by human visual perception and by the need for fine-grained frequency control in implicit codecs, we introduce WA-WD for Re²IC. An illustration of WA-WD is presented in Fig. 12.

**The remainder of this section outlines the motivation, interpretation, theoretical properties, validation experiments, and tunable strategy for this design.**

**Motivation**

- **Consistency with Human Visual System (HVS)** Visual neuroscience shows that early vision performs multi-scale, orientation-selective filtering (V1) followed by larger pooling (V2) (Freeman & Simoncelli, 2011; Balas et al., 2009). WD models V2 via its $\sigma$-map pooling, while wavelet subbands naturally mirror the V1 stage. Although implemented in

---

[7]This actually motivates us to incorporate region-wise coding scheme in the context of implicit codec.

[8]However, Eqn. 11 *assumes feature components are uncorrelated*, an assumption that rarely holds in practice. Our WA-WD addresses this by explicitly modeling intra-block correlations before WD computation.

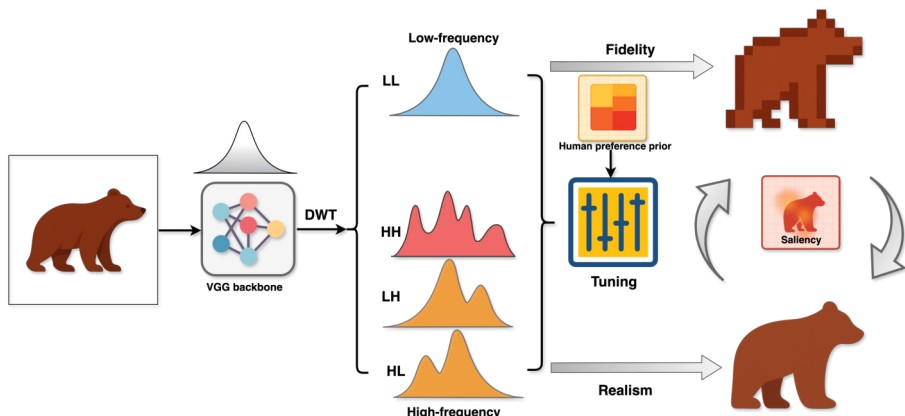

Figure 12: Illustration of Wavelet-WD: features are decomposed into subbands. The LL band captures structural content and can be optimized with a lower $\sigma$ range, while other bands can be adapted into a slightly higher $\sigma$ range to better model perceptual realism.

feature space, the structure of WA-WD thus follows this two-stage HVS structure, providing an interpretable motivation: wavelet decomposition (V1-like) + WD pooling (V2-like) yields a distortion that captures human sensitivity to structure and texture.

- **Inheriting the metric property from WD.** A core design principle of WA-WD is to retain the desirable theoretical properties of the original WD (Section D.1) while improving perceptual alignment and region-wise controllability for compression. Rather than modifying the WD formulation, we apply an invertible orthonormal wavelet transform to the feature representation and compute WD independently within each subband. Since $W$ is orthonormal and invertible, the metric properties of WD are preserved under this transformation: if $WD(\cdot, \cdot)^{1/p}$ is a *metric*, then $WD_{\text{wave}}(\cdot, \cdot)^{1/p}$ is *also* a metric (see Theorem 1).

- **Making the independence assumption more justified.** Practical WD computation relies on approximating local feature statistics as independent Gaussians (Ballé et al., 2025), assumptions that are often violated due to strong spatial and cross-channel correlations in neural features. Wavelet subbands, however, are known to be approximately decorrelated across scale and orientation for natural images (Figueiredo & Nowak, 2001; Simoncelli, 1999), making them far more compatible with the diagonal-covariance model used in WD approximation. Computing WD per subband thus make its statistical assumptions hold more closely. As formalized in Corollary 1, WA-WD provides a tighter bound on the approximation error, resulting in more perceptually consistent optimization for compression.

- **Towards a More Robust Implicit PMF.** Practical WD relies on an implicit PMF formed by VGG features combined with Gaussian-pyramid downsampling. However, VGG features are anisotropic and frequency-biased; after Gaussian smoothing, these biases are remixed across scales, easily creating spectral blind spots that undermine the "no spectral nulls" requirement (Theorem III.1 in (Qiu et al., 2024)) for stable WD behavior. WA-WD mitigates this by applying a wavelet transform before pooling, separating features into frequency-consistent subbands and preventing bias remixing. This potentially leads to more complete spectral coverage, a more balanced implicit PMF, and a more stable WD optimization with reduced metric degeneracy.

- **Fine-grained optimization.** Optimizing WD purely via saliency over mixed-frequency features can make high-frequency variance dominate, suppressing structural gradients and causing drift in non-salient regions. As a result, WD may overlook regional structure and frequency characteristics, leading to over-resampled textures in smooth non-salient areas (see Fig. 31) or excessive fidelity in high-frequency salient regions (see Fig. 32). To address this, LL/LH/HL/HH are optimized separately in Re$^2$IC, where LL provides clean structural-fidelity signals, while the high-frequency bands guide texture realism. Combined

Table 4: Comparison of standard overfitted codec with WD and Re$^2$IC with WA-WD, where the advantages of Re$^2$IC with WA-WD are summarized in green.

| Difference | WD codec (C3/WDs) | WA-WD codec (Re$^2$IC) |
|---|---|---|
| Feature space (Frequency-aware) | VGG feature domain. Optimization with mixed frequency gradients. | Wavelet-decomposed space. Optimization with **frequency-aware gradients**. |
| HVS alignment (Explicit and more aligned) | HVS-V2 processing. Modeling fovea-to-periphery integration with WD $\overset{\sim}{\leftrightarrow}$ HVS-V2. | HVS-V1 + HVS-V2 processing. DWT for **explicit** multi-scale, orientation decomposition $\overset{\sim}{\leftrightarrow}$ HVS-V1; WD $\overset{\sim}{\leftrightarrow}$ HVS-V2. |
| Feature assumption (Better justified) | Independent Gaussian over VGG. Neural features are highly correlated, so the independence assumption is often violated. | Independent Gaussians after DWT. DWT yields **orthogonal, decorrelated** subbands, making the independence assumptions more realistic within each subband. |
| Implicit PMF (Closer to the "no spectral nulls") | VGG ⊗ Gaussian pyramid. Anisotropic VGG bias combined with Gaussian smoothing further mixes and amplifies existing frequency biases, creating more pronounced spectral blind spots. | VGG ⊗ DWT ⊗ Gaussian pyramid. DWT separates frequency bands, preventing bias from being remixed across scales/frequencies and providing more **complete spectral coverage** for a more **balanced and robust** "PMF". |
| Granularity control (Adaptive tuning) | Coarse and scale-only control. Perception modeling (pooling) is controlled only by the Gaussian-pyramid scale, smoothing all frequencies jointly at each level. | Fine-grained and subband-wise control. Perception modeling is performed per subband, enabling **subband-wise control** or weights and **frequency-aware modulation** (e.g., flexible structure vs. texture modeling). |
| Coding strategy (Towards "Faster, cheaper, better") | Standard instance-wise overfitting. Standard overfitting each sample with a WD target, where each sample is fitted via a single neural function. | Region-wise perception modeling. Modeling perception region-by-region leads to **faster** convergence with a **lower-complexity** network, while enabling **more flexible** bit allocation under WA-WD and achieving a **better** RP. |
| Tunable metric (More human preference aligned) | WD computed in the VGG feature space. WD is computed directly in the neural space, whose behavior is largely determined by the network backbone, with limited flexibility. | Tunable WD in decomposed feature space. Each subband's contribution to WD ($\sigma$-range or weighting) can be **independently tuned**, enabling **adaptation** to various preference or degradation characteristics and a more **flexible** R–D–P trade-off. |

with the design of Re$^2$IC, this frequency and region-aware optimization can yield more stable, perceptually consistent compression while allowing tunable subband/region weights for further HVS alignment.

- **Tunable metric.** Serving as a metric, WD is computed directly in neural feature space, making its behavior largely determined by the backbone and therefore difficult to adjust. WA-WD introduces subband-level control (via tunable weights or $\sigma$-ranges). This allows adaption to different perceptual preferences or degradation characteristics and provides a more flexible RDP trade-off.

**Interpretation**

- Intuitive meaning of the $\sigma$-map: $\sigma$ is a spatial map controlling the size of the pooling window. For small $\sigma$, WA-WD models foveal vision under different sub-bands, which enforces high-fidelity reconstruction. For large $\sigma$, it models peripheral vision with more tolerance to texture resampling. Thus, $\sigma$ directly controls the realism–fidelity trade-off in a way that mimics the HVS.

- Frequency-aware optimization: Unlike WD, which mixes all frequencies, WA-WD separately optimizes distortions in structural (LL) and textural (LH/HL/HH) subbands. As a result, WA-WD measures perceptual differences in *multi-scale, frequency-decomposed feature statistics*, producing frequency-aware distortion and gradient that supports fine-grained DP control and more flexible, perceptually aligned bit allocation.

In summary, WA-WD preserves the core structure of WD while placing it in a more *interpretable* and *HVS-consistent* form for image compression. Specifically, this change has several practical implications: the *independence assumptions* underlying the WD approximation hold more closely, the resultant *implicit PMF* becomes more structurally coherent, and the *subband-wise decomposition* naturally supports region-wise perceptual modulation. Together, these properties improve the *stability, flexibility,* and *effectiveness* of bit allocation for RP optimization in image compression, while

preserving the original metric behavior of WD. A detailed comparison between $\text{Re}^2\text{IC}$ (WA-WD) and C3 (WDs) is provided in Table 4.

**Theoretical Properties**   WA-WD satisfies several desirable theoretical properties:

- **Metric preservation.** As shown in Theorem 1, WA-WD inherits the metric property of WD: if $WD(\cdot, \cdot)^{1/p}$ is a *metric*, then $WD_{\text{wave}}(\cdot, \cdot)^{1/p}$ is *also* a metric.
- **Tighter approximation.**   As shown in Corollary 1, WA-WD better satisfies the independent-Gaussianized approximation used in WD, yielding a strictly tighter bound on the resultant approximation error.
- **Statistics properties.** As shown in Corollary 2, compared with WD, WA-WD preserves a similar mean but differs in variance, where it incorporates cross-covariance terms and makes the metric explicitly sensitive to intra-block correlations. When $\sigma = 0$, both WD and WA-WD can reduce to mean squared error (MSE), measuring only fidelity distortion.

**Validation experiments**   To evaluate the benefits of WA-WD for frequency-aware and region-wise optimization, we provide two illustrative examples in Fig. 31 and Fig. 32. Additional experiments over the loss function can be seen in Table 11. Experiments on tunable WA-WD are shown in Fig. 14. Experiments for the DWT effect can be seen in Figs. 15, 18, and 19. The comparisons with other DWT-based codec can be seen in Figs. 16, 17.

**Tunable strategy**   As an independent perceptual metric (shown in Fig. 12), WA-WD can be tuned for different scenarios. The design principle is to emphasize fidelity in salient regions with low-frequency features, while assigning more realism to non-salient regions with high-frequency components. Here we list two typical tuning methods:

- **Tuning the sub-band weight.** Assigning different weights to subbands provides a simple way to control their distortion contributions. In general, the LL subband receives a higher weight, while HL, LH, and HH are assigned lower, equal weights. Specifically, in our setting, for simplicity, they are learnable parameters.
- **Tuning the $\sigma$-map per subband.** For a more fine-grained optimization, we can compute WD separately for each subband with distinct $\sigma$ ranges: LL uses a narrow range to preserve fidelity, HL and LH share a moderately larger range, and HH is assigned the largest range to better capture high-frequency distortions.

In this way, even within the same saliency region, different bands can be optimized differently according to their frequency characteristics.

## E   PROOF

### E.1   PROOF OF THEOREM 1: METRIC PRESERVATION

Let $\boldsymbol{W} \in \mathbb{R}^{d \times d}$ denote an orthonormal discrete wavelet transform (DWT) and $WD(\cdot)$ denote the Wasserstein distortion. Define the wavelet Wasserstein distortion as $WD_{\text{wave}}(\boldsymbol{z}, \hat{\boldsymbol{z}}) = WD(\boldsymbol{W}\boldsymbol{z}, \boldsymbol{W}\hat{\boldsymbol{z}})$. Then the following hold:

- If PMFs $q_\sigma$ have no spectral nulls and $d(\cdot)$ is a metric (i.e, $WD(\boldsymbol{z}, \hat{\boldsymbol{z}})^{1/p}$ is a metric), $WD_{\text{wave}}(\boldsymbol{z}, \hat{\boldsymbol{z}})^{1/p}$ is a metric.
- If $\phi(\cdot)$ is invertible (i.e, $WD(\boldsymbol{x}, \hat{\boldsymbol{x}})^{1/p}$ is a metric), $WD_{\text{wave}}(\boldsymbol{x}, \hat{\boldsymbol{x}})^{1/p}$ is also a metric.

*Proof.* Since $WD(\boldsymbol{z}, \hat{\boldsymbol{z}})^{1/p}$ is a metric, whenever $d(\cdot)$ is a metric and $q_\sigma$ has no spectral nulls, and since $\boldsymbol{W}$ is linear and invertible, it follows that $WD_{\text{wave}}(\boldsymbol{z}, \hat{\boldsymbol{z}})^{1/p}$ inherits all metric properties from $WD(\boldsymbol{z}, \hat{\boldsymbol{z}})^{1/p}$. In particular, $WD_{\text{wave}}$ is symmetric and satisfies $WD_{\text{wave}}(\boldsymbol{z}, \boldsymbol{z}) = 0$.

For any $\boldsymbol{z} \neq \hat{\boldsymbol{z}}$, the invertibility of $\boldsymbol{W}$ guarantees $\boldsymbol{W}\boldsymbol{z} \neq \boldsymbol{W}\hat{\boldsymbol{z}}$. Because $q_\sigma$ has no spectral nulls, therefore

$$q_\sigma * (\boldsymbol{W}\boldsymbol{z}) \; \neq \; q_\sigma * (\boldsymbol{W}\hat{\boldsymbol{z}}), \tag{12}$$

where $*$ denotes circular convolution (Oppenheim, 1999). Let $\boldsymbol{y}$ and $\hat{\boldsymbol{y}}$ denote these pooled signals. Since $W_p$ is a metric (Villani et al., 2008), $\boldsymbol{y} \neq \hat{\boldsymbol{y}}$ implies $W_p^p(y_l, \hat{y}_l) > 0$ for some $l$, and hence $WD_{\text{wave}}(\boldsymbol{z}, \hat{\boldsymbol{z}}) > 0$.

If $\phi$ is invertible, then $\boldsymbol{x} \neq \hat{\boldsymbol{x}}$ implies $\boldsymbol{z} \neq \hat{\boldsymbol{z}}$, and thus $WD_{\text{wave}}(\boldsymbol{x}, \hat{\boldsymbol{x}}) > 0$ as well. The triangle inequality for $WD_{\text{wave}}(\boldsymbol{x}, \hat{\boldsymbol{x}})$ and $WD_{\text{wave}}(\boldsymbol{z}, \hat{\boldsymbol{z}}) > 0$ follows directly from the fact that $W_p$ is a metric and from Minkowski's inequality.

In a nutshell, $WD_{\text{wave}}^{1/p}$ is a metric on the feature space whenever PMF has no spectral nulls, and it becomes a metric on the input space when the feature extractor $\phi$ is invertible. $\qquad\square$

**Remark 1** (Equivalence to practical blockwise implementation). *In practice, with the diagonal-Gaussian assumption, WA-WD can be implemented by computing the WD independently in each subband and summing their contributions with equal weight.*

Since the DWT is assumed orthonormal, which can be decomposed as $\boldsymbol{W}^\top = \begin{bmatrix} \boldsymbol{W}_{LL}^\top & \boldsymbol{W}_{LH}^\top & \boldsymbol{W}_{HL}^\top & \boldsymbol{W}_{HH}^\top \end{bmatrix}$. Under the practical diagonal-Gaussian assumption, the Wasserstein distance becomes blockwise additive: $WD(\boldsymbol{W}\boldsymbol{z}, \boldsymbol{W}\hat{\boldsymbol{z}}) = \sum_{b \in \{LL, LH, HL, HH\}} WD(\boldsymbol{W}_b \boldsymbol{z}, \boldsymbol{W}_b \hat{\boldsymbol{z}})$.

### E.2 PROOF OF LEMMA 1: ERROR BOUND FOR DIAGONAL-GAUSSIAN APPROXIMATION

Let $\boldsymbol{X}$ and $\boldsymbol{Y}$ be local Gaussian patches with covariances $\boldsymbol{\Sigma}_{\boldsymbol{X}}$ and $\boldsymbol{\Sigma}_{\boldsymbol{Y}}$, and let $\widetilde{WD}$ denote the diagonal-Gaussian approximation of $WD$. Then there exists a constant $C > 0$, depending only on the eigenvalue bounds of $\boldsymbol{\Sigma}_{\boldsymbol{X}}$ and $\boldsymbol{\Sigma}_{\boldsymbol{Y}}$, such that

$$|WD(\boldsymbol{X}, \boldsymbol{Y}) - \widetilde{WD}(\boldsymbol{X}, \boldsymbol{Y})| \leq C\big(\|\text{offdiag}(\boldsymbol{\Sigma}_{\boldsymbol{X}})\|_F + \|\text{offdiag}(\boldsymbol{\Sigma}_{\boldsymbol{Y}})\|_F\big). \tag{13}$$

If the eigenvalues satisfy

$$0 < m \leq \lambda_{\min}(\boldsymbol{\Sigma}_{\boldsymbol{X}}), \lambda_{\min}(\boldsymbol{\Sigma}_{\boldsymbol{Y}}) \leq \lambda_{\max}(\boldsymbol{\Sigma}_{\boldsymbol{X}}), \lambda_{\max}(\boldsymbol{\Sigma}_{\boldsymbol{Y}}) \leq M < \infty, \tag{14}$$

then the constant is:

$$C = 2\sqrt{d}\left(1 + \frac{M}{2m} + \frac{M^{3/2}}{2m^{3/2}}\right). \tag{15}$$

*Proof.* Consider $\boldsymbol{X} \sim \mathcal{N}(\boldsymbol{\mu}_X, \boldsymbol{\Sigma}_{\boldsymbol{X}})$ and $\boldsymbol{Y} \sim \mathcal{N}(\boldsymbol{\mu}_Y, \boldsymbol{\Sigma}_{\boldsymbol{Y}})$ be $d$-dimensional Gaussian feature vectors with symmetric positive definite covariance matrices $\boldsymbol{\Sigma}_{\boldsymbol{X}}, \boldsymbol{\Sigma}_{\boldsymbol{Y}} \in \mathbb{R}^{d \times d}$. The exact squared 2-Wasserstein distance between the two Gaussians is:

$$WD(\boldsymbol{X}, \boldsymbol{Y}) = \|\boldsymbol{\mu}_X - \boldsymbol{\mu}_Y\|_2^2 + \text{Tr}\left(\boldsymbol{\Sigma}_{\boldsymbol{X}} + \boldsymbol{\Sigma}_{\boldsymbol{Y}} - 2\big(\boldsymbol{\Sigma}_{\boldsymbol{Y}}^{1/2}\boldsymbol{\Sigma}_{\boldsymbol{X}}\boldsymbol{\Sigma}_{\boldsymbol{Y}}^{1/2}\big)^{1/2}\right). \tag{16}$$

Meanwhile, the diagonal-Gaussian approximation in (Qiu et al., 2024, Eq. 12), obtained by replacing the covariances with their diagonal parts, is: $\widetilde{WD}(\boldsymbol{X}, \boldsymbol{Y}) = \|\boldsymbol{\mu}_X - \boldsymbol{\mu}_Y\|_2^2 + \|\boldsymbol{\sigma}_X - \boldsymbol{\sigma}_Y\|_2^2$, where $\boldsymbol{\sigma}_X, \boldsymbol{\sigma}_Y$ collect the square roots of the diagonal entries of $\boldsymbol{\Sigma}_{\boldsymbol{X}}, \boldsymbol{\Sigma}_{\boldsymbol{Y}}$. Define the diagonal parts $\boldsymbol{\Sigma}_{\boldsymbol{X}}^d \triangleq \text{diag}(\boldsymbol{\Sigma}_{\boldsymbol{X}})$ and $\boldsymbol{\Sigma}_{\boldsymbol{Y}}^d \triangleq \text{diag}(\boldsymbol{\Sigma}_{\boldsymbol{Y}})$:

$$\widetilde{WD}(\boldsymbol{X}, \boldsymbol{Y}) = \|\boldsymbol{\mu}_X - \boldsymbol{\mu}_Y\|_2^2 + \text{Tr}\left(\boldsymbol{\Sigma}_{\boldsymbol{X}}^d + \boldsymbol{\Sigma}_{\boldsymbol{Y}}^d - 2\big((\boldsymbol{\Sigma}_{\boldsymbol{Y}}^d)^{1/2}\boldsymbol{\Sigma}_{\boldsymbol{X}}^d(\boldsymbol{\Sigma}_{\boldsymbol{Y}}^d)^{1/2}\big)^{1/2}\right). \tag{17}$$

Using linearity of the trace and the triangle inequality, we can bound this difference by a constant (depending on $d$) times

$$
\left| WD(\boldsymbol{X}, \boldsymbol{Y}) - \widetilde{WD}(\boldsymbol{X}, \boldsymbol{Y}) \right|
$$

$$
= \left| \mathrm{Tr}\left( \boldsymbol{\Sigma}_{\boldsymbol{X}} + \boldsymbol{\Sigma}_{\boldsymbol{Y}} - 2(\boldsymbol{\Sigma}_{\boldsymbol{Y}}^{1/2} \boldsymbol{\Sigma}_{\boldsymbol{X}} \boldsymbol{\Sigma}_{\boldsymbol{Y}}^{1/2})^{1/2} \right) - \mathrm{Tr}\left( \boldsymbol{\Sigma}_{\boldsymbol{X}}^{d} + \boldsymbol{\Sigma}_{\boldsymbol{Y}}^{d} - 2((\boldsymbol{\Sigma}_{\boldsymbol{Y}}^{d})^{1/2} \boldsymbol{\Sigma}_{\boldsymbol{X}}^{d} (\boldsymbol{\Sigma}_{\boldsymbol{Y}}^{d})^{1/2})^{1/2} \right) \right|
$$

$$
= \left| \mathrm{Tr}\left( (\boldsymbol{\Sigma}_{\boldsymbol{X}} + \boldsymbol{\Sigma}_{\boldsymbol{Y}} - 2(\boldsymbol{\Sigma}_{\boldsymbol{Y}}^{1/2} \boldsymbol{\Sigma}_{\boldsymbol{X}} \boldsymbol{\Sigma}_{\boldsymbol{Y}}^{1/2})^{1/2}) - (\boldsymbol{\Sigma}_{\boldsymbol{X}}^{d} + \boldsymbol{\Sigma}_{\boldsymbol{Y}}^{d} - 2((\boldsymbol{\Sigma}_{\boldsymbol{Y}}^{d})^{1/2} \boldsymbol{\Sigma}_{\boldsymbol{X}}^{d} (\boldsymbol{\Sigma}_{\boldsymbol{Y}}^{d})^{1/2})^{1/2}) \right) \right|
$$

$$
= \left| \mathrm{Tr}\left( (\boldsymbol{\Sigma}_{\boldsymbol{X}} - \boldsymbol{\Sigma}_{\boldsymbol{X}}^{d}) + (\boldsymbol{\Sigma}_{\boldsymbol{Y}} - \boldsymbol{\Sigma}_{\boldsymbol{Y}}^{d}) - 2\left[ (\boldsymbol{\Sigma}_{\boldsymbol{Y}}^{1/2} \boldsymbol{\Sigma}_{\boldsymbol{X}} \boldsymbol{\Sigma}_{\boldsymbol{Y}}^{1/2})^{1/2} - ((\boldsymbol{\Sigma}_{\boldsymbol{Y}}^{d})^{1/2} \boldsymbol{\Sigma}_{\boldsymbol{X}}^{d} (\boldsymbol{\Sigma}_{\boldsymbol{Y}}^{d})^{1/2})^{1/2} \right] \right) \right|
$$

$$
\leq \left| \mathrm{Tr}(\boldsymbol{\Sigma}_{\boldsymbol{X}} - \boldsymbol{\Sigma}_{\boldsymbol{X}}^{d}) \right| + \left| \mathrm{Tr}(\boldsymbol{\Sigma}_{\boldsymbol{Y}} - \boldsymbol{\Sigma}_{\boldsymbol{Y}}^{d}) \right| + 2 \left| \mathrm{Tr}((\boldsymbol{\Sigma}_{\boldsymbol{Y}}^{1/2} \boldsymbol{\Sigma}_{\boldsymbol{X}} \boldsymbol{\Sigma}_{\boldsymbol{Y}}^{1/2})^{1/2} - ((\boldsymbol{\Sigma}_{\boldsymbol{Y}}^{d})^{1/2} \boldsymbol{\Sigma}_{\boldsymbol{X}}^{d} (\boldsymbol{\Sigma}_{\boldsymbol{Y}}^{d})^{1/2})^{1/2}) \right|
$$

$$
\leq C_0 \left( \|\boldsymbol{\Sigma}_{\boldsymbol{X}} - \boldsymbol{\Sigma}_{\boldsymbol{X}}^{d}\|_F + \|\boldsymbol{\Sigma}_{\boldsymbol{Y}} - \boldsymbol{\Sigma}_{\boldsymbol{Y}}^{d}\|_F + \left\| (\boldsymbol{\Sigma}_{\boldsymbol{Y}}^{1/2} \boldsymbol{\Sigma}_{\boldsymbol{X}} \boldsymbol{\Sigma}_{\boldsymbol{Y}}^{1/2})^{1/2} - ((\boldsymbol{\Sigma}_{\boldsymbol{Y}}^{d})^{1/2} \boldsymbol{\Sigma}_{\boldsymbol{X}}^{d} (\boldsymbol{\Sigma}_{\boldsymbol{Y}}^{d})^{1/2})^{1/2} \right\|_F \right),
$$

$$\tag{18}$$

where $C_0 = 2\sqrt{d}$, derived from $|\mathrm{Tr}(\boldsymbol{M})| \leq \sqrt{d}\|\boldsymbol{M}\|_F$.

With (Higham, 2008, Theorem 6.2), the matrix square root is Fréchet differentiable and Lipschitz continuous on the set of symmetric positive definite matrices: $\|\boldsymbol{A}^{1/2} - \boldsymbol{B}^{1/2}\| \leq L \|\boldsymbol{A} - \boldsymbol{B}\|$, where $L = \frac{1}{\lambda_{min}(\boldsymbol{A})^{1/2} + \lambda_{min}(\boldsymbol{B})^{1/2}}$, where $\lambda_{min}$ is the smallest eigenvalues. We can then bound the last term: $\|\boldsymbol{A} - \boldsymbol{B}\|_F = \left\| (\boldsymbol{\Sigma}_{\boldsymbol{Y}}^{1/2} \boldsymbol{\Sigma}_{\boldsymbol{X}} \boldsymbol{\Sigma}_{\boldsymbol{Y}}^{1/2}) - ((\boldsymbol{\Sigma}_{\boldsymbol{Y}}^{d})^{1/2} \boldsymbol{\Sigma}_{\boldsymbol{X}}^{d} (\boldsymbol{\Sigma}_{\boldsymbol{Y}}^{d})^{1/2}) \right\|_F \leq \|\boldsymbol{T_1}\|_F + \|\boldsymbol{T_2}\|_F$, where

$$
A - B = \boldsymbol{\Sigma}_{\boldsymbol{Y}}^{1/2} \boldsymbol{\Sigma}_{\boldsymbol{X}} \boldsymbol{\Sigma}_{\boldsymbol{Y}}^{1/2} - (\boldsymbol{\Sigma}_{\boldsymbol{Y}}^{d})^{1/2} \boldsymbol{\Sigma}_{\boldsymbol{X}}^{d} (\boldsymbol{\Sigma}_{\boldsymbol{Y}}^{d})^{1/2}
$$

$$
= \underbrace{\boldsymbol{\Sigma}_{\boldsymbol{Y}}^{1/2} \boldsymbol{\Sigma}_{\boldsymbol{X}} \boldsymbol{\Sigma}_{\boldsymbol{Y}}^{1/2} - \boldsymbol{\Sigma}_{\boldsymbol{Y}}^{1/2} \boldsymbol{\Sigma}_{\boldsymbol{X}}^{d} \boldsymbol{\Sigma}_{\boldsymbol{Y}}^{1/2}}_{\text{depends only on } \boldsymbol{\Sigma}_{\boldsymbol{X}} - \boldsymbol{\Sigma}_{\boldsymbol{X}}^{d}} + \underbrace{\boldsymbol{\Sigma}_{\boldsymbol{Y}}^{1/2} \boldsymbol{\Sigma}_{\boldsymbol{X}}^{d} \boldsymbol{\Sigma}_{\boldsymbol{Y}}^{1/2} - (\boldsymbol{\Sigma}_{\boldsymbol{Y}}^{d})^{1/2} \boldsymbol{\Sigma}_{\boldsymbol{X}}^{d} (\boldsymbol{\Sigma}_{\boldsymbol{Y}}^{d})^{1/2}}_{\text{depends only on } \boldsymbol{\Sigma}_{\boldsymbol{Y}} - \boldsymbol{\Sigma}_{\boldsymbol{Y}}^{d}}
$$

$$
= \underbrace{\boldsymbol{\Sigma}_{\boldsymbol{Y}}^{1/2} (\boldsymbol{\Sigma}_{\boldsymbol{X}} - \boldsymbol{\Sigma}_{\boldsymbol{X}}^{d}) \boldsymbol{\Sigma}_{\boldsymbol{Y}}^{1/2}}_{\boldsymbol{T_1}} + \underbrace{\left( \boldsymbol{\Sigma}_{\boldsymbol{Y}}^{1/2} \boldsymbol{\Sigma}_{\boldsymbol{X}}^{d} \boldsymbol{\Sigma}_{\boldsymbol{Y}}^{1/2} - (\boldsymbol{\Sigma}_{\boldsymbol{Y}}^{d})^{1/2} \boldsymbol{\Sigma}_{\boldsymbol{X}}^{d} (\boldsymbol{\Sigma}_{\boldsymbol{Y}}^{d})^{1/2} \right)}_{\boldsymbol{T_2}}. \tag{19}
$$

For $T_1$ with $C_1 = \|\boldsymbol{\Sigma}_{\boldsymbol{Y}}^{1/2}\|_2^2 \leq M$:

$$
\|\boldsymbol{T_1}\|_F = \|\boldsymbol{\Sigma}_{\boldsymbol{Y}}^{1/2} (\boldsymbol{\Sigma}_{\boldsymbol{X}} - \boldsymbol{\Sigma}_{\boldsymbol{X}}^{d}) \boldsymbol{\Sigma}_{\boldsymbol{Y}}^{1/2}\|_F \leq \|\boldsymbol{\Sigma}_{\boldsymbol{Y}}^{1/2}\|_2^2 \|\boldsymbol{\Sigma}_{\boldsymbol{X}} - \boldsymbol{\Sigma}_{\boldsymbol{X}}^{d}\|_F \leq C_1 \|\boldsymbol{\Sigma}_{\boldsymbol{X}} - \boldsymbol{\Sigma}_{\boldsymbol{X}}^{d}\|_F. \tag{20}
$$

For $T_2$, with $C_2' = \|D_X\|_2(\|A_Y\|_2 + \|B_Y\|_2) \leq 2M^{3/2}$, and $C_2 = \frac{M^{3/2}}{\sqrt{m}}$: let $A_Y = \boldsymbol{\Sigma}_{\boldsymbol{Y}}^{1/2}$, $B_Y = (\boldsymbol{\Sigma}_{\boldsymbol{Y}}^{d})^{1/2}$ and $D_X = \boldsymbol{\Sigma}_{\boldsymbol{X}}^{d}$, where $T_2 = A_Y D_X A_Y - B_Y D_X B_Y = (A_Y - B_Y) D_X A_Y + B_Y D_X (A_Y - B_Y)$. We can have:

$$
\begin{aligned}
\|T_2\|_F &\leq \|(A_Y - B_Y) D_X A_Y\|_F + \|B_Y D_X (A_Y - B_Y)\|_F \\
&\leq \|A_Y - B_Y\|_F \|D_X\|_2 \|A_Y\|_2 + \|B_Y\|_2 \|D_X\|_2 \|A_Y - B_Y\|_F \\
&= (\|D_X\|_2 \|A_Y\|_2 + \|B_Y\|_2 \|D_X\|_2) \|A_Y - B_Y\|_F \\
&= C_2' \|A_Y - B_Y\|_F \leq C_2 \|\boldsymbol{\Sigma}_{\boldsymbol{Y}} - \boldsymbol{\Sigma}_{\boldsymbol{Y}}^{d}\|_F. 
\end{aligned} \tag{21}
$$

Using equation 19, equation 20, and equation 21, we can have:

$$
\|A - B\|_F \leq C_1 \|\boldsymbol{\Sigma}_{\boldsymbol{X}} - \boldsymbol{\Sigma}_{\boldsymbol{X}}^{d}\|_F + C_2 \|\boldsymbol{\Sigma}_{\boldsymbol{Y}} - \boldsymbol{\Sigma}_{\boldsymbol{Y}}^{d}\|_F. \tag{22}
$$

Finally, substitute into the last term of equation 18, with $L_1 = \frac{1}{2m}$:

$$
\begin{aligned}
\|A^{1/2} - B^{1/2}\|_F &\leq L_1 \|A - B\|_F \\
&\leq L_1 (C_1 + C_2) \left( \|\boldsymbol{\Sigma}_{\boldsymbol{X}} - \boldsymbol{\Sigma}_{\boldsymbol{X}}^{d}\|_F + \|\boldsymbol{\Sigma}_{\boldsymbol{Y}} - \boldsymbol{\Sigma}_{\boldsymbol{Y}}^{d}\|_F \right). 
\end{aligned} \tag{23}
$$

Let $L = L_1(C_1 + C_2) \leq \frac{1}{2m}(M + \frac{M^{3/2}}{\sqrt{m}})$, we can obtain:

$$\left\| (\boldsymbol{\Sigma_Y}^{1/2} \boldsymbol{\Sigma_X} \boldsymbol{\Sigma_Y}^{1/2})^{1/2} - ((\boldsymbol{\Sigma_Y^d})^{1/2} \boldsymbol{\Sigma_X^d} (\boldsymbol{\Sigma_Y^d})^{1/2})^{1/2} \right\|_F \leq L\Big( \|\boldsymbol{\Sigma_X} - \boldsymbol{\Sigma_X^d}\|_F + \|\boldsymbol{\Sigma_Y} - \boldsymbol{\Sigma_Y^d}\|_F \Big). \tag{24}$$

The first two terms are exactly $\|\mathrm{offdiag}(\boldsymbol{\Sigma_X})\|_F$ and $\|\mathrm{offdiag}(\boldsymbol{\Sigma_Y})\|_F$. Combining these estimates and absorbing constants into a single $C > 0$ (depends only on the $\boldsymbol{\Sigma_X}, \boldsymbol{\Sigma_Y}$) yields:

$$\left| WD(\boldsymbol{X}, \boldsymbol{Y}) - \widetilde{WD}(\boldsymbol{X}, \boldsymbol{Y}) \right| \leq C\Big( \|\boldsymbol{\Sigma_X} - \boldsymbol{\Sigma_X^d}\|_F + \|\boldsymbol{\Sigma_Y} - \boldsymbol{\Sigma_Y^d}\|_F \Big)$$

$$= C\Big( \|\mathrm{offdiag}(\boldsymbol{\Sigma_X})\|_F + \|\mathrm{offdiag}(\boldsymbol{\Sigma_Y})\|_F \Big), \tag{25}$$

which proves the claim with $C = 2\sqrt{d}(1 + \frac{M}{2m} + \frac{M^{3/2}}{2m^{3/2}})$. $\square$

**Remark 2** (The bounded-eigenvalue assumption). *Bounding the eigenvalues of $\boldsymbol{\Sigma_X}$ and $\boldsymbol{\Sigma_Y}$ within $[m, M]$ is a standard requirement (Wang et al., 2024) ensuring the matrix square root is Lipschitz continuous. In deep feature spaces, this assumption is mild: local covariances are empirically well-conditioned due to batch normalization, bounded nonlinearities. As a result, feature statistics can remain within a compact region across images and datasets, making the bounded-eigenvalue assumption compatible with WD-based image compression.*

### E.3  PROOF OF COROLLARY 1: TIGHTER DIAGONAL-GAUSSIAN APPROXIMATION

*Proof.* Let $\boldsymbol{X}$ and $\boldsymbol{Y}$ be local Gaussian patches with covariances $\boldsymbol{\Sigma_X}$ and $\boldsymbol{\Sigma_Y}$. Let $\widetilde{WD}$ and $\widetilde{WD}_{\mathrm{wave}}$ denote their diagonal-Gaussian approximations in the original and wavelet domains. Suppose the original covariances $\boldsymbol{\Sigma_X}, \boldsymbol{\Sigma_Y}$ satisfy the eigenvalue bounds $mI \preceq \boldsymbol{\Sigma_X}, \boldsymbol{\Sigma_Y} \preceq MI$, and let $\tilde{\boldsymbol{\Sigma}}_X = W\boldsymbol{\Sigma_X}W^\top, \tilde{\boldsymbol{\Sigma}}_Y = W\boldsymbol{\Sigma_Y}W^\top$ be the covariances after an orthonormal wavelet transform $W$. Since orthonormal transforms preserve eigenvalues, both $\tilde{\boldsymbol{\Sigma}}_X$ and $\tilde{\boldsymbol{\Sigma}}_Y$ satisfy the same bounds:

$$mI \preceq \tilde{\boldsymbol{\Sigma}}_X, \tilde{\boldsymbol{\Sigma}}_Y \preceq MI. \tag{26}$$

Under the same constant $C = 2\sqrt{d}\Big(1 + \frac{M}{2m} + \frac{M^{3/2}}{2m^{3/2}}\Big)$. appearing in Lemma 1, the approximation errors satisfy:

$$|WD(\tilde{\boldsymbol{X}}, \tilde{\boldsymbol{Y}}) - \widetilde{WD}(\tilde{\boldsymbol{X}}, \tilde{\boldsymbol{Y}})| \leq C\Big( \|\mathrm{offdiag}(\tilde{\boldsymbol{\Sigma}}_X)\|_F + \|\mathrm{offdiag}(\tilde{\boldsymbol{\Sigma}}_Y)\|_F \Big), \tag{27}$$

$$|WD(\boldsymbol{X}, \boldsymbol{Y}) - \widetilde{WD}(\boldsymbol{X}, \boldsymbol{Y})| \leq C(\|\mathrm{offdiag}(\boldsymbol{\Sigma_X})\|_F + \|\mathrm{offdiag}(\boldsymbol{\Sigma_Y})\|_F), \tag{28}$$

and because natural-image wavelet coefficients are well known to be approximately decorrelated (Figueiredo & Nowak, 2001; Simoncelli, 1999):

$$\|\mathrm{offdiag}(\tilde{\boldsymbol{\Sigma}}_X)\|_F \ll \|\mathrm{offdiag}(\boldsymbol{\Sigma_X})\|_F, \qquad \|\mathrm{offdiag}(\tilde{\boldsymbol{\Sigma}}_Y)\|_F \ll \|\mathrm{offdiag}(\boldsymbol{\Sigma_Y})\|_F, \tag{29}$$

we obtain a tighter error bound for WA-WD:

$$C\Big( \|\mathrm{offdiag}(\tilde{\boldsymbol{\Sigma}}_X)\|_F + \|\mathrm{offdiag}(\tilde{\boldsymbol{\Sigma}}_Y)\|_F \Big) \leq C\big( \|\mathrm{offdiag}(\boldsymbol{\Sigma_X})\|_F + \|\mathrm{offdiag}(\boldsymbol{\Sigma_Y})\|_F \big). \tag{30}$$

Thus, WA-WD yields a tighter error bound of WD whenever the wavelet transform reduces off-diagonal covariance energy. $\square$

**Remark 3** (Analyzing bounds rather than pointwise comparisons). *A direct pointwise comparison of WD approximation error before and after a wavelet transform is generally infeasible, since the exact Wasserstein term involves a matrix square root (see Eqn. 10) whose behavior depends on image content and local feature correlations. Thus, the quantity $|WD - \widetilde{WD}|$ can vary across patches and cannot be compared fairly without strong assumptions over source distributions. This motivates analyzing error bounds rather than pointwise differences. Bounding the approximation error in terms of off-diagonal covariance energy provides a content-independent criterion. Because DWT substantially reduces cross-correlation in natural images, these bounds become systematically tighter after DWT. Hence, a bound-based analysis captures the intrinsic advantage of WA-WD even when pointwise comparison is not tractable.*

### E.4 Proof of Corollary 2: Statistical properties of WA-WD

*Proof.* As shown in Sections 3.1 and 3.2, under independent Gaussian approximation, the standard local WD between each feature component $z_i$ and its reconstruction $\hat{z}_i$ is:

$$D_{i,\sigma} = (\mu_i - \hat{\mu}_i)^2 + \left(\sqrt{V_i} - \sqrt{\hat{V}_i}\right)^2, \tag{31}$$

where $\mu_i$ and $V_i$ denote the local mean and variance of each element:

$$\mu_i = \sum_k q_\sigma(k)\, z_{i+k}, V_i = \sum_k q_\sigma(k)\left(z_{i+k} - \mu_i\right)^2. \tag{32}$$

The final WD is averaged over all locations as: $WD(\boldsymbol{z}, \hat{\boldsymbol{z}}) = \frac{1}{|\mathcal{I}|} \sum_{i \in \mathcal{I}} D_{i,\sigma}$.

For WA-WD, the resultant pooled mean and variance are:

$$\mu_i^b = \sum_k q_\sigma(k)\, z_{i+k}^b = h_b^\top \Big( \sum_k q_\sigma(k)\, w_{i+k} \Big) \triangleq \boldsymbol{h_b^T \bar{w}_i}, \tag{33}$$

$$V_i^b = \sum_k q_\sigma(k)\left(z_{i+k}^b - \mu_i^b\right)^2 = \sum_k q_\sigma(k)\big(\boldsymbol{h_b^T}(\boldsymbol{w_{i+k}} - \boldsymbol{\bar{w}_i})\big)^2 = \boldsymbol{h_b^T C_i h_b}, \tag{34}$$

where $\boldsymbol{C_i} \triangleq \sum_k q_\sigma(k)(\boldsymbol{w_{i+k}} - \boldsymbol{\bar{w}_i})(\boldsymbol{w_{i+k}} - \boldsymbol{\bar{w}_i})^\top$ denotes a pooled $4 \times 4$ *block covariance matrix*, $z_i^b = \boldsymbol{h_b^\top w_i}$ denotes the Haar transform coefficients, with the Haar basis $\boldsymbol{h_b} \in \mathbb{R}^4$, and $\boldsymbol{w_i} \in \mathbb{R}^4$ is the vectorization of each feature block. With independence approximation, the perband WD and the overall WA-WD are given as: $D_{i,\sigma}^b = (\mu_i^b - \hat{\mu}_i^b)^2 + \left(\sqrt{V_i^b} - \sqrt{\hat{V}_i^b}\right)^2$ and $WD_{\text{wave}}(\boldsymbol{z}, \hat{\boldsymbol{z}}) = \frac{1}{4} \sum_{b \in \{LL, LH, HL, HH\}} \frac{1}{|\mathcal{I}_b|} \sum_{i \in \mathcal{I}_b} D_{i,\sigma}^b$, respectively.

1. **Mean preservation.** Compare Eqn. 32 and Eqn. 33, since both pooling and the Haar transform are linear,
$$\mu_i^b = \boldsymbol{h_b^\top \mu_i},$$
meaning WA-WD preserves the first-order statistics of WD up to an orthonormal transform.

2. **Variance decomposition with correlation sensitivity.** From Eqn. 34, the variance of the each subband for WA-WD is obtained as
$$V_i^b = \boldsymbol{h_b^\top C_i h_b}.$$
This explicitly depends on all entries of $C_i$, including cross-covariances (off-diagonal terms). The variance for WD (as shown in Eqn. 32), in contrast:

$$V_i = \sum_k q_\sigma(k)\left(z_{i+k} - \mu_i\right)^2.$$

uses only $\text{diag}(\boldsymbol{C_i})$. Hence WA-WD captures intra-block correlations that standard WD ignores.

3. **Reduction to MSE when $\sigma = 0$.** When the pooling scale $\sigma(i) = 0$ at a location $i$, the pooling kernel degenerates to a single spatial point. This yields $D_{i,0} = (z_i - \hat{z}_i)^2$, i.e., WD reduces to the pointwise squared error. For WA-WD, the same happens in the wavelet domain. By orthonormality of $W$ (Parseval's theorem), we can have:

$$\sum_b (z_i^b - \hat{z}_i^b)^2 = \|W z_i - W \hat{z}_i\|_2^2 = \|z_i - \hat{z}_i\|_2^2.$$

Thus, when $\sigma = 0$ everywhere, both WD and WA-WD reduce exactly to mean squared error (MSE), reflecting pure fidelity distortion without any pooling.

$\square$

To further illustrate their variance difference, we provide several block-distribution examples in Table 5, demonstrating how WA-WD captures intra-block correlations.

Table 5: Special cases of block covariance structures and their effect on wavelet-WD, where $p$ means the indexes of pixels in a block and $\sigma_p^2$ denotes their variance ($\sigma^2$ is the common variance).

| Block distribution | Effect over Wavelet-WD |
|---|---|
| Independent | $C = \text{diag}(\sigma_1^2, \sigma_2^2, \sigma_3^2, \sigma_4^2)$. No cross-terms; wavelet variances reduce to weighted sums of pixel variances. The difference from WD is minimal. |
| Fully correlated | $C = \sigma^2 \mathbf{1}\mathbf{1}_{4\times 4}$. All energy collapses into $LL$: $V^{LL} = \sigma^2$, while $V^{LH} = V^{HL} = V^{HH} = 0$. Wavelet-WD focuses on low-frequency distortion for smooth areas. |
| i.i.d. noise | $C = \sigma^2 I_{4\times 4}$. Energy spreads evenly across subbands: $V^{LL} = V^{LH} = V^{HL} = V^{HH} = \sigma^2$. Represents uncorrelated, isotropic fluctuations. |
| Directional correlation | Strong horizontal correlation (horizontal edges) $\Rightarrow V^{LH}$ dominates. Strong vertical correlation (vertical edges) $\Rightarrow V^{HL}$ dominates. |
| Checkerboard correlation | Variance concentrates in $HH$ subband, which captures diagonal alternations. |
| Partial correlation | $C = \sigma^2[(1-\rho)I_{4\times 4} + \rho\mathbf{1}_{4\times 4}]$, interpolating between i.i.d. ($\rho = 0$) and fully correlated ($\rho = 1$). Then $V^{LL} = \sigma^2(1 + 3\rho)$, while $V^{LH} = V^{HL} = V^{HH} = \sigma^2(1 - \rho)$. Energy shifts smoothly from high-frequency to low-frequency as $\rho$ increases. |

Table 6: Experiments for additional diffusion baselines: TACO (Lee et al., 2024) and CDC (Yang & Mandt, 2023) on CLIC2020, where we select the baseline with their most relevant rate regimes. **Red** and blue denote the best and second-best results. Re$^2$IC consistently delivers **state-of-the-art or leading** performance across FID, DISTS, KID, CLIP-IQA, and MANIQA, even compared with the current strongest diffusion model, TACO, at the target medium rate.

| | Model | BPP↓ | PSNR↑ | MS-SSIM↑ | LPIPS↓ | DISTS↓ | FID↓ | KID↓ | NIQE↓ | MUSIQ↑ | ClipIQA↑ | MANIQA↑ |
|---|---|---|---|---|---|---|---|---|---|---|---|---|
| Medium | VTM | 0.1480 | **32.39** | **0.9665** | 0.2495 | 0.1837 | 89.83 | 47.54 | 5.3504 | 57.34 | 0.4230 | **0.3521** |
| | HiFiC | 0.1420 | 29.83 | 0.9594 | 0.0589 | 0.0695 | 18.81 | 2.33 | **3.3044** | **59.91** | 0.5140 | 0.3390 |
| | TACO | 0.1425 | 30.79 | 0.9555 | **0.0394** | 0.0771 | 19.42 | 4.17 | 4.4278 | 58.95 | 0.5419 | 0.3168 |
| | Re$^2$IC | 0.1500 | 28.53 | 0.9422 | 0.0754 | **0.0381** | **11.82** | **1.41** | 4.4016 | 57.24 | **0.5509** | 0.3429 |
| High | HiFiC | 0.2720 | 31.84 | **0.9774** | 0.0380 | 0.0492 | 12.74 | 1.40 | **3.4455** | 60.23 | 0.5142 | 0.3406 |
| | CDC | 0.2439 | 27.51 | 0.9522 | 0.0836 | 0.0508 | 13.12 | 1.84 | 4.0203 | 59.78 | 0.4528 | 0.3453 |
| | TACO | 0.2390 | **33.45** | 0.9756 | **0.0351** | 0.0756 | 18.10 | 3.71 | 4.1096 | **60.73** | 0.5409 | 0.3379 |
| | Re$^2$IC | 0.2970 | 30.41 | 0.9636 | 0.0483 | **0.0187** | **7.10** | **0.80** | 4.1808 | 58.89 | **0.5492** | **0.3486** |

# F    ADDITIONAL EXPERIMENTS AND VISUALIZATIONS

## F.1    ADDITIONAL PERCEPTUAL-CODEC BASELINES

This section presents additional diffusion-based perceptual codecs, with results reported in Tables 6 and 7. Here, we focus solely on perceptual performance and intentionally ignore computational complexity to provide a broader comparison. For fairness, we report results at the closest available bitrate to our target operating points.

Our comparison focuses on low-complexity implicit codecs, whose rate–perception–complexity characteristics differ fundamentally from diffusion- or GAN-based methods. Although diffusion models achieve high perceptual scores on metrics like MS-SSIM or LPIPS, Re2IC delivers competitive or even superior FID/DISTS/KID/NIQE/CLIP-IQA/MANIQA performance with far lower complexity, highlighting the effectiveness of implicit codecs for perceptual modeling.

Table 7: Experiments with more diffusion baselines, TACO (Lee et al., 2024) on Kodak, where we select the baseline with the most relevant rate regimes. **Red** and blue indicate the best and second-best results, respectively. HiFiC and TACO are excluded from the high-BPP regime comparisons since their rates are significantly higher or lower. Re$^2$IC consistently delivers **state-of-the-art or leading** performance across FID, DISTS, NIQE, ClipIQA, and MANIQA, even compared with the current strongest diffusion model, TACO, at the target medium rate.

| | Model | BPP↓ | PSNR↑ | MS-SSIM↑ | LPIPS↓ | DISTS↓ | FID↓ | NIQE↓ | MUSIQ↑ | ClipIQA↑ | MANIQA↑ |
|---|---|---|---|---|---|---|---|---|---|---|---|
| | VTM | 0.182 | **30.10** | **0.9520** | 0.2538 | 0.1767 | 147.87 | 5.1808 | 70.68 | 0.4827 | 0.4084 |
| | TACO | 0.183 | 28.78 | 0.9399 | **0.0390** | 0.0800 | 41.05 | 4.2099 | 72.7770 | **0.6989** | 0.4028 |
| Medium | C3/WDs | 0.183 | 26.23 | 0.9324 | 0.1150 | 0.0786 | 49.87 | 5.5448 | 68.49 | 0.5062 | 0.3899 |
| | HiFiC | 0.183 | 27.56 | 0.9456 | 0.0665 | 0.0889 | 54.89 | **2.7894** | 73.76 | 0.6562 | **0.4576** |
| | Re$^2$IC | 0.183 | 25.65 | 0.9006 | 0.1017 | **0.0572** | 37.05 | 4.0302 | 71.86 | 0.6149 | 0.4388 |
| | VTM | 0.287 | **31.84** | **0.9694** | 0.1864 | 0.1416 | 106.87 | 4.6125 | 73.13 | 0.5623 | **0.4586** |
| High | TACO | 0.226 | 30.08 | 0.9555 | **0.0367** | 0.0746 | 39.65 | 3.9355 | 73.66 | **0.6835** | 0.4213 |
| | C3/WDs | 0.275 | 27.79 | 0.9529 | 0.0802 | 0.0537 | 35.39 | 4.8479 | 70.15 | 0.5491 | 0.4085 |
| | Re$^2$IC | 0.275 | 26.74 | 0.9231 | 0.0764 | **0.0374** | 26.09 | 3.8981 | 72.27 | 0.6446 | 0.4446 |

## F.2 THEORETICAL COMPLEXITY AND PRACTICAL CODING LATENCY

We present the detailed latency and complexity for decoding in Table 8 and encoding in Table 9. Since our implementation is purely research-oriented and unoptimized, substantial further speed-ups are expected (Blard et al., 2024).

Table 8: Decoding cost breakdown of Re$^2$IC on Kodak dataset across different region numbers, using an NVIDIA RTX 3090 GPU and Intel Core i9-10980XE CPU @ 3.00GHz. Note that all results are reported in **CPU computation**. We can see that the complexity is not significantly increased with the increase in region numbers.

| VTM-19.1 | 293.05 ms (CPU) | | | | | |
|---|---|---|---|---|---|---|
| EVC (S/M/L) | 19.89/24.12/31.91 ms (GPU) | | | | | |
| MLIC$^{++}$ | 364.06 ms (GPU) | | | | | |
| C3/WDs | 270.55 ms (CPU), with decoding complexity: 2626∼ 2925 MACs/pixel | | | | | |
| Re$^2$IC | Chain coding: 12.86 ms (CPU) | | | | | |
| | Module | ARM | GPM | LPN1 | LPN2 | LPN3 | Total |
| ($N = 1$ regions) | CPU decoding time (ms) | 204.27(55.3%) | 155.27(42.0%) | 10.12(2.7%) | – | – | 369.66 |
| | Decoding complexity (MACs/pixel) | 1600 | 716 | 33 | – | – | 2349 |
| ($N = 2$ regions) | CPU decoding time (ms) | 197.91(49.5%) | 181.77(45.5%) | 9.64(2.4%) | 10.45(2.6%) | – | 399.77 |
| | Decoding complexity (MACs/pixel) | 1600 | 959 | 33 | 33 | – | 2625 |
| ($N = 3$ regions) | CPU decoding time (ms) | 199.70(43.8%) | 227.15(49.8%) | 9.43(2.0%) | 10.08(2.2%) | 9.61(2.1%) | 455.97 |
| | Decoding complexity (MACs/pixel) | 1600 | 1202 | 33 | 33 | 33 | 2901 |

Table 9: Encoding cost breakdown of Re$^2$IC on Kodak dataset across different region numbers, using an NVIDIA RTX 3090 GPU and Intel Core i9-10980XE CPU @ 3.00GHz. We can see that the complexity is not significantly increased with the increase in region numbers. Encoding complexity can be approximated as three times the decoding cost. Note that here, the complexity of the feature backbone can vary, as it depends on the backbone and the slice number, where the latency is reported with VGG for both C3/WDs and Re$^2$IC.

| VTM-19.1 | 87.13 s (CPU) | | | | | | |
|---|---|---|---|---|---|---|---|
| EVC (S/M/L) | 22.43/36.19/47.87 ms (GPU) | | | | | | |
| MLIC$^{++}$ | 271.91 ms (GPU) | | | | | | |
| C3/WDs | 70.24s/1k steps (GPU) | | | | | | |
| Re$^2$IC (1k) | EML-Net partition time: ≃ 50 ms (GPU), Chain coding: 119.58 ms (CPU) | | | | | | |
| | Module | ARM | GPM | LPN1 | LPN2 | LPN3 | Feature backbone | Total |
| ($N = 1$ regions) | GPU encoding time (s/1k) | 6.84(6.9%) | 8.11(8.2%) | 1.93(2%) | – | – | 82.07(82.9%) | 98.95 |
| | Encoding complexity (kMACs/pixel) | 4800 | 2148 | 99 | – | – | – | – |
| ($N = 2$ regions) | GPU encoding time (s/1k) | 7.45(7.2%) | 12.14(11.7%) | 1.33(1.3%) | 1.45(1.4%) | – | 80.99(78.4%) | 103.36 |
| | Encoding complexity (kMACs/pixel) | 4800 | 2877 | 99 | 99 | – | – | – |
| ($N = 3$ regions) | GPU encoding time (s/1k) | 7.09(6.6%) | 15.10(14.1%) | 0.81(0.8%) | 1.12(1.0%) | 1.06(1.0%) | 82.11(76.5%) | 107.29 |
| | Encoding complexity (kMACs/pixel) | 4800 | 3606 | 99 | 99 | 99 | - | – |

## F.3 MORE EXPERIMENTS AND ABLATION STUDY

This section provides more quantitative results in Figures 13, 14.

To validate the loss-function design, we first remove the DWT during optimization and observe a substantial degradation in RP performance for both C3 and $Re^2IC$ (see Table 11), confirming the effectiveness of our WA-WD formulation.

Table 10: Ablation study of different mechanisms for the RP performance. Higher DISTS, FID, and LPIPS indicate worse RP performance (shown in darker blue).

| Model Variant | DISTS ↓ vs. FID ↓ vs. LPIPS ↓ | | |
|---|---|---|---|
| | bpp≈0.113 | bpp≈0.183 | bpp≈0.275 |
| Proposed $Re^2IC$ scheme | 0.086/58.21/0.1432 | 0.0572/37.05/0.1017 | 0.037/26.09/0.0764 |
| ⇒ w/o wavelet transform | 0.104/67.10/0.1547 | 0.066/41.52/0.1044 | 0.042/28.06/0.0765 |
| ⇒ w/o region partition | 0.117/76.58/0.1732 | 0.076/49.37/0.1167 | 0.053/33.45/0.0817 |
| ⇒ w/o LPN | 0.115/74.93/0.1690 | 0.077/49.39/0.1136 | 0.053/34.07/0.0796 |
| ⇒ w/o GPM | 0.131/86.77/0.1941 | 0.087/53.62/0.1303 | 0.057/34.17/0.0952 |
| ⇒ w/o common randomness | 0.151/124.7/0.2613 | 0.110/75.55/0.1852 | 0.081/48.92/0.1285 |

Table 11: Ablation study for the role of WA-WD as a loss function across different implicit codecs, where the red background indicates better performance.

| Model Variant | DISTS ↓ vs. FID ↓ vs. LPIPS ↓ | | |
|---|---|---|---|
| | bpp≈0.113 | bpp≈0.183 | bpp≈0.275 |
| $Re^2IC$ / WDs | 0.104/67.10/0.1547 | 0.066/41.52/0.1044 | 0.042/28.06/0.0765 |
| ⇒ WA-WD | 0.086/58.21/0.1432 | 0.057/37.05/0.1017 | 0.037/26.09/0.0764 |
| C3 / WDs | 0.118/72.83/0.1717 | 0.079/49.87/0.1150 | 0.054/35.39/0.0802 |
| ⇒ WA-WD | 0.090/62.30/0.1514 | 0.059/40.14/0.1047 | 0.041/28.51/0.0783 |

We then assess the contribution of each mechanism by ablating components one at a time and comparing their performance against the full $Re^2IC$ model, with results reported in Table 10.

Removing the LPN and region-partition strategy causes the expected similar degradation, discarding LPNs and relying solely on a GPM essentially reduces the model to single-region overfitting. Eliminating the GPM results in a pure dual-overfitting setup similar to C3, yielding an even larger drop in performance, which highlights the importance of local–global perceptual modeling. Together, these ablations validate the effectiveness of our region-based perceptual modeling and demonstrate that each component contributes meaningfully to the overall performance of $Re^2IC$. Interestingly, we find that common randomness (CR) plays a critical role in improving RP performance for $Re^2IC$, consistent with observations in (Ballé et al., 2025). In $Re^2IC$, CR becomes even more influential because it allows more flexible utilization across both spatial and frequency dimensions.

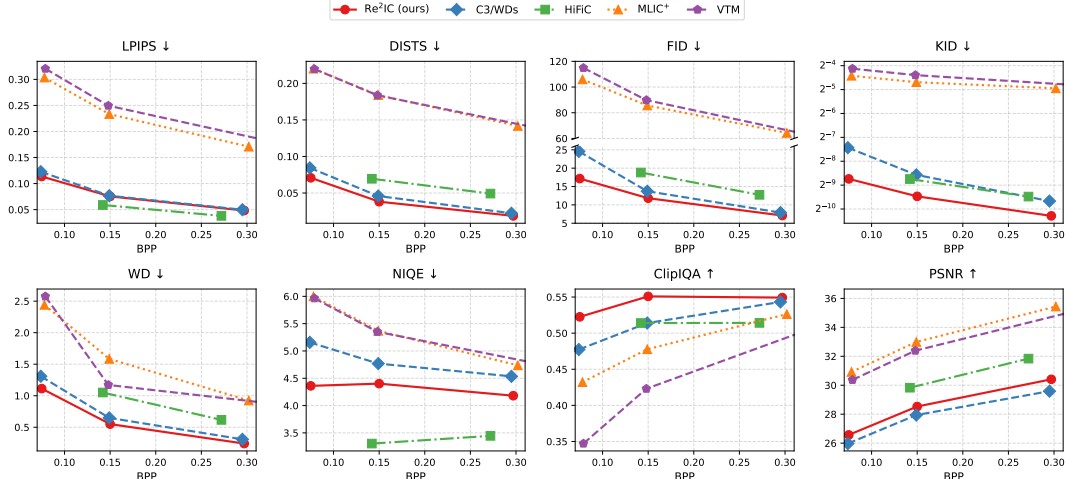

Figure 13: Rate-distortion and -perception curves on CLIC2020. Arrows in the title indicate whether lower is better (↓), or higher is better (↑).

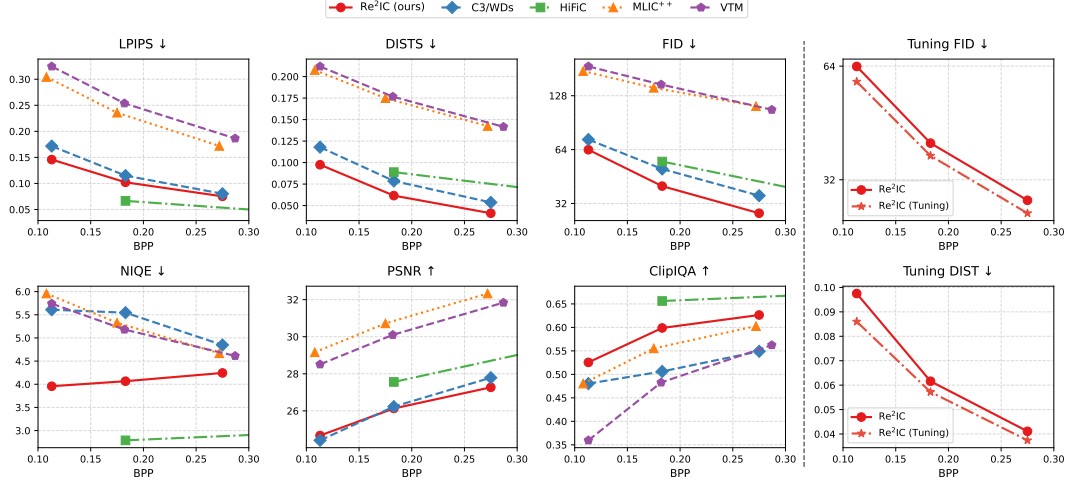

Figure 14: Rate-distortion and -perception curves on Kodak. Arrows in the title indicate whether lower is better (↓), or higher is better (↑).

### F.4   TECHNICAL COMPARISON FOR DWT

This section provide additional analysis of DWT in the context of $Re^2IC$.

**Comparison with other DWT-based codecs.**    The use of wavelet transforms for frequency-aware optimization has been explored in prior works (Fu et al., 2024; Yin et al., 2025; Ma et al., 2019). However, their roles, objectives, and technical pathways for $Re^2IC$ and these DWT-based codecs are fundamentally different.

- **Different compression paradigms with DWT.**

  Prior wavelet-based codecs (e.g., WeConvene) use the DWT as an **analysis transform** to decorrelate latent features, produce sparser subbands, and improve entropy coding efficiency. **Their goal is to obtain better distribution learning for the entropy model.**

  In contrast, Re2IC is an implicit codec. It performs **NO** transform coding, and does **NOT** learn a distribution or introduce feature extraction. Instead, it overfits each individual image and **directly models human perception** on a per-sample basis. Thus, DWT in Re2IC is not

used to improve compression of latent codes, but to improve the **perceptual optimization metric itself**.

- **Different purpose in Re2IC.**
  In Re2IC, DWT is introduced to:
  - **Improve the statistical validity of WD approximation.** DWT reduces cross-correlations and yields a tighter diagonal-Gaussian approximation, making WD approximation more stable and accurate.
  - **Provide a tunable, frequency-aware perceptual metric.** WA-WD enables a tunable perception metric and allows flexible balancing of fidelity and realism by adjusting subband weights or $\sigma$-maps.
  - **Enable joint spatial–frequency perception modeling.** Combined with our region-wise overfitting scheme, WA-WD explicitly models perception along both spatial saliency and frequency structure, which is essential for realism-enhanced implicit codecs.

- **Different conceptual motivation.**
  Our use of wavelets is driven by HVS principles: WD's foveal/peripheral pooling alone cannot capture frequency-selective perceptual cues. WA-WD integrates wavelet-based frequency structure to jointly model realism and fidelity, something transform-based codecs do not aim to address.

Thus, although both methods use DWT, Re2IC is fundamentally different, which targets introducing a wavelet-domain perceptual distortion metric specifically designed for implicit codecs.

**Effect of joint optimized entropy models.** Our method adopts a joint entropy model across regions, coupled with a local–global perception modulation mechanism. This section analyzes how different entropy–modeling strategies affect performance. For fairness, we consider 2-region scenarios (with more regions, our benefits are expected to be larger due to the significantly increased bpp costs for entropy models), and match the overall parameter budget across settings (joint entropy: two models with dimensions $16 \times 16$, $16 \times 16$, $16 \times 2$; separate entropy: $24 \times 24$, $24 \times 24$, $24 \times 2$). As shown in Table 12, the joint entropy model achieves better performance, validating our design.

Table 12: Ablation study for different settings of entropy models for Kodak images.

| Model Variant | PSNR (dB) ↑ / bpp ↓ | | | |
|---|---|---|---|---|
| Re$^2$IC / joint entropy model | 29.04/0.156 | 32.57/0.361 | 34.59/0.531 | 39.30/1.11 |
| $\Rightarrow$ separate entropy models | 29.04/0.158 | 32.54/0.364 | 34.58/0.535 | 39.24/1.15 |

Table 13: Ablation study for RD performance across different methods on Kodak dataset.

| Model | PSNR (dB) ↑ / bpp ↓ | | | | | | BD-rate over VTM-19.1 % ↓ |
|---|---|---|---|---|---|---|---|
| WeConvene | 30.07/0.163 | 30.88/0.203 | 32.53/0.309 | 34.29/0.452 | 36.04/0.637 | 37.96/0.897 | −8.37 |
| Re$^2$IC / MSE | 29.12/0.151 | 32.61/0.356 | 34.60/0.525 | 36.65/0.743 | 39.33/1.107 | 41.26/1.448 | −1.45 |

**Visualization of Feature Correlations.** To validate that DWT improves the independent-Gaussian assumption underlying WA-WD, we visualize feature correlations before and after wavelet decomposition. As shown in Fig. 15, increasing the number of DWT levels progressively reduces inter-band correlations, producing more independent coefficients and better supporting the diagonal-Gaussian approximation used in WA-WD.

**Frequency-domain optimization objectives.**

- First, we compare RD performance across two DWT-based methods, and the results are presented in Table 13. We observe that although WeConvene achieves stronger RD performance, it does so at the cost of significantly higher decoding complexity. In contrast,

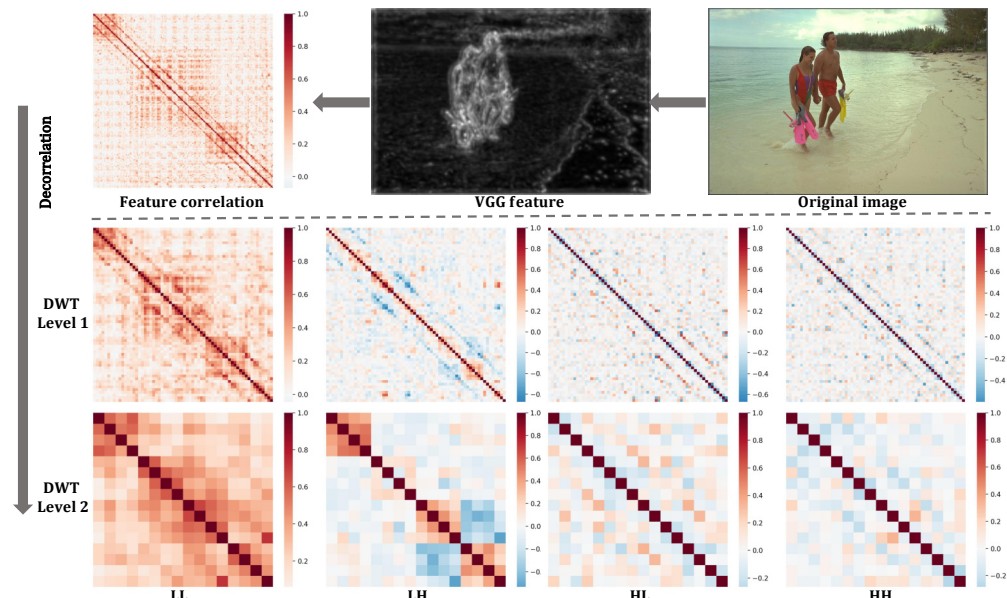

Figure 15: Visualization of feature correlations across wavelet subbands. Increasing DWT levels progressively reduces inter-band correlations, yielding more independent coefficients and better supporting the diagonal-Gaussian approximation used in WA-WD.

Re$^2$IC delivers competitive RD results, achieving a 6.92% BD-rate reduction relative to VTM-19.1, while maintaining substantially lower complexity.

- Then, we analyze the RP performance across two DWT methods. As shown in Figs. 16 and 17, Re$^2$IC significantly improves the perception performance under lower decoding complexity and lower bpp cost.

- Last, we conduct an ablation study over each sub-band for the WA-WD optimization. Figs. 18 and 19 show tunable weights can enhance the RP performance, Table 2 quantifies the performance gain from tuning sub-band weights.

**Wavelet decomposition levels.** Using more wavelet levels can, in principle, yield more independent features and more accurate WA-WD estimation (as illustrated in Fig. 15). However, in practice, higher levels cause VGG-downsampled features to become too small for stable WD estimation (five VGG stages combined with 2–3 DWT levels reduce resolution by factors of 128–256 $\times$), which also introduces higher computational cost. Although alternative backbones or scale settings could mitigate this, changing the backbone would break fairness with baselines such as C3/WDs. Since a single DWT level already provides strong performance with low overhead, we adopt 1-level DWT in this work and leave multi-level extensions for future study.

**Effect of each sub-band** Figs. 18 and 19 present ablations over different subband configurations. We compare using equal subband weights versus learnable tuned weights, and observe that tuned weights yield significantly better RP performance, confirming the benefit of subband-aware optimization. Removing the LL subband leads to severe degradation: fundamental structures such as lighting, shading, and global geometry collapse, indicating that LL is essential for maintaining fidelity. Removing only LH, HL, or HH subbands can produce acceptable reconstructions but noticeably reduces high-frequency details, consistent with prior findings in (Deng et al., 2019), confirming that high-frequency subbands primarily govern fine textures.

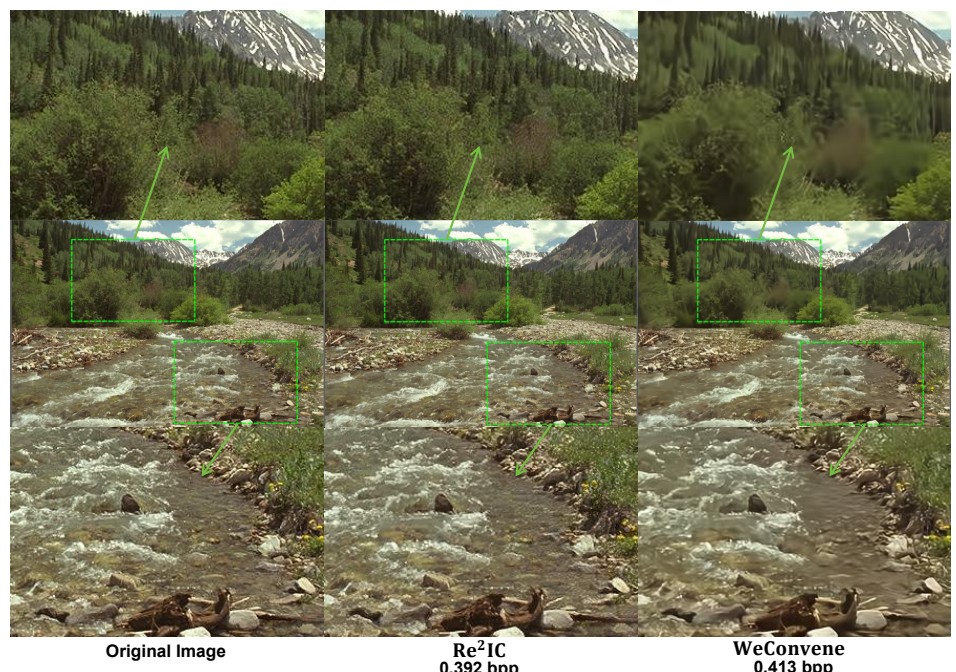

Figure 16: Visualizations for different DWT-based codecs.

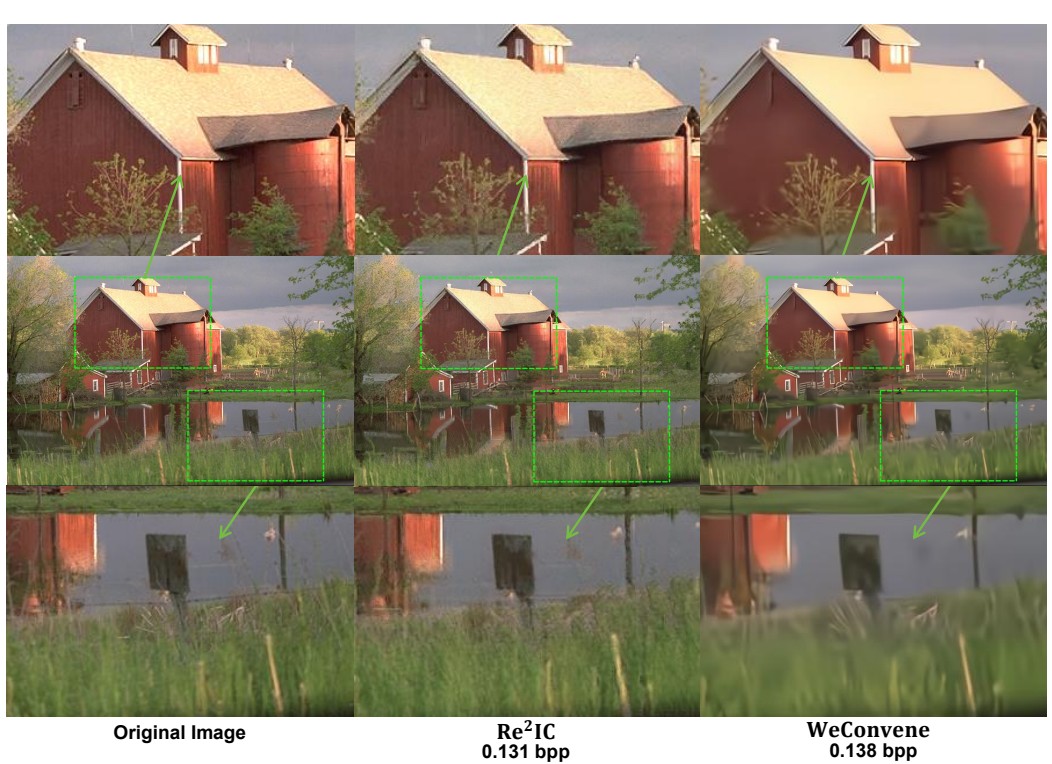

Figure 17: Visualizations for different DWT-based codecs.

### F.5 LATENT BIT COST AND VISUALIZATION

We further visualize the latent vectors in Fig. 6 and their bit allocation in Fig. 20.

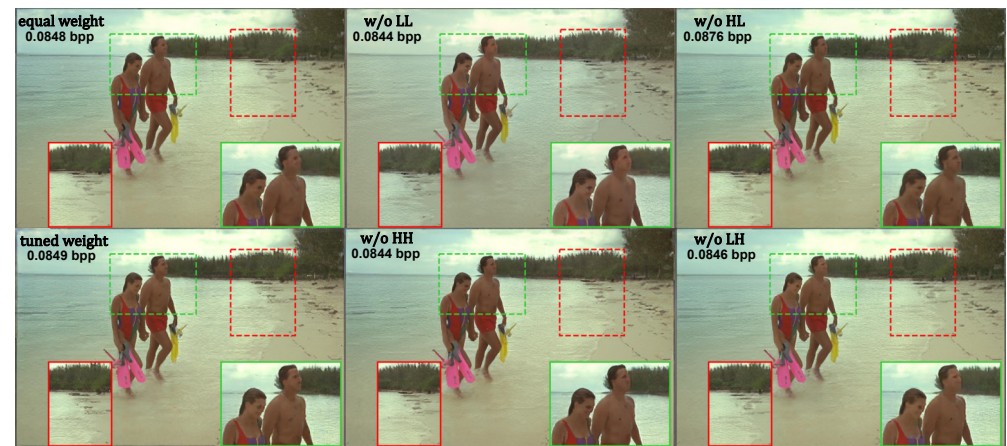

Figure 18: Ablation study on WA-WD subbands for kodim 12, where we can observe that learned subband weights outperform equal weighting, yielding higher RP performance. Excluding the LL band causes major degradation in global structure and illumination, whereas excluding LH/HL/HH preserves overall fidelity but reduces high-frequency textures.

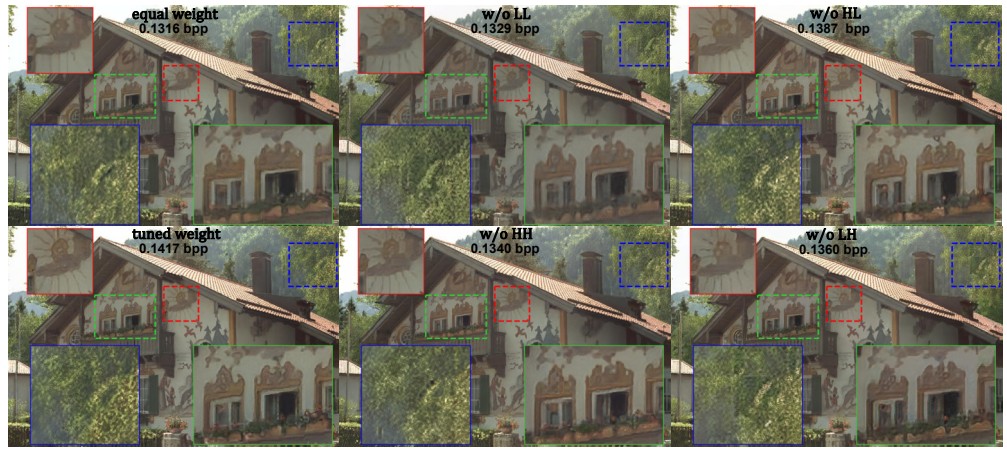

Figure 19: Ablation study on WA-WD subbands for kodim 24, where we can observe that learned subband weights outperform equal weighting, yielding higher RP performance. Excluding the LL band causes major degradation in global structure and illumination, whereas excluding LH/HL/HH preserves overall fidelity but reduces high-frequency textures.

### F.6 ARTIFACTS FROM CONTOUR COMPRESSION

Artifact effects of contour compression is provided in Fig. 21

### F.7 FAST CONVERGENCE

RD and RP convergence is ploted in Figs. 8 and 22.

### F.8 KODAK DATASET

For the Kodak dataset, more visualizations are provided as Figs. 23 25 26 27 28.

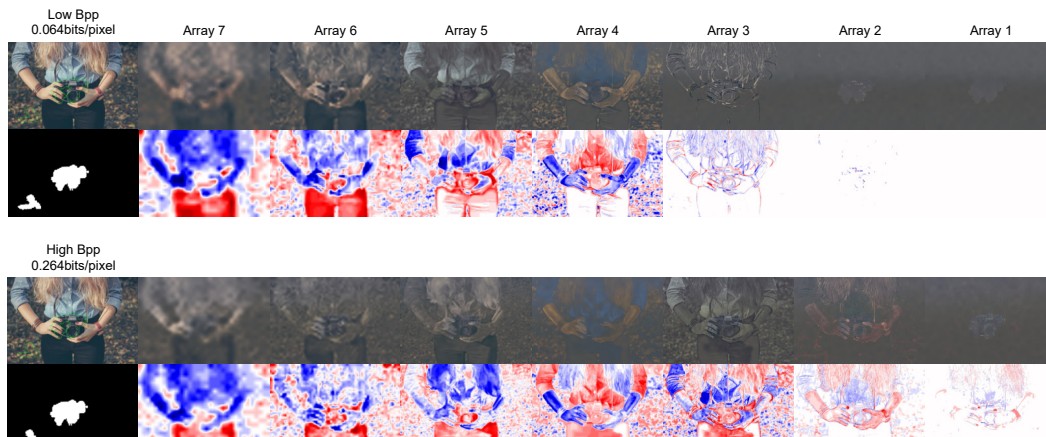

Figure 20: Visualization of the latent vectors (two salient regions as an example), where latent vectors are arranged from low to high resolution. The first row shows reconstructions where all but the array are set to zero. The second row displays the raw arrays, upscaled to match the output resolution. At low bit rates, Re$^2$IC captures rich low-frequency features into low-resolution arrays, then progressively adds high-frequency details with resolution, yielding a more balanced bit allocation across arrays and stronger cross-frequency interaction than prior overfitted codecs, which typically ignore the effect of the lowest resolution arrays/latent vectors, as visualized in (Kim et al., 2024; Wu et al., 2025).

### F.9 CLIC2020 DATASET

For the CLIC2020 dataset, more visualizations are provided as Figs. 29, 30, 31, 32. Since detailed rates of some CLIC2020 baselines are not released, we report results using images from the same rate regime and provide visual comparison in supplementary materials.

## G LIMITATIONS AND FUTURE WORK.

The low decoding cost of Re$^2$IC is particularly beneficial in multi-user streaming scenarios, where encoding is performed once offline and many users decode the same content. This remains an inherent limitation of all overfitted codecs, including C3, Cool-Chic, and LotteryCodec. At the same time, our region-wise perception modeling strategy offers a promising direction for accelerating encoding: by decomposing the image into more homogeneous regions, optimization becomes significantly faster and more stable. This can be further improved through complementary techniques such as meta-learning or mixed-precision training, which we expect will substantially reduce encoding time in future work.

While Re$^2$IC accelerates convergence, its reliance on a VGG backbone still raises encoding complexity, making it best suited for one-time encoding with large-scale reuse in perception-critical scenarios (e.g., streaming, cloud gaming, VR). Future work includes developing more efficient perception-tailored backbones to reduce encoding cost, and extending the framework to video with motion- and attention-guided region partitioning.

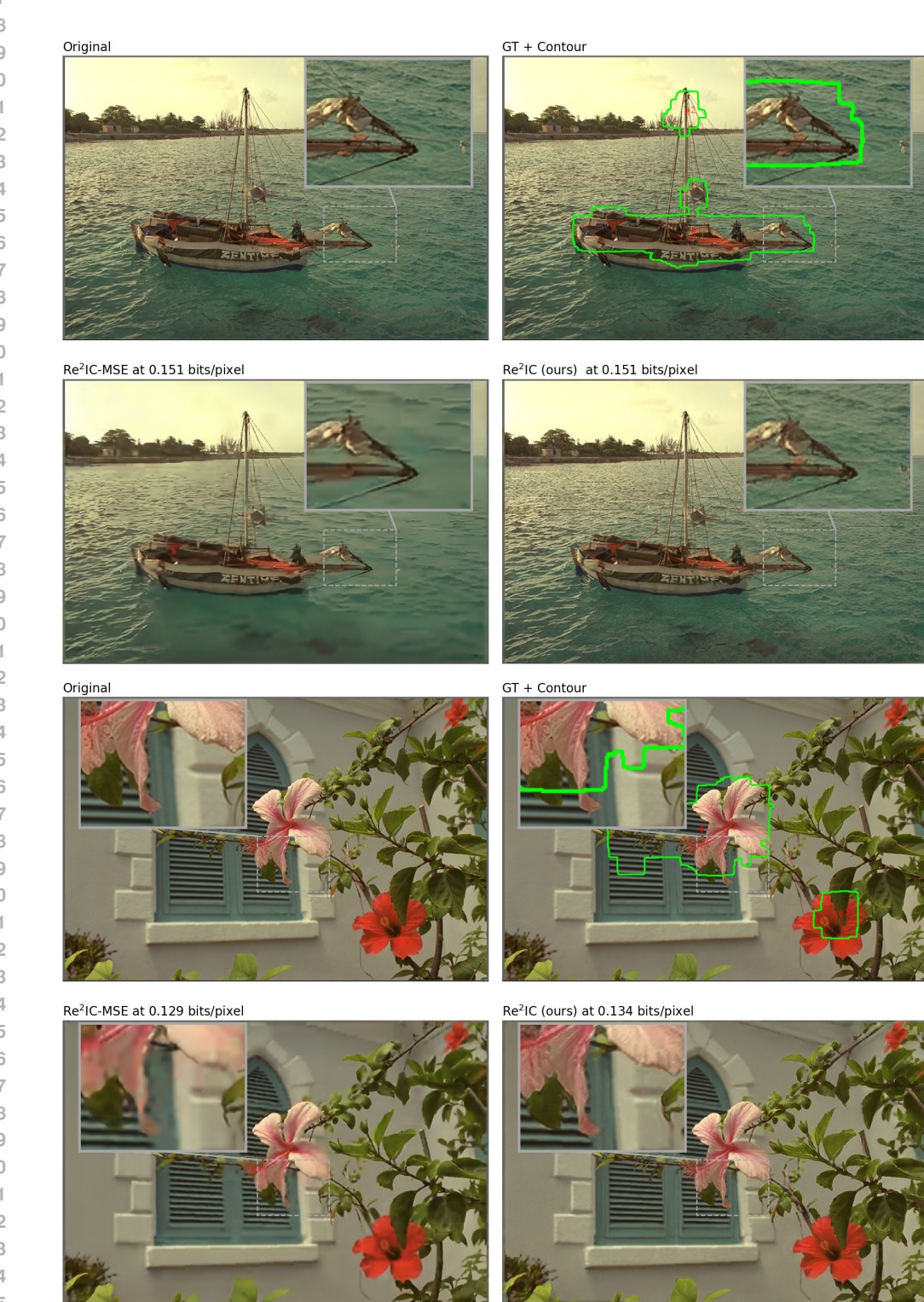

Figure 21: Effect of inaccurate contours (green indicates the region for local-fitting): with MSE optimization, artifacts appear around decoded contours where different LPNs meet. In contrast, Re$^2$IC, aided by perceptual modeling and GPM, eliminates these artifacts and achieves perceptual realism closer to the original.

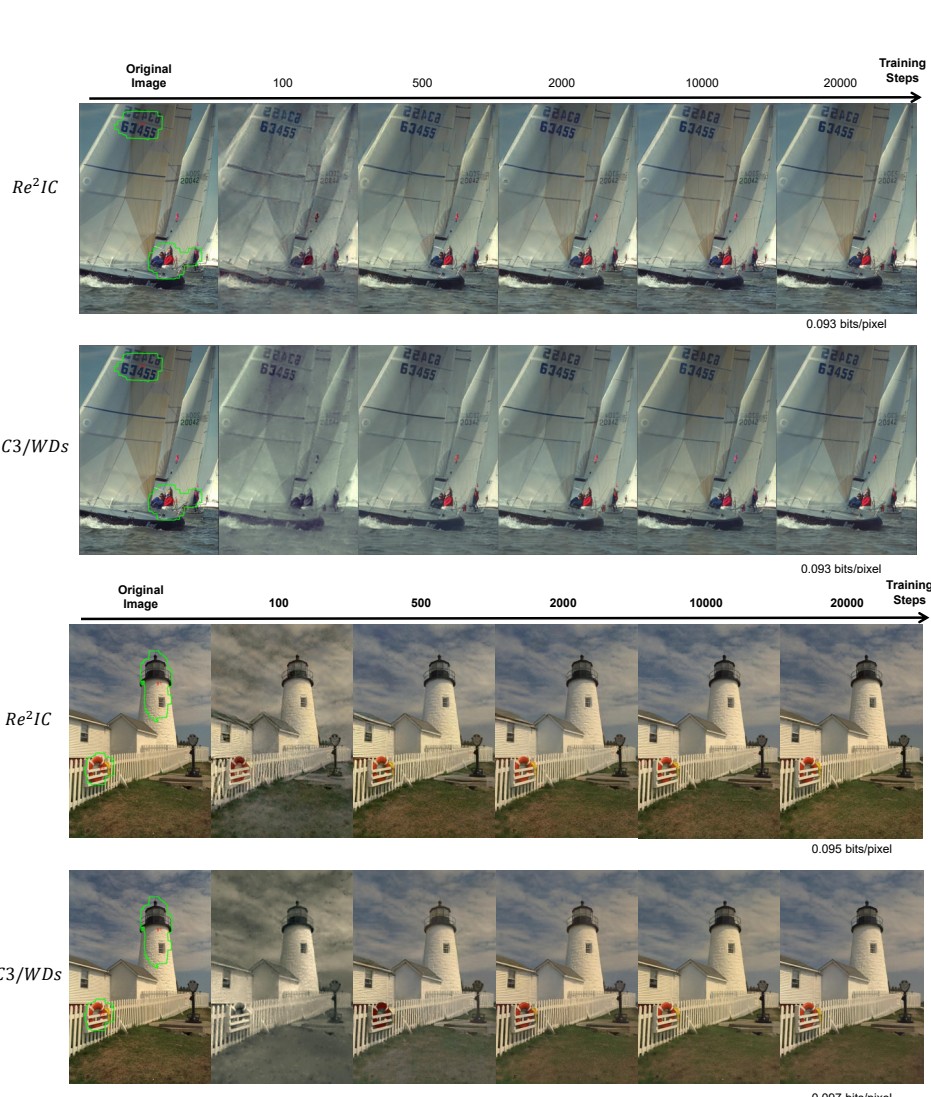

Figure 22: RP convergence comparisons (green indicates the partitioned regions): Re$^2$IC converges faster by first fitting salient regions and then refining others.

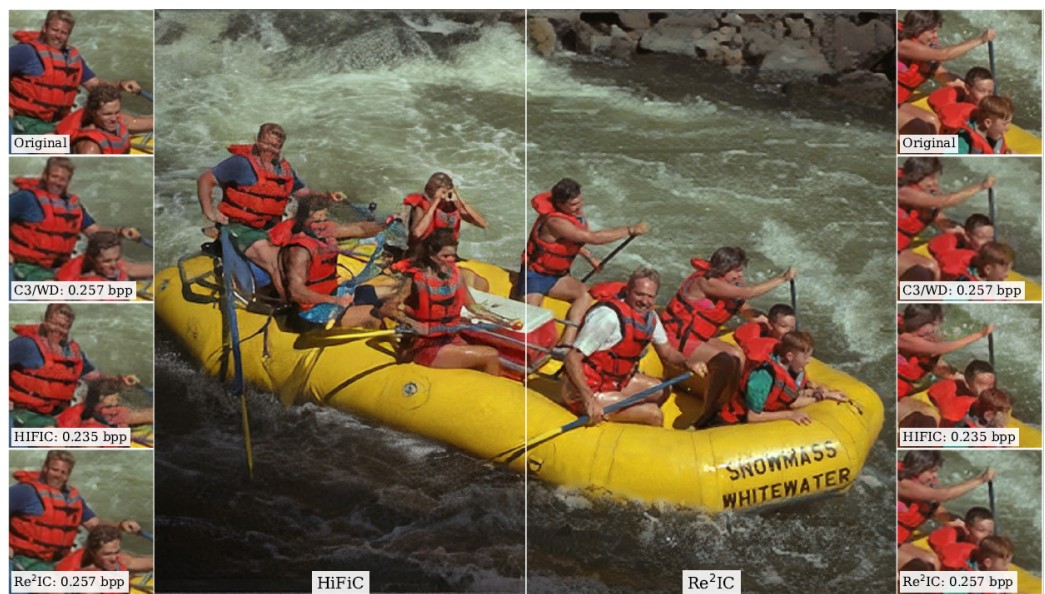

Figure 23: More visual comparison between Re$^2$IC, HiFiC, and C3/WDs, where Re$^2$IC can deliver the best perceptual realism, closely matching the original image.

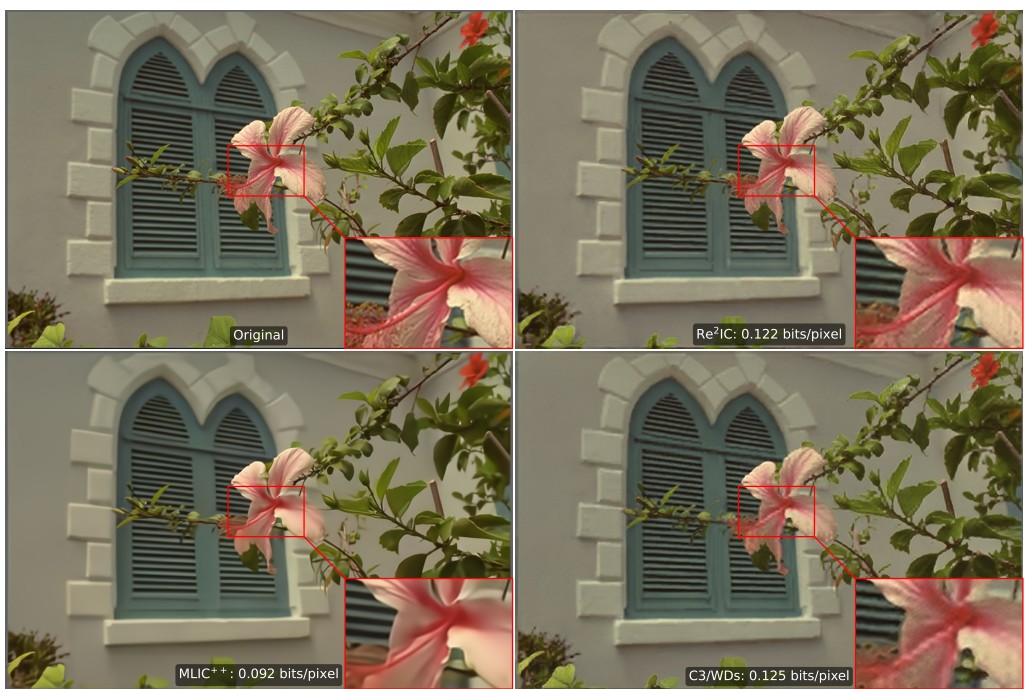

Figure 24: More visual comparison between Re$^2$IC, HiFiC, and C3/WDs, where Re$^2$IC can deliver the best perceptual realism, closely matching the original image.

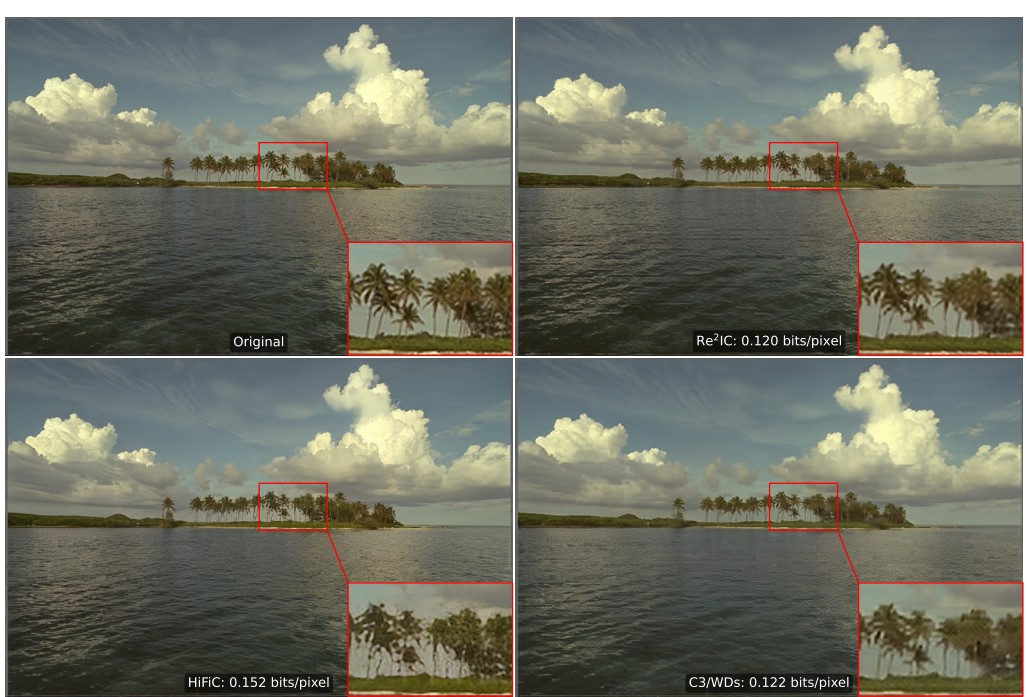

Figure 25: More visual comparison between Re$^2$IC, HiFiC, and C3/WDs, where Re$^2$IC can deliver the best perceptual realism, closely matching the original image.

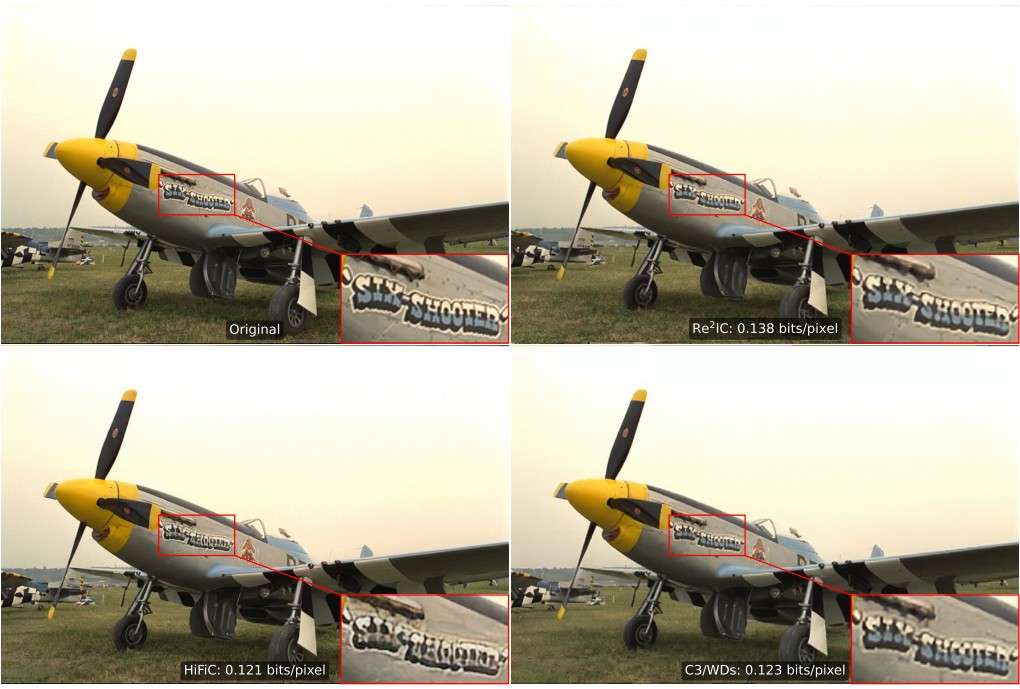

Figure 26: More visual comparison between Re$^2$IC, HiFiC, and C3/WDs, where Re$^2$IC can deliver the best perceptual realism, closely matching the original image.

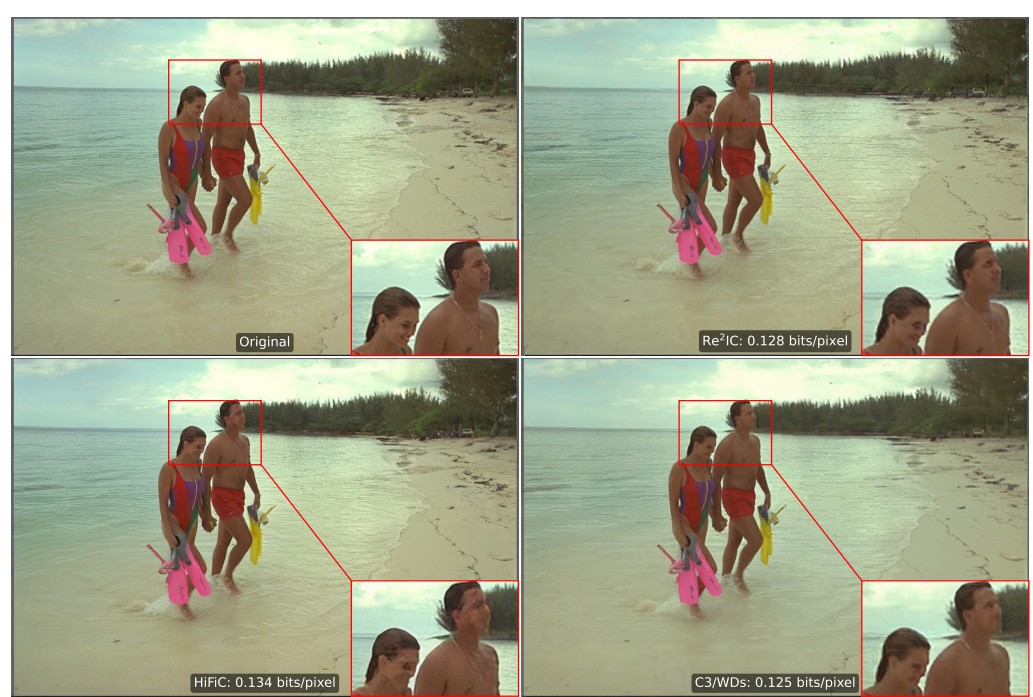

Figure 27: More visual comparison between Re$^2$IC, HiFiC, and C3/WDs, where Re$^2$IC can deliver the best perceptual realism, closely matching the original image.

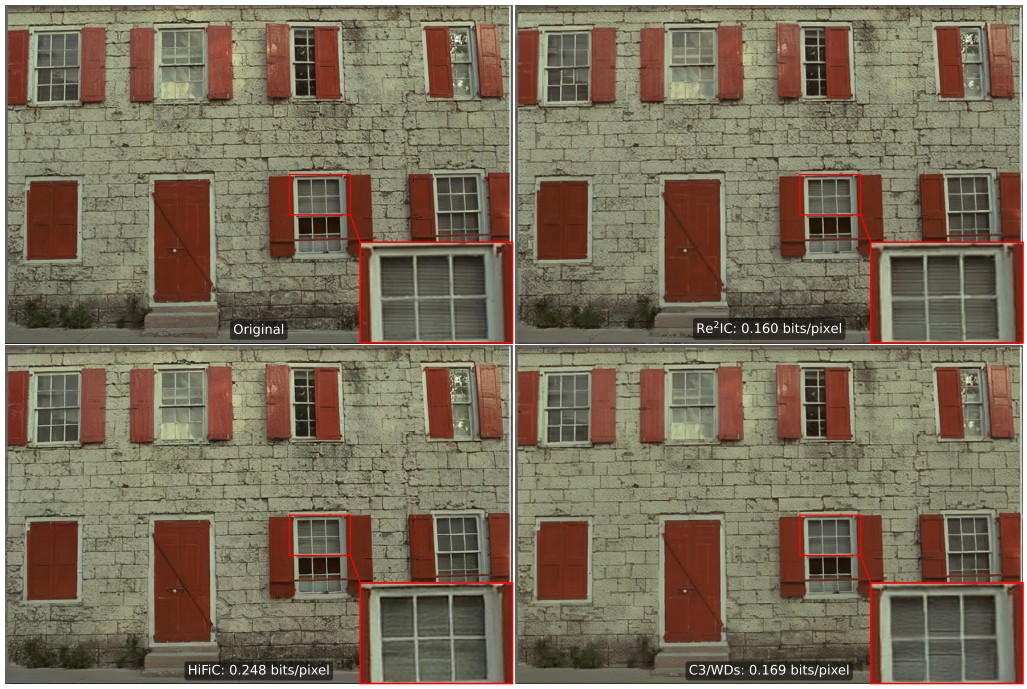

Figure 28: More visual comparison between Re$^2$IC, HiFiC, and C3/WDs, where Re$^2$IC can deliver the best perceptual realism, closely matching the original image.

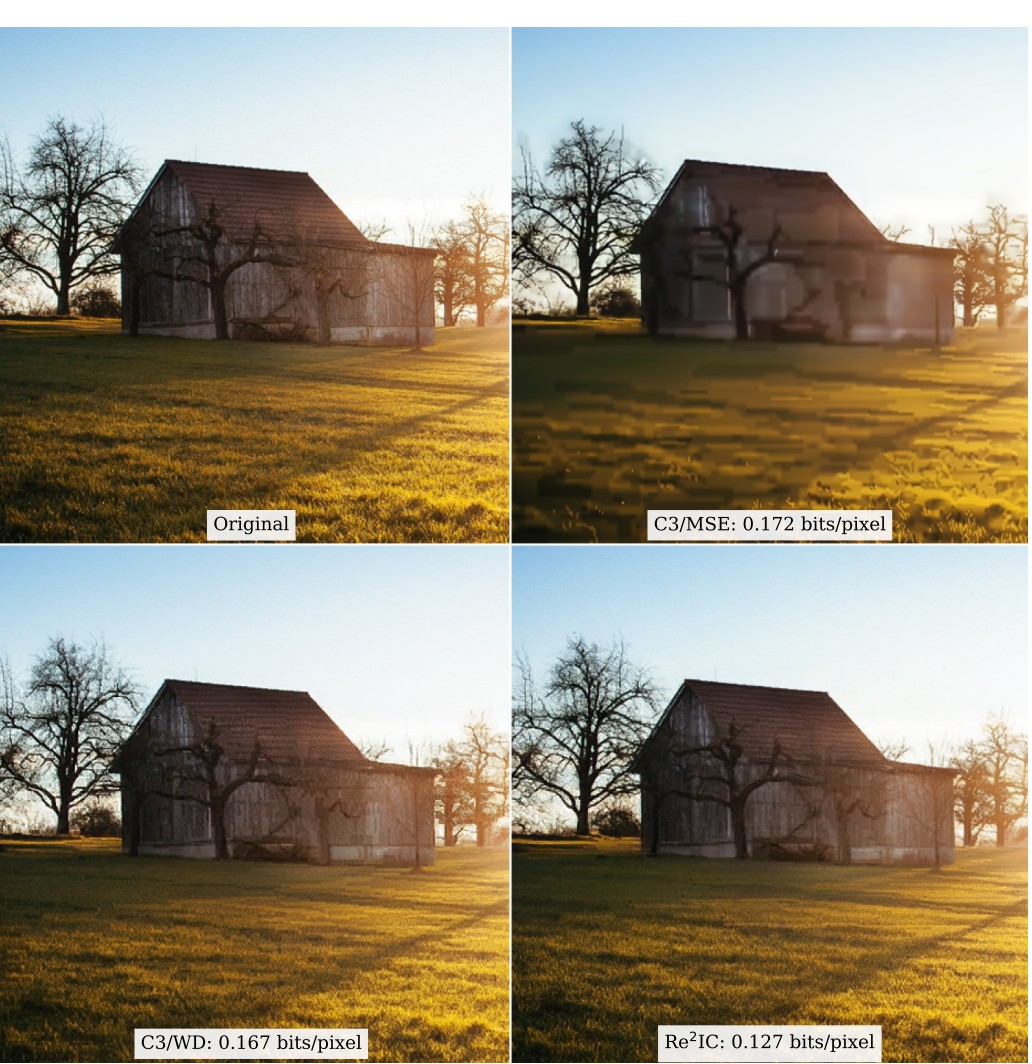

Figure 29: Visual comparisons of image 28 from CLIC2020 dataset (best viewed on screen). The MSE-optimized overfitted codec produces flattened textures due to fidelity-focused optimization, while C3/WD can introduce artifacts around trees and wood patterns. In contrast, Re$^2$IC delivers reconstructions with fewer artifacts at a lower bitrate (0.127 bpp), with reduced artifacts.

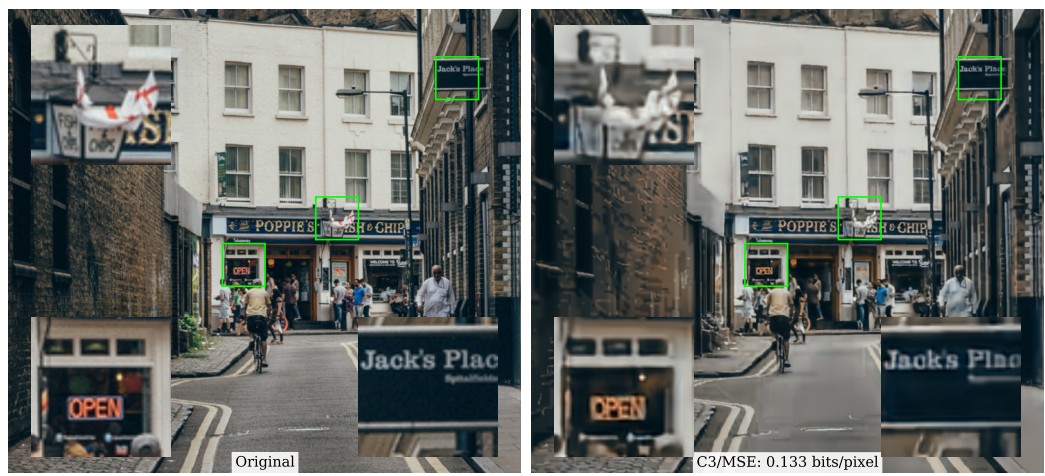

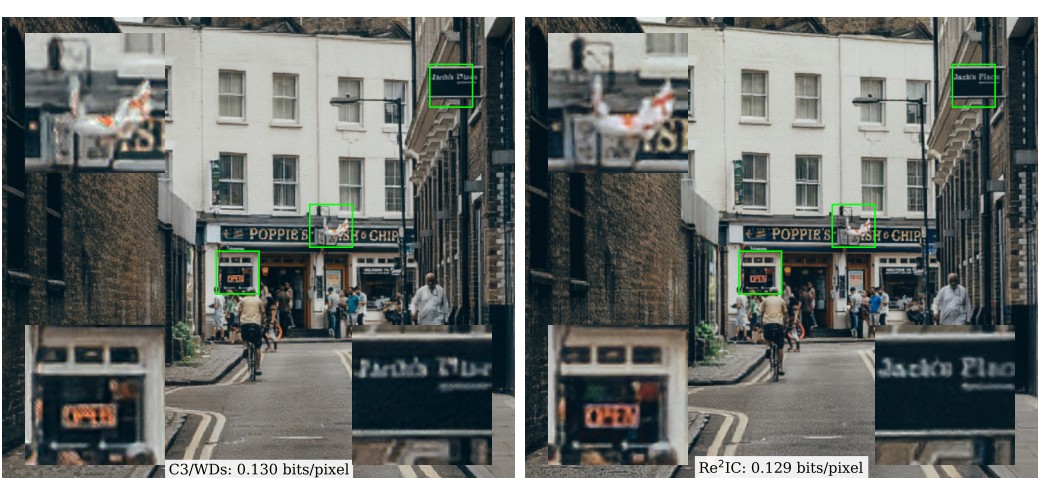

Figure 30: Visual comparisons of image 11 from CLIC2020 dataset (best viewed on screen). Similar trends are observed: MSE-optimized codec tends to smooth textures (e.g., roads, walls), while C3/WDs introduces artifacts in non-salient areas (e.g., flags) and loses fidelity in high-frequency content (e.g., text), often reducing realism. Our Re$^2$IC balances both sides.

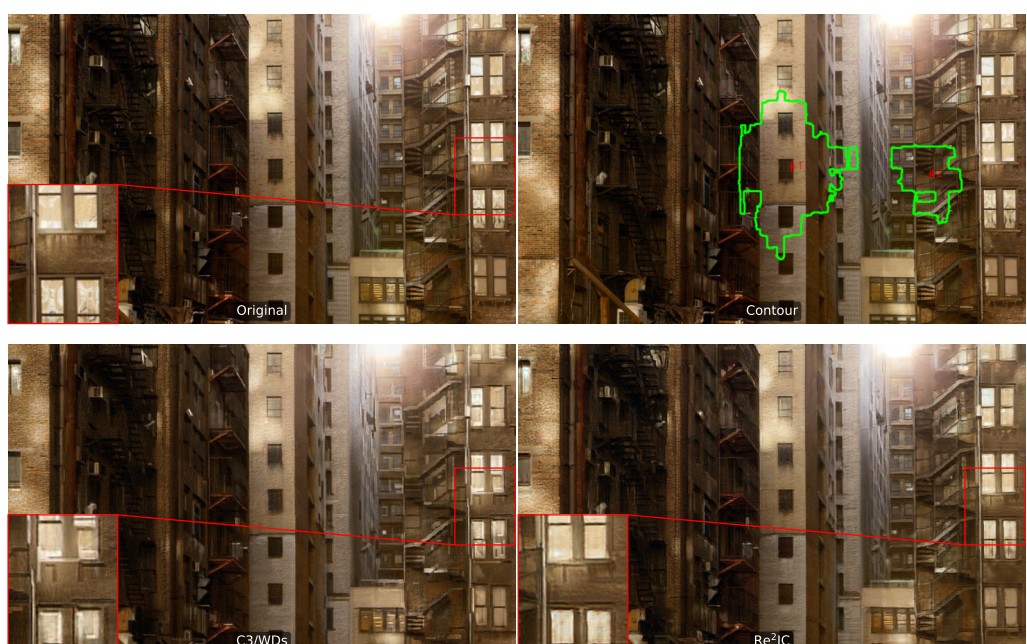

Figure 31: Visual compassion for methods at low-bpp regime ($<= 0.075$). We can see that C3/WDs tends to resample the low-frequency information in non-salient areas. This is because it assigns a larger $\sigma$ to non-salient areas without further considering the internal frequency characteristics. As a result, it loses some structural fidelity in non-salient areas, which has a significant impact on perception, affecting user studies and ratings.

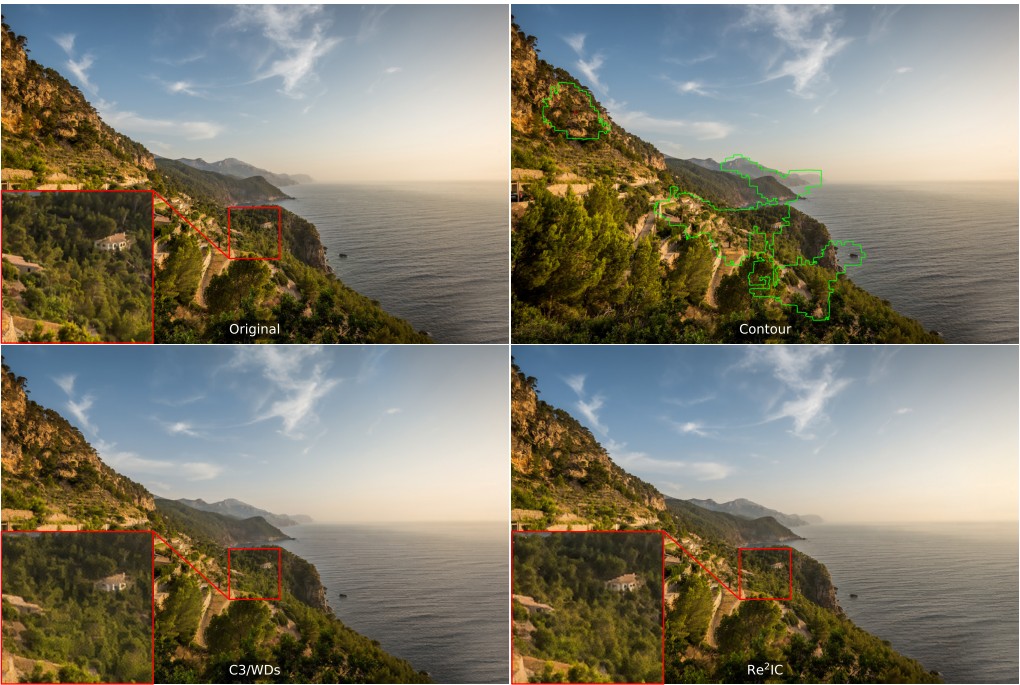

Figure 32: Visual comparison at low-bpp regime ($\leq 0.075$). C3/WDs tends to focus on fidelity in salient regions dominated by high frequencies, assigning lower $\sigma$ to salient areas while ignoring internal frequency characteristics. As a result, it struggles to capture high-frequency details under limited bit budgets, reducing perceptual quality.

