# OpenReview forum: "Re2IC: Realism-Enhanced Region-Based Implicit Codec with Wavelet–Wasserstein Distortion"
_ICLR.cc/2026/Conference — ICLR 2026 Conference Desk Rejected Submission_

### Official Review · Reviewer_mXUQ · 2025-10-20

**Soundness:** 3
**Presentation:** 3
**Contribution:** 2
**Rating:** 2
**Confidence:** 4

**Summary:**

## Summary
* This paper propose Re2IC, a INR based perceptual image codec.
* This paper also propose WA-WD, an improved WD (Wassertein distortion [Wasserstein distortion: Unifying fidelity and realism]) that aligns better with human preference.
* Despite Re2IC achieves better visual quality than C3-WD and HiFiC, the decoding time is significantly increased to 400ms.
* The result on WA-WD, on the other hand, looks more promising.

**Strengths:**

## Strength
* The proposed WA-WD seems to be an interesting addition in wassertein distortion. The authors successfully validate the effectiveness of WA-WD by empirical results (Table 2).

**Weaknesses:**

## Weakness
* The codec, Re2IC is overcomplicated, making the empirical result much less convincing.
  * The proposed Re2IC is composed of too many stages. The obvious consequence is, the decoding latency of Re2IC is too high. According to Table 4, the decoding latency of Re2IC is 400 ms. The reason why C3-WD makes sense is that it is much faster than many other perceptual image codec in terms of decoding time. However, for a perceptual codec with 400 ms decoding time, outperforming Re2IC and HiFiC is no longer good enough. In fact, Re2IC is already slower than many more recent GAN based codec, such as [MS-ILLM] and [TACO]. Besides, it is even slower or on par with recent diffusion based codec [StableCodec]. In that case, the empirical result in terms of human ELO score, is not convincing enough.
* The codec, Re2IC lacks necessary ablation studies.
  * The proposed WA-WD looks nice in performance. So the first thing the authors can do, is to train C3 by WA-WD and present the performance of C3-WA-WD. This ablation study is necessary as it validates the performance of WA-WD as a loss function instead of a IQA metric. Besides, if the result is good enough, it also solves the decoding latency issue of Re2IC as using WA-WD loss should not change the decoding complexity of C3-WD.
  * The design component of Re2IC laso lacks necessary ablation studies. For example, what if we remove global local design? The Re2IC composes of many compoents but non of them are validated by empirical results.

**Questions:**

## Questions
* How is the result of C3-WA-WD like? It both validates the effectiveness of WA-WD as loss function, and also validates the performance of  Re2IC vs C3 using the same loss function.
* How does each component of Re2IC contributes to the system individually?

---

> ### Author Response · Authors · 2025-11-27
> **Response to Reviewer [mXUQ] [Part I]**
>
> We thank the reviewer for their valuable comments. We now present detailed point-by-point responses to each of the comments.
>
> - **For W1:**
>   We would like to **respectfully clarify several possible misunderstandings**:
>
>   1. **Re2IC is not overcomplicated**
>
>      Re²IC follows a **similar overfitting pipeline** as C3 (or even with lower complexity, as evidenced in **Table 8, Page 32**). The additional perceptual components affect **ONLY** the training phase, they do **NOT increase** the decoding complexity, since all those modules are removed once training is complete.
>
>      For inference, the design of Re2IC is **EVEN simpler** than other counterpairs, such as C3/WD (e.g., we remove the conditional entropy coding strategy, adaptive codec setting, and allow more lightweight architectures thanks to our region-based design).
>
>      We note that the only component that may introduce additional cost is the local–global perception modeling module, which is standard in INR literature [R4.1] and can be parallelized in practice.
>
>   2. **Reported latency of Re2IC is on CPU**
>
>      We would like to clarify that the decoding latency reported for Re²IC is entirely on CPU. In contrast, the <400 ms latency reported for perceptual codecs such as TACO, MS-ILLM, and even StableCodec corresponds to GPU decoding time, which is **not comparable**.
>
>      Thus, the “400 ms” in Table 4 should not be interpreted as slow, only as CPU-based. For a more comprehensive analysis and understanding, we provide more baselines under same platform (**Table. 8,9; Page 32**) for latency analysis, where a coding cost breakdown is also provided.
>
>      We also emphasize that our current implementation is **purely research-oriented**. With proper engineering optimizations, such as symmetric/separable kernels, filter-based upsampling, and wavefront decoding [R4.2][R4.3], the latency in Table 8/9 can be reduced **substantially**. These optimizations are directly applicable to Re2IC and would further lower its inference cost.
>
>   3. **Re2IC yeilds fewer encoding steps and cheaper decoding**
>
>      In the end, we also want to highlight that our Re2IC provides an important advantage that **actually contrasts with the reviewer’s concern.** Rather than increasing decoding complexity, it actually yields both **faster** encoding convergence and **less** decoding complexity, potentially addressing one of the main challenges in making implicit codecs practical. To make this clearer, we include additional analysis as below in the revision:
>
>      - **Encoding efficiency.**
>        From an RD perspective, **Fig. 8 (page 10)** shows that Re2IC is able to achieve the same BD-rate as C3 using only **25%** of the encoding steps. As encoding cost is a major limitation of all overfitted codecs, this reduction constitutes a practically **significant contribution.**
>
>        From an RP perspective (**Fig. 8 on Page 10 and Fig. 22 on Page 41**), we observe an even **more pronounced** acceleration. Notably, Re2IC first overfits salient regions before progressively refining the remainder, a behavior that aligns naturally with our explicit region-wise design and reflects key characteristics of human visual perception.
>
>      - **Decoding efficiency.**
>        **Tables 8 (Page 32)** report the complexity and latency results. We observe that the decoding time does not increase significantly with the number of regions, and Re2IC maintains a decoding cost comparable to C3. Note that the current implementation decodes regions serially, but the process is inherently parallelizable, meaning the practical decoding latency can be further reduced with parallel execution.
>
>      In summary, Re2IC does **NOT** introduce extra decoding overhead; instead, it produces **computational gains while speeding up the encoding**, **directly addressing the concern raised by the reviewer**. For clarity, all referenced additional tables and figures have now been integrated into the revised manuscript.

---

> ### Author Response · Authors · 2025-11-27
> **Response to [mXUQ] [Part II]**
>
> - **For W2 (a):**
>   We thank the reviewer for acknowledging the promising performance of Re2IC and for suggesting this additional ablation. Following the suggestion, we have now included C3-WA-WD results in **Table. 11 on Page 33** and added ablation of each component **(Table 10 on Page 33)** into the revised manuscript.
>
>   This experiment in **Table. 10, page 33** demonstrates that WA-WD functions as an **architecture-agnostic** loss, providing **consistent gains** even when applied to a different overfitted codec design.
>
>   We would also like to **respectfully note that**: our original manuscript **also** contained an ablation isolating the contribution of WA-WD as a loss function. Specifically, **Fig 8 (c) on Page 10** evaluates: Re2IC/mse, Re2IC/WDs, and Re2IC/WA-WD and C3/WDs, which directly demonstrates the significant benefit contributed by WA-WD on top of WD.
>
>   Specifically, (1). Re2IC (WDs) and C3 (WDs) provide an ablation of the region-wise architecture. (2). Re2IC (MSE), Re2IC (WDs), and Re2IC (WA-WD) validate the effect of the loss function.  (3). Taken together, these experiments validate that each component of Re2IC (WA-WD and region-wise modeling), provides **independent and cumulative** performance gains.
>
>   Regarding decoding latency, we hope the explanation and experiments above address the concern. As the reviewer correctly pointed out, the perceptual optimization should not affect the inference, meaning Re2IC retains the same lightweight decoding advantages as C3/WD.
>
> - **For W2 (b):**
>   We thank the reviewer for the valuable suggestions. In the revised manuscript, we now provide a **detailed ablation study** for each component of our method. Please refer to **Tables 10,11,12 on page 33-35**, which reports the contribution of each module of Re2IC. These results clearly demonstrate the incremental effect of each component on the final RP performance.
>
>   We would like to note that ablation experiments for WD-based codecs are **computationally expensive**, as each variant requires careful bitrate control, and long training schedules. To support future research, we also provide a detailed ablation table in the revised manuscript (**Table 3 on Page 21**), including the exact experimental settings used for each component.
>
>   In summary, following the reviewer’s suggestions, we have incorporated a comprehensive set of ablations covering both architectural elements and loss functions. We hope this can fully address the reviewer’s concerns.
>
> We **welcome ANY further suggestions/comments** to strengthen the manuscript.
>
> - **Q1:**
>   We present detailed ablation studies in **Tables 11 on page 33**, showing that WA-WD can serve as a universal loss for other overfitted codecs for better RP performance.
>
> - **Q2:**
>   Contribution of each component is presented in **Tables 10,11,12 on page 33-35**. We refer to our response to the W1 (b) for more details.
>
>
> [R4.1: Modulated periodic activations for generalizable local functional representations, CVPR, 2021.]
>
> [R4.2] Blard. T, et al. "Overfitted image coding at reduced complexity." EUSIPCO, 2024.
>
> [R4.3] https://orange-opensource.github.io/Cool-Chic/

---

> > ### Comment · Reviewer_mXUQ · 2025-11-28
> >
> > Thanks for the additional clarification.
> >
> > First off, I would like to clarify two concepts: latency and throughput. Latency refers to the delay of a single image from received to decoded, it is measured in decoding time. Throughput refers to the maximum images can be processed per second given parallelization enabled. Throughput can be measured in MACs/pixel. The authors are emphasising that Re2IC is competitive in terms of decoding throughput (MACs). However, for most of the cases such as Streaming and real-time communication, the latency is what we care most.
> >
> > For now, my primary concern wrt to the decoding latency remains. Currently Table. 8 only shows decoding latency of the proposed approach on CPU, which is obviously inferior to C3 with GPU. I am not sure why the authors keep reporting the latency on CPU instead of GPU. Considering the overhead of chain coding, I am not sure whether the proposed approach has advantage in terms of decoding latency, compared with recent perceptual codec such as [MS-ILLM], [TACO] and [StableCodec]. Usually INR approaches such as COIN is able to achieve $\ge 100$ fps on GPU decoding. For now the chain coding part already takes $\ge 10$ ms, I am not sure whether the decoding latency of the proposed approach remains competitive on GPU.
> >
> > Besides, the additional results in Table 10, 11, 12 is not sufficient. FID is a metric that requires at least 1k-50k images / patches to be evaluated effectively. Usually we test FID on ImageNet / COCO validation split. The current FID in Table 10 and 11 is very high and seems to be computed with Kodak dataset. The FID computed with such small dataset is not reliable. The Table 12 is also highly unusual. The Kodak dataset is already very small. What is the point of using first 10 images of Kodak? I have never seen such evaluation before.

---

> ### Author Response · Authors · 2025-11-28
> **Response to further comment**
>
> Thanks for the reply, which enables discussion toward improving the paper.
> Here still exist several points that need to be clarified to avoid **misunderstandings**:
>
> **1. Clarifying misunderstanding over latency measurement and fairness**
>
> Real-world latency depends heavily on implementation details and system-level factors, which are difficult to control in a fair academic setting. With proper engineering, latency can vary drastically. For example, Cool-Chic [R4.4-4.5] reports less than <100 ms latency (CPU) using an optimized C-API arithmetic coder. This is why we report decoding complexity for a fair comparison from a **research perspective**.
>
> Currently, **ALL prior implicit codecs, [Cool-Chic (v1–v4)], [C3], [LotteryCodec], and others, are evaluated on CPU**. We therefore **follow this established/standard evaluation protocol** to ensure a fair and consistent comparison.
>
> [R4.4] Blard. T, et al. "Overfitted image coding at reduced complexity." EUSIPCO, 2024.
>
> [R4.5] https://orange-opensource.github.io/Cool-Chic/
>
> Importantly, the numbers of latency and throughput are **meaningful** only if all methods are evaluated under the **same platform, same hardware, and with equally optimized implementations.**
>
> As a conclusion, comparing our **CPU** latency with other **GPU**-accelerated generative codecs can be **misleading**. In fact, this contrast can **highlight our contribution**: despite running entirely on CPU, Re2IC achieves leading performance with decoding speeds comparable to other codecs with advanced GPU acceleration, underscoring the strength of its low-complexity design.
>
> **2. Clarifying a misunderstanding: “obviously inferior to C3 with GPU”**
>
> We believe there is an important misunderstanding here that may have affected the review. To the best of our knowledge, **all** existing implicit codecs, **including C3**, are evaluated **using CPU** decoding (**not GPU**). This is the established practice in the field, as implicit codecs typically decode **very fast** on GPUs.  Evaluating on CPU, therefore, provides a fair and stable latency comparison across methods.
>
> **3. Concerns for the chain-coding decoding cost**
>
> Chain-coding here is an explicit, highly efficient coding procedure. Even with our **non-optimized Python implementation on CPU**: Decoding cost is <12 ms for a large region of Kodak images, and a properly optimized implementation would even speed it up. Therefore, chain-coding efficiency should not be viewed as a limiting factor, especially considering its significant benefits for our perception modeling.
>
> **4. Misunderstandings for FPS concerns**
>
> The referred COIN series (GPU) with 100 FPS can also be **misunderstanding**. FPS or latency also depends on the input resolution (for example, COIN++ cannot handle Kodak in one pass, yeilding amplified latency). Given their reported CIFAR-10 decoding speed,100 fps for Kodak is infeasible. As a reference, **30 FPS with GPU A100** for Kodak can be **impressive** (Fig.20 in [R4.6]).
>
> [R4.6] EVC: TOWARDS REAL-TIME NEURAL IMAGE COMPRESSION WITH MASK DECAY
>
> **For a summary of latency concerns: CPU-based latency evaluation of implicit codecs is not only the established standard but also highlights their inherent advantage as lightweight solutions.**

---

> ### Author Response · Authors · 2025-12-03
> **Response to further comment (Part II)**
>
> **5. Two additional clarifications on potential misunderstandings**
>
> (5.1) Comment "FID for only 10 images is weak" and "using first 10 images of Kodak" **are incorrect**
>
> All RP experiments (including FID) are based on **FULL** datasets, strictly following prior work [R1] (**NOT only 10 imgs**).
>
> (Only one RD ablation over the entropy model uses 10 images due to resource constraints during revision, which is now also revised with full dataset results.)
>
> (5.2) FID evaluation protocol
>
> We **strictly follow the evaluation protocol of [R4.7]**, computing FID on patches from two datasets. (HiFiC uses the same patch-based protocol with the same size as ours, but on a different CLIC split.) For completeness, we report FID on both datasets and include **additional 8 perceptual metrics** for a more comprehensive comparison (note that ImageNet/COCO-scale FID are **infeasible** for implicit codec experiments). Since [R4.7] is one of our main baselines, this setup ensures the fairest and most consistent evaluation.
>
> Additionally, high absolute FID values are expected in such settings; in these cases, the relative differences are the meaningful indicator of perceptual quality, an evaluation practice also standard in image restoration works [R4.8, R4.9].
>
> [R4.7] High-Fidelity Generative Image Compression (NeurIPS)
>
> [R4.8] Seesr: Towards semantics-aware real-world image super-resolution, (CVPR). 2024.
>
> [R4.9] Regularization by Texts for Latent Diffusion Inverse Solvers, (ICLR). 2025.
>
> Nevertheless, we **appreciate** the reviewer’s comment and response. Following this, we have now **revised all these 3 tables** with additional DISTS/LPIPS metrics to thoroughly address the reviewer’s concerns.
>
> **For a summary of FID concerns: we strictly follow the established protocols, and additionally provide 8 complementary metrics to ensure a thorough and comprehensive evaluation, with all corresponding tables revised.**

---

### Official Review · Reviewer_NuCH · 2025-10-27

**Soundness:** 3
**Presentation:** 2
**Contribution:** 2
**Rating:** 4
**Confidence:** 3

**Summary:**

This paper addresses the high perceptual quality-low decoding complexity trade-off in image compression by proposing Re²IC (saliency-guided region-based implicit codec with local-global modulation) and WA-WD (wavelet-Wasserstein distortion, a human-aligned perceptual metric with 94.6% Pearson/92.3% Spearman correlations). Experiments on Kodak/CLIC2020 show Re²IC outperforms HiFiC (rate-perception) with <1% decoding cost and surpasses overfitted codecs while accelerating convergence—its key contributions lie in Re²IC’s low-complexity high-performance design, spatiotemporal optimization via integrated modules, new SOTA for overfitted codecs, and WA-WD’s tunable perceptual utility.

**Strengths:**

This paper demonstrates notable strengths across originality, quality, clarity, and importance: In originality, it exhibits meaningful innovation by addressing the "perceptual quality-decoding complexity" dilemma in image compression—proposing Re²IC  with saliency-guided region-based implicit modeling (a novel take on balancing local fitting and global context via LPN-GPM collaboration) and wavelet-Wasserstein distortion (WA-WD), which innovatively integrates wavelet frequency decomposition with Wasserstein distance to capture feature correlations ignored by traditional metrics, while also functioning as a standalone human-aligned perceptual indicator. In quality, its experimental design is rigorous and high-caliber: it covers diverse datasets (Kodak, CLIC2020), comprehensive baselines (traditional VTM, autoencoder-based MLIC++, generative HiFiC, overfitted C3/WDs), and multi-faceted evaluations (quantitative metrics like MACs/pixel, LPIPS, and PSNR; qualitative visual comparisons across scenes; subjective user studies with Elo scoring), ensuring reliable, robust validation of Re²IC ’s superiority (outperforming HiFiC with <1% decoding cost). In clarity, the paper is highly accessible: it structures technical content logically to unpack complex architectures/metrics, and provides detailed appendices (implementation details, ablation studies) to clarify design choices, making core innovations and results easy to follow.

**Weaknesses:**

While this paper has achieved notable results in the field of low-complexity perceptual image compression, it still has four key shortcomings: First, at the level of core optimization technology, the idea of using wavelet transform to achieve frequency-aware optimization bears high similarity to existing related works—these works separate frequency-domain features through multi-scale transforms (e.g., wavelets, wavelet packets) to balance fidelity and perceptual quality, making the innovation less prominent in terms of technical pathways. A case in point is the paper WeConvene: Learned Image Compression with Wavelet-Domain Convolution and Entropy Mode. Second, there is an obvious imbalance in complexity optimization: although it focuses on achieving a significant reduction in decoding complexity (less than 1% of HiFiC’s), it fails to perform synchronous optimization and quantitative analysis of encoding complexity. Third, the coverage of dataset scenarios is insufficient, as experiments are only validated on two types of static image datasets, namely Kodak and CLIC2020.

**Questions:**

Conduct a fine-grained comparative analysis with representative works such as WeConvene: for example, in a dedicated "Technical Comparison" section, compare the two types of works across dimensions including "wavelet decomposition levels (single-level vs. multi-level)", "frequency-domain optimization objectives (fidelity-only vs. fidelity-perception balance)", and "integration methods with entropy coding (independent optimization vs. joint optimization)", while highlighting innovative points such as how the WA-WD proposed in this paper captures feature correlations that are ignored by wavelet-domain convolution in WeConvene. Supplement systematic encoding complexity measurement: add indicators such as "encoding latency (milliseconds)" , "encoding MACs/pixel", and "per-image training time" to the experimental results, and integrate these indicators with decoding complexity into the same table to facilitate intuitive comparison. Expand the scope of datasets for static image testing: incorporate representative new datasets.

---

> ### Author Response · Authors · 2025-11-27
> **Response to Reviewer [NuCH]**
>
> We thank the reviewer for their valuable comments and we now present detailed point-by-point responses to each of the questions.
>
> 1. **W1.**
>    We thank the reviewer for pointing out this interesting and relevant line of work. While both methods employ wavelet transforms, their roles, objectives, and technical pathways are **fundamentally different**. We explain this from two perspectives:
>
>    (1) **Different compression paradigms with DWT.** Prior wavelet-based codecs use the DWT as an **analysis transform** to decorrelate latent features, thereby producing sparser subbands for improved coding efficiency. **Their goal is to obtain better distribution learning for the entropy model.**
>
>    In contrast, Re2IC is an implicit codec. It performs **NO** transform coding, and does **NOT** learn a distribution or perform feature extraction. Instead, it overfits each individual image and **directly models human perception** at the instance level. Thus, DWT in Re2IC is not used to improve compression of latent codes, but to improve the **perceptual optimization metric itself**.
>
>    (2) **Different purpose of DWT for WA-WD.**
>    In Re2IC, DWT is introduced to:
>
>    - **Improve the statistical validity of WD approximation.** DWT reduces cross-correlations for diagonal-Gaussian approximation, making WD approximation more stable and accurate (see **Corollary 1 on page 5**).
>    - **Provide a tunable perceptual metric.** As shown in **Table II** and **Fig. 7 on page 9**, WA-WD can function as an effective perceptual metric and enables flexible control of the fidelity–realism balance through adjustable subband weights, which further strengthens perceptual optimization for the implicit codec.
>    - **Enables spatial–frequency–aware perceptual overfitting.**
>      Together with our region-wise overfitting strategy, DWT-based WA-WD models perception jointly in spatial (saliency) and frequency (subband structure) domains, which is crucial for realism-enhanced implicit codecs (**as evidenced in Figs. 31–32 on page 47**).
>
>    Thus, although both methods incorporate DWT, Re2IC is fundamentally different in purpose and formulation. Our method introduces a perception-guided, wavelet-domain distortion metric **tailored for implicit codecs**. These distinctions are discussed in **Appendix F.4 (page 34--38)**, and we further include comparisons with WeConvene in **(Table 13 on page 35; Figs. 15-17 on page 36-37; Fig. 8 on page 10)**, given their relevance.
>
> 2. **W2.**
>    The reviewer is correct. As acknowledged in the limitations section, high encoding complexity remains a fundamental bottleneck for **all overfitted codecs**, including ours. These methods are therefore most appropriate for offline encoding/online decoding scenarios, such as streaming, where the content is **encoded once but decoded many times**.
>
>    **In fact, this concern also highlights one of our core contributions**. Our region-based perception modeling strategy assigns different regions to distinct neural functions, allowing each region to be optimized with a more homogeneous distribution. This significantly **accelerates convergence** and reduces encoding cost: **Re2IC achieves the similar RD/RP performance as C3 using only 20-30% encoding steps** (see **Fig. 8 on Page 10**). We believe this **improvement is critical for moving implicit codecs toward practical deployment**.
>
>    To clarify this point, we have added the following text to the manuscript (**page 39**) and added **Fig. 8 on page 10** in our main body to **highlight our contribution for faster encoding**:
>
>    “The low decoding cost of Re$^2$IC is particularly beneficial in multi-user streaming scenarios, where encoding is performed once offline and many users decode the same content. This remains an inherent limitation of all overfitted codecs, including C3, Cool-Chic, and LotteryCodec. At the same time, our region-wise perception modeling strategy offers a promising direction for accelerating encoding: by decomposing the image into more homogeneous regions, optimization becomes significantly faster and more stable. This can be further improved through complementary techniques such as meta-learning or mixed-precision training, which we expect will substantially reduce encoding time in future work.”

---

> ### Author Response · Authors · 2025-11-27
> **Response to NuCH**
>
> 3. **W3.**
>    For evaluation, we adopt two standard and widely used benchmarks (Kodak and CLIC2020) in implicit codec research, **same to [R3.1-R3.4]**:
>
>    [R3.1] Ladune, Théo, et al. "Cool-chic: Coordinate-based low complexity hierarchical image codec." CVPR. 2023.
>    [R3.2] Kim, Hyunjik, et al. "C3: High-performance and low-complexity neural compression from a single image or video." CVPR. 2024.
>    [R3.3] Wu, Haotian, et al. "LotteryCodec: Searching the Implicit Representation in a Random Network for Low-Complexity Image Compression." ICML. 2025.
>    [R3.4] Balle, Jona, et al. "Good, cheap, and fast: Overfitted image compression with Wasserstein distortion." CVPR. 2025.
>
>    Developing implicit codecs on additional datasets (such as Tecnick) is not common, as it requires substantial computational resources, often leading to development cycles spanning **many months**. Unlike autoencoder-based codecs, implicit codecs cannot reuse checkpoints across datasets; all baselines must be **retrained**, making large-scale cross-dataset comparisons **infeasible** from a research perspective. For fairness, we therefore use each method’s **officially reported optimal results (on Kodak and CLIC2020)**, as in all previous literature.
>
>    This computational challenge is even **more severe** for perceptual codecs: for instance, the most recent perception-oriented implicit codec [R3.4] reports results **only on CLIC2020**. Following this setting, we strictly adopt the Challenge on Learned Image Compression (CLIC) human-rating protocol (dataset, evaluation model, and bitrate alignment), which provides one of the **most established, carefully designed human-judgment frameworks** for learned image compression, ensuring fairness, reproducibility, and comparability. For completeness, we **additionally include perceptual comparisons on Kodak**, **going beyond [R3.4]**.
>
>    Additionally, **user-study feasibility** also imposes practical limitations for more datasets: currently, each Elo study requires more than **4 hours** of human ratings for each user. Using larger datasets would significantly increase fatigue and variance, undermining evaluation reliability.
>
>    We hope this clarifies that our evaluation is not sparse, but rather **follows the standardized human-rating protocol** used in perceptual compression and implicit codecs, **fully consistent or going beyond** with recent literature [R3.1–R3.4].
>
>    However, we agree with the reviewer on the benefits of broadening the evaluation scope, and therefore **additionally include three IQA datasets (Fig. 7 on page 9)** in the revised manuscript. These IQA results complement the compression studies and **further strengthen the value of WA–WD as a standalone perceptual metric**, beyond the image compression setting.
>
> 4. **Q1.**
>    An additional section "Technical Comparison for DWT" has been added on **page 34-38.**, where 'Comparison with other DWT-based codecs', 'Effect of joint optimized entropy models', 'Visualization of Feature Correlations', 'Frequency-domain optimization objectives', 'effect of each sub-band', and 'wavelet decomposition levels' are all incorporated.
>
> 5. **Q2.**
>    Encoding and decoding complexity, as well as runtime latency, are now reported in **Tables 8–9, page 32**. For datasets, we follow the standard practice in implicit codecs on **Kodak and CLIC**, where comparable overfitted codec baselines are available. Meanwhile, we additionally include **three IQA datasets (Fig. 7, page 9)** to more comprehensively assess WA-WD as a perceptual metric.
>
> **In summary, we clarified the conceptual and technical differences from prior DWT-based codecs (with added baselines), explained the rationale for datasets, and demonstrated Re2IC's encoding advantages. We also added the requested dedicated technical comparison section, comprehensive complexity evaluations, detailed ablations, and three additional IQA datasets.**

---

### Official Review · Reviewer_SiBX · 2025-10-27

**Soundness:** 2
**Presentation:** 2
**Contribution:** 2
**Rating:** 2
**Confidence:** 4

**Summary:**

The paper introduces a realism-oriented codec based on wavelet–Wasserstein distortion (WA-WD), a new version of the recently proposes Wasserstein distortion. The authors aim to achieve high perceptual quality at low decoding complexity. The key contributions are (1) a region-wise perceptual modeling strategy using local perceptual networks (LPNs), which are modulated by a global perceptual modulator (GPM), and (2) a frequency-aware distortion metric, WA-WD, which decomposes Wasserstein distortion across wavelet subbands. Experiments show that Re2IC is competitive with baselines such as C3/WDs and HiFiC in terms of rate–perception-distortion performance, while maintaining lower decoding costs. The authors claim that the WA-WD can act as an effective perceptual metric.

**Strengths:**

1. The paper tackles an important problem: perceptual image compression at low computational cost. This is a longstanding research problem that remains unsolved.
2. The authors provide several comparisons with previous compression schemes, yielding compelling rate-perception-distortion results at low computational costs.
3. The introduction of frequency-aware control within the Wasserstein-distortion framework is an interesting and valuable direction.
4. WA-WD may be an interesting new kind of perceptual metric.

**Weaknesses:**

1. The paper's presentation and writing clarity are, in my opinion, insufficient. The mathematical exposition is dense and unstructured. Key definitions are not explained well enough, such as the Wasserstein distortion-where does it come from, why is it important, and why the authors chose to build on top of it.
2. Weak theoretical grounding. The motivation for decomposing the Wasserstein distortion via wavelets lacks justification beyond intuition. No formal analysis shows why this approach improves perceptual correlation.
3. Unclear interpretation of WA-WD. The paper does not explain what WA-WD measures in perceptual terms- how it connects to human visual sensitivity, and how the $\sigma$-map should be understood intuitively?
4. The paper reads more like a collection of formulas and results rather than a clear narrative explaining why each design choice is beneficial.
5. Incomplete discussion on related work. For example, the authors omit recent diffusion-based compression papers [1,2,3], which are essential for contextualizing the contribution. The authors should discuss, and perhaps even compare performance against these, highlighting the trade-off between computation cost and performance.
6. The authors propose a perceptual metric and claim high correlation with human judgments, but the evaluation is sparse and unsatisfying. I also don't see a reference to Table 2 from the paper, which shows the main results. Since the authors claim a highly performant perceptual metric in the abstract, I expected thorough evaluations showing that it's indeed highly performant. However, the evaluations are conducted only on one dataset, without any explanation regarding the nature of the experiment (sec. 4.3).


[1] Yang and Mandt. Lossy Image Compression with Conditional Diffusion Models. In NeurIPS 2023.

[2] Vonderfecht and Liu. Lossy Compression with Pretrained Diffusion Models. In ICLR 2025.

[3] Ohayon et al. Compressed Image Generation with Denoising Diffusion Codebook Models. In ICML 2025.

**Questions:**

What is the performance of the perceptual metric on common datasets such as LIVE, TID2013, KADID10k, BAPPS, etc.?

---

> ### Author Response · Authors · 2025-11-27
> **Response to Reviewer [SiBX]**
>
> We thank the reviewer for the valuable comments and have revised the manuscript accordingly, substantially improving its clarity, structure, and theoretical presentation. Before providing the point-by-point responses,  we would like to **respectfully clarify several possible misunderstandings**:
>
> - **Main contributions.** The core contribution of this work is a realism-enhanced implicit codec that achieves **faster convergence, superior rate–perception trade-off, and low decoding complexity**. Re2IC is the **first** implicit codec to surpass the generative-based HiFiC in RP performance with **substantially lower** decoding cost,  **directly targeting the challenge noted by the reviewer**.
>
>   Moreover, **encoding cost**, a major bottleneck for all overfitted codecs, is substantially reduced by Re2IC, which requires only **20-30%** encoding steps used in prior methods. This **significantly** improves practicality and moves implicit codecs closer to real-world deployment.
>
> - **Purpose of perceptual-metric evaluation.** The perceptual-metric analysis in Section 4.3 is performed in the context of **compression-judgment**, using CLIC human rating protocol. This differs fundamentally from general IQA benchmarks, which contain diverse synthetic degradations outside the scope of perceptual compression. Therefore, Table 2 is **NOT** intended to correspond to **any** existing IQA dataset. It measures alignment with the human-rating results from [R2.3], which is the most relevant human-alignment ground-truth **for perceptual compression**. We believe this setup offers the **fairest and most relevant comparison**, since WD is the optimal reference point here for evaluating whether WA-WD provides further improvements.
>
>   However, to address the reviewer’s concern, we now also include **comprehensive evaluations on standard IQA datasets**, showing that WA-WD **generalizes well beyond compression** and can function as a strong general perceptual metric. We thank the reviewer for this suggestion, which **further strengthens our contribution**.
>
> - **Per-sample perceptual modeling insight.** We would like to clarify that the design of Re2IC differs fundamentally from other perceptual codecs. Instead of learning a generative distribution, Re2IC **directly models human perception** sample by sample (evidenced by SRCC/PCC), thereby enabling high perceptual quality at extremely low decoding cost. (This is also why we conduct perceptual-metric evaluation specifically under perceptual-compression settings from [R2.3].)
>
>   Because WA-WD is **tunable** (subband weights or $\sigma$-ranges), this also opens a **new design avenue** for implicit codecs: **first select or tune** the perception parameters for the target application or bpp regimes, and **then overfit** the codec to this parameterized perception model. This two-stage approach achieves both **low overhead** and **strong perceptual performance**, and provides flexibility for different use cases.

---

> ### Author Response · Authors · 2025-11-27
> **Detailed response to each comment of Reviewer [SiBX]**
>
> Then, we present detailed responses to each comment.
>
> 1. **W1.**
>    We have substantially **reorganized Section 3.1 (page 4)** to improve clarity. In a nutshell, WD measures perceptual differences in feature space by explicitly modelling foveal and peripheral sensitivity through a spatially varying $\sigma$-map, which is typically derived from saliency map. Such $\sigma$-map modulates tolerance to texture resampling, thereby enabling flexible, perception-guided bit allocation in image compression.
>
>    Furthermore, we have additionally provided **detailed definition with derivations, interpretations, illustrative figures, theoretical properties, and practical implementation details** in Appendix D.1 (**page 22-23**). We hope those revisions address the reviewer’s concerns regarding clarity and presentation.
>
> 2. **W2.**
>    In the revised manuscript, we have substantially strengthened the theoretical grounding of WA-WD.
>    Our motivation of wavelet-domain decomposition is now supported by **two formal results** and **one statistical analysis,** complemented by **new ablation studies**.
>    A brief summary is provided below, and full details are given in Section 3.2 and Appendix E (**page 5 and pages 26-30**) of the manuscript.
>
>    - **Metric preservation (Theorem 1)**  : We prove that WA-WD preserves the metric structure of the original WD.
>    - **Tighter diagonal-Gaussian approximation (Corollary 1 and Lemma 1)**: In practical implementations, WD relies on a diagonal-Gaussian independence assumption, which typically does not hold in neural feature space. We show that the resultant approximation error for distortion becomes tighter in the wavelet domain, making WA-WD a statistically more accurate and reliable approximation for implicit codec optimization.
>    - **Improved variance modeling and correlation sensitivity (Corollary 2)**. We further analyze the variance structure for both WD and WA-WD, and show that WA-WD explicitly captures intra-block correlations, whereas standard WD ignores these correlations under a diagonal-Gaussian assumption. This explains why WA-WD aligns better with human perception, which is highly sensitive to frequency-specific and correlation-driven distortions.
>    - We also revised the manuscript with additional tables for comparison (**Tables 4, 5 on Page 25 and 31**), figures for illustration (**Fig. 12 on Page 24**), and experiments for validation (**Fig. 15 on Page 36**) in Appendix D.2.
>    - Finally, we conduct ablation studies on each component and wavelet subband, along with general IQA evaluations (**Fig. 7 on page 9; Tables 10, 11 on page 33; Figs. 18,19 on page 38**), all consistently validate the benefits of WA-WD beyond intuition.

---

> ### Author Response · Authors · 2025-11-27
> **Detailed response to each comment of Reviewer [SiBX] [Part II]**
>
> 3. **W3.**
>    In the revised manuscript, we have substantially clarified the perceptual meaning of WA-WD. The key points are summarized below:
>
>    - **What WA-WD measures perceptually.** Unlike WD, which mixes all frequencies, WA-WD decomposes features into structural (LL) and textural (LH/HL/HH) subbands. Thus, WA-WD measures perceptual differences in *multi-scale, frequency-decomposed feature statistics*, producing frequency-aware distortion signals that better align with human perception and enable finer regional control for Re2IC compression.
>      This design is also supported by findings in visual neuroscience [R2.1]. Early visual processing involves two stages: (i) V1 performs multi-scale, orientation-selective filtering, which is closely approximated by wavelet subbands; and (ii) V2 aggregates local statistics through spatial pooling, analogous to WD’s $\sigma$-dependent pooling. Although implemented in feature space, WA-WD mirrors this two-stage HVS structure, providing an interpretable perceptual motivation.
>
>    - **Intuitive meaning of the $\sigma$-map.** $\sigma$ controls the size of the pooling window. For small $\sigma$, WA-WD models foveal vision under different sub-bands, which enforces high-fidelity reconstruction. For large $\sigma$, it models peripheral vision with more tolerance to texture resampling. Thus, $\sigma$ directly controls the **realism–fidelity trade-off** in a way that mirrors human visual sensitivity.
>
>    To further improve interpretability, we include **new diagrams, explanations, and experimental ablations demonstrating**. All the above statements can be found in our revised manuscripts (Section 3.2 **on pages 4-5** and Appendix D.2 on **pages 23–31**).
>
> 4. **W4.**
>    We have reorganized the manuscript and added a comprehensive ablation study in **Tables 10, 11, 12 on page 33, 35; Figs. 18-19 on page 38** to validate the effect of each sub-band for Re2IC. Experiments demonstrate the cumulative performance gains from both the architectural design and the WA-WD formulation (also see **Fig. 8 on page 10**).
>
> 5. **W5.**
>    In the revised manuscript, we added a related-work section to summarize recent diffusion-based codecs (**page 3**). **Additional baselines** such as **TACO** [R2.4] and **CDC** [R2.2] (**diffusion-based** models in **Tables 6 and 7 on page 31-32**) and WeConvene (**DWT-based** model, **Table 13 and Figs. 15,16,17 on page 35-36**) are also included.
>
>    We added the following sentences to the paper on **page 31**:
>
>    "Our comparison focuses on low-complexity implicit codecs, whose rate–perception–complexity characteristics differ fundamentally from diffusion- or GAN-based methods. Although diffusion models achieve high perceptual scores on metrics like MS-SSIM or LPIPS, Re2IC delivers competitive or even superior FID/DISTS/KID/NIQE/CLIP-IQA/MANIQA performance with far lower complexity, highlighting the effectiveness of implicit codecs for perceptual modeling."
>
> 6. **W6.**
>    As clarified earlier, Table 2 is **not intended as a general IQA benchmark**; it evaluates alignment with the human-rating (Elo) results for compression from [R2.3], which is the appropriate setting for validating perceptual modeling in implicit codecs.
>
>    To fully address the reviewer’s concern, we additionally evaluate WA-WD on **standard IQA datasets (CSIQ, TID2013, LIVE)**, showing that it **generalizes well beyond compression**-specific settings and achieves **leading performance** (see **Fig. 7, page 9**).
>
>    We also note that these strong results are obtained using only a **plain VGG backbone**, and even **significantly higher performance** is expected with improved backbones or tuned subband weights.
>
> 7. **Q1.**
>    Following the suggestion, we have **added IQA experiments** on **three standard datasets (CSIQ, TID2013, LIVE)**. Details are provided in our response to W6.
>
>
> [R2.1] A summary statistic representation in peripheral vision explains visual crowding. Journal of Vision, 2009.
>
> [R2.2] Lossy Image Compression with Conditional Diffusion Models, NeurIPS, 2023
>
> [R2.3] Good, Cheap, and Fast: Overfitted Image Compression with Wasserstein Distortion, CVPR, 2025.
>
> [R2.4] Neural Image Compression with Text-guided Encoding for both Pixel-level and Perceptual Fidelity, ICML, 2024
>
> **In summary, we significantly reorganized the related sections, added the requested formal theoretical analysis with expanded explanations and motivations, and incorporated additional related work, baselines, and experiments. We clarified the misunderstandings regarding our previous perception-metric evaluation and included experiments in three new IQA datasets.**

---

### Official Review · Reviewer_3Qcj · 2025-10-29

**Soundness:** 3
**Presentation:** 3
**Contribution:** 3
**Rating:** 6
**Confidence:** 4

**Summary:**

The paper proposes Re2IC, a Realism-Enhanced Region-Based Implicit Codec that achieves perceptual image compression at low decoding complexity. The method integrates saliency-guided regional modeling with wavelet–Wasserstein distortion (WA-WD), enabling fine-grained spatial–spectral control for perception–fidelity balance. Experiments and human evaluations demonstrate that Re2IC achieves superior rate–perception trade-offs compared with both overfitted and generative codecs, while maintaining extremely low complexity.

**Strengths:**

1. The use of human preference evaluation alongside quantitative metrics is reasonable and supports the argument that WA-WD aligns more closely with perceptual quality.
2. Compared with previous INR-based perceptual compression methods, the proposed approach demonstrates clear improvement in both realism and efficiency.

**Weaknesses:**

1. The paper does not include comparisons with several recent perceptual optimization methods.

**Questions:**

1. The authors should compare Re2IC with recent perceptually optimized methods such as TACO, Diffusion-based codecs, to strengthen the empirical validation.
2. In Section 4.1, the baseline C3/WDs is mentioned. Does this refer to Balle et al. (2025) or to the original C3 (Kim et al., 2024)? Please clarify the specific implementation used and ensure proper citation consistency.

---

> ### Author Response · Authors · 2025-11-27
> **Response to Reviewer [3Qcj]**
>
> We thank the reviewer for the valuable comments. Following the suggestions, we provide detailed, point-by-point responses to all concerns below.
>
> - **W1.**
>   - (1). We have added a **new subsection** summarizing recent diffusion-based perceptual codecs (**page 3**).
>   - (2). We have also included **additional baselines**, including diffusion-based codecs (**CDC and TACO**), and the DWT-based codec **WeConvene**. Results are presented in **Tables 6–7 on page 31 and 32**. We can observe that the proposed Re2IC can still deliver **leading RP performance** in various perception metrics (such as FID/DISTS/KID/CLIPIQA/MANIQA).
>   - (3).  In addition to the above new experiments, we have conducted **new IQA and human-alignment experiments** (**Fig. 7 on page 9**), demonstrating that the proposed WA-WD holds independent value as a perceptual metric with leading performance.
>
> - **Q1.** We thank the reviewer for the suggestion and have added these **new baselines**. For further details, please refer to our response to W1.
>
> - **Q2.** The reviewer is correct, and we thank the reviewer for bringing this to our attention. All perception-oriented baselines (C3/WDs) correspond to Ballé et al. (2025), while the fidelity-oriented baseline (C3/MSE) follows Kim et al. (2024). We have now corrected the notation and added explicit citations throughout the manuscript.
>
> **In summary, we have revised the manuscript to include the requested new baselines and the corresponding related-work section.**
>
> We hope the above clarifications can address the reviewer’s concerns, and we warmly welcome any further questions or suggestions to help us improve the manuscript.

---

### Author Response · Authors · 2025-11-27
**Summary of the revision**

We thank reviewers for their valuable comments. We have carefully revised the manuscript and believe it is **now significantly improved**. The revisions are **highlighted in blue** in our updated manuscript.

Before addressing each comment in detail, we summarize the major changes made in this version.

- A1. We revised the related section and added a dedicated section on Wasserstein distortion (WD) and wavelet–Wasserstein distortion (WA-WD), including **their definitions, motivations, interpretations, theoretical properties, implementation details, along with additional tables and figures** for better clarity (**page 22–26**).

- A2. To better demonstrate performance gains, we complement the extensive experiments with **additional theoretical analyses** (Theorem 1 and Corollaries 1–2 on **page 5** with their proof on **page 26-30**). These results formally show that the DWT makes the approximation of WA-WD **more accurate** during implicit-codec optimization, thereby improving stability and perceptual effectiveness.

- A3. We added a **new subsection** on perceptual-optimized codecs (**page 3**), including recent diffusion-based methods. We also incorporated **additional baselines**, such as diffusion-based (TACO, CDC) and DWT-based baselines (WeConcene). See **Tables 6 and 7, on page 31 and 32; Fig.8 on page 10; Table 13 on Page 35; Fig. 16-17 on Page 37** for details.

- A4. We have added **comprehensive ablation studies**, such as **effect of each component** (**Table 10 on Page 33**), **effect of the loss function for different codecs** (**Table 11 on Page 33**), **theoretical complexity and practical latency** of decoding/encoding process (**Table 8/9 on Page 32**), **design of entropy models** (**Table 12 on Page 35**), **comparasion with other DWT-based approaches** (**Table 13 on page 35 and Figs. 16,17 on page 37**), and **effect of each sub-bands (Fig. 15 on page 36 and Figs. 18,19 on page 38)**.

    We would like to highlight that fair ablation studies are **computationally expensive**, particularly due to the need to search for a target bpp for different scenarios. For transparency and reproducibility, we provide all detailed settings in **Table 3, Page 21**.

- A5. The benefits of **significantly reduced encoding steps** are highlighted in **Fig. 8 (Page 10), alongside Fig 22 on page 41**, offering a new solution to one of the major challenges faced by all implicit codecs.

- A6. We have added new experiments on **image quality assessment on various datasets (CSIQ, TID2013, and LIVE )** (**Fig. 7 on Page 9**). The results show that WA-WD achieves leading performance, even when using only a plain VGG backbone and untuned subband weights. Further improvements with perceptual-oriented backbones and optimized subband weighting can be expected. We believe this **significantly strengthens our contribution**.

- A7. To illustrate the benefits of tunable WA-WD, we report results from both the equal-weight and the tuned version of Re2IC on Kodak (**Table 2, page 9; Fig. 14, page 34**), demonstrating its flexible distortion–perception trade-off.

- A8. Given the inherent subjectivity of user studies and the open challenges in perceptual quality assessment, we provide detailed **one-by-one perceptual comparisons** in supplementary materials. An interactive interface is included (**Fig 9 on Page 17**), allowing reviewers to flip between methods for easy, one-by-one visual comparisons.

In summary, we thank reviewers for their valuable suggestions and comments. The revised manuscript now includes **broader evaluations**, richer visualizations, **strengthened theoretical analysis**, **expanded qualitative and quantitative studies**, and **new IQA experiments**. Together, these additions substantially broaden the scope and depth of the work. We hope the revision now offers a clearer and more complete presentation of our contributions.

 **We welcome any further suggestions** to improve the paper within its intended scope.

---

> ### Author Response · Authors · 2025-12-03
> **Summary of response for each reviewer**
>
> We sincerely thank the ACs and all reviewers for their time and constructive feedback. Their comments have led to substantial improvements to the paper. Below, we briefly summarize the main revisions made in response to each reviewer, to help the AC and reviewers efficiently go through the changes.
>
> - **To Reviewer [3Qcj]:** We added a new subsection introducing recent perceptual codecs and included corresponding suggested additional baseline experiments.
>
>   **Main related revisions: [A3],[A6]**
>
> - **To Reviewer [SiBX]:** We significantly reorganized Sections 3–4 for clarity; added suggested formal theoretical analysis; expanded explanation and motivation for methods; added related works with additional baselines. Following the suggestions, we have added three new IQA datasets (CSIQ, TID2013, LIVE) and clarified the misunderstandings over our previous perception metric evaluation, and conducted comprehensive ablations to support WA-WD’s effectiveness.
>
>   **Main related revision: [A1],[A2],[A3],[A4],[A6],[A7],[A8]**
>
> - **To Reviewer [NuCH]:** We clarified the conceptual and technical differences between our method and prior DWT-based codecs (and added corresponding DWT-codec baselines), and explained the rationale for evaluating on Kodak and CLIC2020. We demonstrated Re2IC’s significant encoding advantages as an implicit codec. Following the reviewer’s suggestions, we added a dedicated technical comparison section including all requested analyses and experiments, expanded the manuscript with comprehensive complexity evaluations (encoding/decoding latency and MACs/pixel), detailed ablation studies, and incorporated three additional IQA datasets.
>
>   **Main related revision: [A3],[A4],[A5],[A6],[A7]**
>
> - **To Reviewer [mXUQ]:** We have added all requested ablation studies, expanded the evaluation tables with additional perceptual metrics, clarified that CPU latency reporting is the standard practice for all implicit codecs, and explained our patch-based FID protocol in alignment with prior baselines. These revisions also validate that the proposed method achieves significantly faster encoding and cheaper decoding.
>
>   **Main related revision: [A4],[A5],[A7],[A8]**
>
> Across all reviews, we have **carefully addressed all concerns** and **significantly strengthened** the manuscript with **new baselines, theory, clarity, ablations, and experiments**.

---

### Note · Program_Chairs · 2026-01-17
**Submission Desk Rejected by Program Chairs**

The following references in this submission do not refer to real documents and/or have major errors in bibliographic information:

 Zhengzhong Tu, Yilin Wang, Jianyi Wang, Chao Ma, and Alan C Bovik. Maniqa: Multi-dimension attention network for no-reference image quality assessment. In Proceedings of the IEEE/CVF Conference on Computer Vision and Pattern Recognition, pp. 1191-1200, 2023.